# High-resolution mapping of time since disturbance and forest carbon flux from remote sensing and inventory data to assess harvest, fire, and beetle disturbance legacies in the Pacific Northwest

Huan Gu[1], Christopher A. Williams[1], Bardan Ghimire[1,2], Feng Zhao[3], Chengquan Huang[3]

[1]Graduate School of Geography, Clark University, Worcester, MA 01610, USA
[2]Earth Sciences Division, Lawrence Berkeley National Laboratory, Berkeley, CA 94720, USA
[3]Department of Geographical Sciences, University of Maryland, College Park, MD 20742, USA

*Correspondence to*: Huan Gu (HuGu@clarku.edu, guhuan114031@gmail.com)

**Abstract.** Accurate assessment of forest carbon storage and uptake is central to understanding the role forests play in the global carbon cycle and policy-making aimed at mitigating climate change. Disturbances have highly diverse impacts on forest carbon dynamics, representing a challenge to quantify and report. Time since disturbance is a key intermediate determinant that aided the assessment of disturbance-driven carbon emissions and removals legacies. We proposed a new methodology of quantifying time since disturbance and carbon flux across forested landscapes in the Pacific Northwest (PNW) at a fine scale (30 m) by combining remote sensing (RS) based disturbance year, disturbance type, and aboveground biomass with forest inventory data. When a recent disturbance was detected, time since disturbance can be directly determined by combination of three RS-derived disturbance products; and if not, time since last stand-clearing was inferred from RS-derived 30 m biomass map and field inventory-derived species-specific biomass accumulation curves. Net ecosystem productivity (*NEP*) was further mapped based on carbon stock and flux trajectories derived from Carnegie-Ames-Stanford Approach (CASA) model in our prior work that described how *NEP* changes with time following harvest, fire, or bark beetle disturbances of varying severity. Uncertainties from biomass map and forest inventory data were propagated by probabilistic sampling to provide a statistical distribution of stand age and *NEP* for each forest pixel. We mapped mean, standard deviation and statistical distribution of stand age and *NEP* at 30 m in the PNW region. Our map indicated a net ecosystem productivity of 5.9 Tg C y$^{-1}$ for forestlands circa 2010 in the study area, with net uptake in relatively mature (>24 year old) forests (13.6 Tg C y$^{-1}$) overwhelming net negative NEP from tracts that have recent harvest (-6.4 Tg C y$^{-1}$), fires (-0.5 Tg C y$^{-1}$), and bark beetle outbreaks (-0.8 Tg C y$^{-1}$). The approach will be applied to forestlands in other regions of the conterminous US to advance a more comprehensive monitoring, mapping and reporting the carbon consequences of forest change across the US.

# 1 Introduction

Disturbances profoundly alter ecosystems often with legacies that persist for decades to centuries (Turner, 2010). Correspondingly, time since disturbance is a key determinant of ecosystem structure, composition, and function (Jenny, 1980; Chapin et al., 2012). It is also a primary control on many components of the forest carbon cycle, such as live biomass, coarse woody debris biomass, forest floor biomass, biomass accumulation, and so forth (Bradford et al., 2008). Considering time since disturbance is therefore essential for quantifying and predicting a wide range of ecological functions, including carbon stocks and fluxes, which are highly dynamic following disturbances, presenting a significant challenge for carbon budge assessment (Williams et al., 2014).

A number of prior studies have sought to incorporate the time since last stand-clearing disturbance, analogous to forest stand age, as a determinant or predictor of carbon fluxes and stocks (Cohen et al., 1996; Chen et al., 2002; Chen et al., 2003; Law et al., 2004; Turner et al., 2004; Liu et al., 2011; Pan et al., 2011; Williams et al., 2012; Zhang et al., 2012; Williams et al., 2014). Data from the Forest Inventory and Analysis (FIA) offers one source for characterizing time since stand-replacing disturbance at broad scales (Williams et al., 2012; Williams et al., 2014). Stand ages were recorded by coring and dating of large trees in forest plots (FIA, 2015), because high severity, stand replacing events that level all canopy dominants and even understory individuals leave a clearer mark of disturbance timing (Schoennagel et al., 2004). Aboveground live biomass also reflects the time since disturbance in so far, as biomass exhibits a predictable rate of accumulation following stand replacement. The rate of biomass accumulation is influenced by a wide range of factors, such as climate, soil and site fertility, species composition, successional dynamics, the type of stand-replacing disturbance, and impacts from varying severity disturbance events (Johnson et al., 2000). The complex combination of how these factors are distributed across landscapes challenges generic characterization of time since disturbance, and most importantly post-disturbance forest carbon dynamics. Though forest inventory remains one of the only ways of quantifying time since disturbance, it is an imperfect surrogate for time since disturbance at broad scales, because for low severity, partial disturbance events, time since disturbance can be difficult to quantify in field surveys.

Large-area fine-scale assessments of forest carbon stocks and fluxes require spatially extensive and continuous characterization of time since disturbance across landscapes at a scale of being able to detect small-scale disturbance events, typically around 100 m or less. FIA stand age plot data can be used to create a continuous forest stand age map at resolutions of 250 m in conterminous US (Pan et al., 2011). However, such efforts are significantly limited by sparse plot coverage, poor representation of partial disturbances, and the fact that stand age, disturbance legacies, and carbon stocks and fluxes all vary widely at 250 m and coarser scales. Quantification of carbon stocks and fluxes based on coarse-scale stand age information could be biased (Zhang et al., 2012). Thus, field inventory imputed carbon stocks and fluxes can only provide rough guidance for forest carbon management, monitoring and verification at small scales (Wilson et al., 2013).

A number of remote sensing (RS) data and methods are available to quantify disturbance timing or disturbance timing related attributes. Time-series Landsat data and disturbance detection methods generate spatially extensive characterization

of contemporary disturbance events and magnitude, providing a direct estimate of time since disturbance (Cohen et al., 2002; Goward et al., 2008; Huang et al., 2010; Zhu et al., 2012). However, such disturbance products only record events that occurred within the last several decades, thus missing the long-lasting legacies from disturbances that occurred before the beginning of the relevant remote sensing observations. Fortunately, such long-lasting legacy effects are partially captured

from RS-derived stand ages or aboveground biomass (Cohen et al., 1995; Saatchi et al., 2011; Kellndorfer et al., 2013). Stand age maps directly determined from RS only offer rough age estimates binned into several classes over a large area, while aboveground biomass products estimated from optical, radar and lidar data can provide pixel-level biomass (Cohen et al., 1995; Saatchi et al., 2011; Kellndorfer et al., 2013). Such aboveground biomass products have the potential to support inferences about additional properties of a given forest stand such as stand age and disturbance legacy, particularly when

considered within a well-defined regional context of typical biomass accumulation rates for a local set of edaphic, climatic, and forest-type settings (Zhang et al., 2014). There may be considerable ambiguity and confusion arising from incomplete information, as well as the potential for a range of field conditions that yield similar aboveground biomass or forest structure. Nonetheless, RS-derived forest biomass still provides a valuable way of characterizing the pixel-scale (e.g. 30m or 250 m) legacy effects of disturbance that occurred prior to RS observations, which is required for quantifying carbon stock

recovery and carbon uptake and release rates over large areas. Combining disturbance products with other available disturbance layers is needed to distinguish among different disturbance types and severities to help assess carbon balance as a consequence of varying disturbance type and severity (Cohen et al., 2002).

Disturbance events are highly heterogeneous in space, with both occurrence and severity varying interactively with a wide range of site factors, resulting in highly diverse impacts on forest carbon dynamics (Turner, 2010). The effects of

disturbances on forest carbon in US forests were assessed and simulated with time since disturbance from FIA data by a number of growth-based or process-based modelling approaches (e.g. Williams et al., 2016). This generates regional carbon stock and flux trajectories with time since disturbance following fire, bark beetle and harvest with varying severity (Turner et al., 2004; Turner et al., 2007; Williams et al., 2012; Ghimire et al., 2012; Williams et al., 2014; Ghimire et al., 2015; Raymond et al., 2015). Previous studies that used model-derived trajectory curves to map carbon fluxes as a consequence of

disturbances mainly focused on only one of the disturbance types, but accounting for multiple disturbance types is necessary for more accurately mapping and reporting carbon dynamics (Williams et al., 2016).

This study estimates and maps time since disturbance at a fine scale of 30 m from RS-derived products and FIA-derived biomass growth curves, and then maps net ecosystem productivity (*NEP*) based on disturbance history, time since disturbance and carbon flux legacy. The specific objectives in this study are to: (1) introduce a method for inferring a pixel's

representative time since disturbance from RS-derived biomass and disturbance products at the 30 m resolution; (2) map *NEP* based on pre-existing, model-derived carbon stock and flux trajectories that describe how *NEP* changes with time following harvest, fire, or bark beetle disturbances of varying severity; (3) propagate uncertainties from RS-derived biomass products and FIA into uncertainty quantification of stand age and *NEP*. Our research represents an approach to map carbon

stocks and fluxes at a high resolution across the conterminous US in support of national carbon monitoring, reporting, and management.

## 2 Materials and Methods

### 2.1 Overview

Time since disturbance for each forest pixel was identified with one of the following two approaches depending on whether a recent disturbance was detected. The indicators of forest disturbances including disturbance type and year were determined from combination of three RS-derived disturbance products based on assumed rules. For those pixels that have been mapped as having a recent disturbance since the beginning of the relevant RS observations, time since disturbance was directly estimated by the difference between the target mapping year (year 2010 in this study) and the year of the last observed

disturbance For forest pixels that were not disturbed during the time span of the disturbance product, we inferred time since last stand-clearing disturbance, which is also called "stand age" (Masek and Collatz, 2006); terms "time since disturbance" and "stand age" are used interchangeably for recently undisturbed forest pixels thereafter. Stand age was inferred from RS-derived biomass data by finding the typical stand age that corresponds to each pixel's biomass according to field inventory-derived biomass-age curves, known as yield tables in forestry. The curves were sampled from FIA data and specific to forest

type group and site productivity class. Consequently, maps of forest type group and site productivity aid pixel-level determination of which biomass-age curve is to be used for each pixel.

*NEP* in 2010 across the PNW region was mapped based on carbon stock and flux trajectories derived from Carnegie-Ames-Stanford Approach (CASA) model in our prior work describing how *NEP* changes with time following harvest, fire, or bark beetle disturbances of varying severity (Ghimire et al., 2012, Williams et al. 2014, Ghimire et al., 2015). *NEP* curves with

time since disturbance vary by forest type and site productivity class, and are unique to post-harvest (Williams et al. 2012), post-fire (Ghimire et al., 2012) and post-bark beetle (Ghimire et al., 2015) disturbance types. *NEP* trajectories were applied to pixels with attributes of time since disturbance, forest type group, site productivity class, disturbance type, and disturbance severity to estimate carbon fluxes in forests caused by post-disturbance growth and decomposition locally and regionally.

**2.2 Inferring time since disturbance from remote sensing and inventory data**

### 2.2.1 Time since disturbance for disturbed forest pixels

North American Forest Dynamics (NAFD) disturbance products, Monitoring Trends in Burn Severity (MTBS) and Aerial Detection Surveys (ADS) polygons were used to determine whether and when forest pixels were disturbed during 1986 to 2010 (Table 1). NAFD products include 25 annual and two time-integrated forest disturbance maps with spatial resolution of

30 m for the conterminous United States (CONUS) (Goward et al., 2015). These maps were derived from annual time series

Landsat images from 1986 to 2010 using the Vegetation Change Tracker (VCT) algorithm (Huang et al., 2009; Huang et al., 2010). In this paper, we used one of the time-integrated data layers, which maps the year of the most recent forest disturbance between 1986 and 2010. The MTBS project maps annual burned area and burn severity at 30 m resolution across all lands of the United States from 1984 to 2014 (Eidenshink et al., 2007). Burned areas were determined by the differenced Normalized Burn Ratio (dNBR) index calculated across time-series Landsat images. MTBS defines burn severity classes based on distribution of dNBR values and ecological settings. We integrated the annual MTBS data from 1986 to 2010 into two images: (1) year of the most recent fire event and (2) burn severity corresponding to the recent fire, and applied a NAFD forest area mask to the integrated maps. The ADS program conducts annual surveys to investigate forest injury caused by insect outbreaks using aircraft observations since 1997, and generates polygons recording a number of attributes including disturbance year, areas and number of trees killed by insects per area. We selected polygons attacked by bark beetles from 1997 to 2010, converted the number of trees killed by bark beetles per area to biomass killed per area by multiplying county-level FIA-derived average aboveground biomass per tree for corresponding forest types, and then binned biomass killed per area into different bark beetle severity levels (Ghimire et al., 2015). Those polygons were rasterized into two images with a cell size of 30 m: (1) year of bark beetle occurrence and (2) the severity of bark beetle outbreak represented by the amount of live biomass killed, with a NAFD forest mask applied to these two images.

Preprocessed layers of NAFD (Fig. 1a), MTBS (Fig. 1b) and ADS (Fig. 1c) data characterized the year of most recent disturbance events. These three layers were integrated to create a single 30 m resolution image of disturbance type associated with the last disturbance between 1986 and 2010. Since the NAFD disturbances have not yet been fully attributed to disturbance type, and because some pixels are recorded as having experienced more than one disturbance type, we made four simplistic rules to define a single disturbance type to each pixel. These assumptions are based on the rationales: (1) MTBS records most of the notable fire events in the region, (2) harvest events are one of the most ubiquitous stand replacing disturbance types active in the region, (3) ADS-mapped polygons of bark beetle infestations often include unaffected stands as has been reported in the literature (Meddens et al., 2012; Vanderhoof et al., 2014). Our four rules were: (1) When NAFD and MTBS overlap, if the two events are within 3 years we assigned fire to the pixel, and if the two were separated by more than 3 years, we assigned whichever event type was most recent event with harvest for NAFD, and fire for MTBS. (2) When NAFD and ADS overlap, if the two events were separated by more than 3 years, harvest was assigned to the overlapping areas, but if they occurred within three years of each other, bark beetle outbreak was assigned. (3) When MTBS and ADS overlap, the overlapping areas were assigned fire. (4) Harvest was assigned to all remaining disturbed pixels identified by NAFD. The year of last disturbance for each disturbed pixel was then assigned based on the year of disturbance in each corresponding disturbance data product. Time since disturbance for disturbed pixels was then calculated as the difference between target mapping year of 2010 and the year of last disturbance.

### 2.2.2 Time since disturbance for recently undisturbed forest pixels

For the remaining forest pixels having no disturbance detected during 1986 to 2010, national biomass datasets were used to identify the corresponding stand age inferred from biomass-age curves that are specific to forest type groups and site productivity classes (Table 1). Mapped strata of forest type group and site productivity were used to determine the appropriate biomass-age curve to be used in referring stand age from biomass.

Biomass-age curves were derived from the FIA database, sampled to provide means and sampling errors for two attributes: aboveground dry weight of live trees and area of forest land. The ratio of these two attributes provides aboveground live wood biomass per area. We obtained the ratios and associated errors for the PNW region through the USDA Forest Service FIA EVALIDator online tool (http://apps.fs.fed.us/Evalidator/evalidator.jsp). This yielded biomass per area within strata of forest type groups (28 classes), stand age (11 classes) and site productivity (7 classes) (Table S1). We combined the original 7 site productivity classes into high and low productivity classes, defined by the rate of forest volume growth as 120 to 225+ cubic feet/acre/year and 20 to 119 cubic feet/acre/year respectively. Ratios and sampling errors were recalculated for each forest type group, age class and site productivity based on this grouping. Biomass-age curves were fitted following Williams et al. (2012) by parameterizing a wood production model that best matches the field inventory data. A Monte Carlo approach was used to incorporate uncertainty in the biomass per unit area with one hundred samples of the biomass at each age class drawn probabilistically. We then fitted corresponding one hundred curves for each forest type and productivity class, providing a distribution of biomass at each stand age from years 1 to 200.

Pixel-level biomass was obtained from the National Biomass and Carbon data set for the Year 2000 (NBCD 2000) (Fig. 2a). The 30 m resolution biomass map was developed based on empirical modeling combining FIA data, InSAR data from 2000 SRTM, and Landsat ETM+ optical remote sensing (Kellndorfer et al., 2012). Only biomass estimates for undisturbed pixels were used for inferring stand age. Differences in forest masks between NAFD disturbance and NBCD biomass products led to a number of pixels having a biomass recorded as zero. These were replaced by the mean biomass of nearby undisturbed pixels with the same forest type and site productivity within this region. The 250 m forest type group maps we used were created by USDA Forest Service, and were derived from MODIS composite images in combination with FIA data and nearly 100 other geospatial data layers, portraying 28 forest type groups across the contiguous United States (Ruefenacht et al., 2008) (Fig. 2b). Differences in map resolution between disturbance and forest type maps led forest type to be undefined for some pixels along forest edges, so we assigned the forest types of the nearest pixel. Site productivity maps were also derived from FIA data (Fig. 2c) with the following procedure. The FIA dataset was sampled to obtain the area of each county across the region that is of each forest type group and site productivity class. We then created a continuous map of county numbers on a 0.01 degree grid, overlayed forest types, and integrated those with the data on each county's area of high and low productivity classes for the forest type that was most abundant in the pixel. This yielded a map of productivity class fractions, where each pixel has a fraction high productivity (summed over classes 1 to 3 spanning 120 to 225+ cubic feet/acre/year) and fraction low productivity (summed over classes 4 to 6 spanning 20 to 119 cubic feet/acre/year). In reality,

site productivity is unlikely to vary across the 30 m pixel scale as much as it does at the county scale, whereas high and low site productivity fractions are likely to vary across counties in some cases. However, an improved characterization is not available at this time.

For each recently undisturbed forest pixel, we extracted its biomass ($B$), forest type ($T$) and fraction of high productivity ($f_{high}$), and then retrieved 100 biomass trajectories for forest type $T$ and for high and low productivity classes respectively. If the pixel was located at a high productivity site ($f_{high} = 1$), we treated 100 biomass curves for high productivity as 100 biomass realizations at stand ages from 0 to 200. All the biomass values among those realizations that lie within 20% of the pixel's observed $B$ were pooled, and corresponding stand ages were derived (Fig. 3a). We then calculated the mean, standard deviation and each of the 10*th* quantiles from the pooled stand ages (10*th*, 20*th*, 30*th*, ..., 80*th*, 90*th* quantiles of stand age) (Fig. 3b). The quantiles provided a frequency distribution of stand age for the individual pixel. Similarly, if the pixel was entirely of low site productivity ($f_{high} = 0$), we followed the above steps but using trajectories for low productivity class to derive stand age distribution for low productivity (Fig. 3c). In reality, $f_{high}$ is almost always between 0 and 1 (maximal $f_{high} =$ 0.996, minimal $f_{high} = 0.015$ in PNW). In order to reflect high/low productivity proportion of the total, we combined the two distributions above (one for high and the other for low productivity classes) by making copies of the two distributions with 10\*$f_{high}$ copies for the high productivity and 10\*$(1 - f_{high})$ copies for the low productivity. We calculated the mean, standard deviation and quantiles from the combined distribution of stand age (Fig. 3d). Since year 2010 was the target year for our mapping of stand age and carbon fluxes while biomass maps were generated for the year 2000, we simply added 10 years to the inferred ages to get adjusted stand ages. Using the above procedure across all undisturbed forest pixels, we generated maps of the mean and standard deviation of stand age.

Finally, we merged the stand age map for undisturbed forest pixels with the time since disturbance map for disturbed pixels to obtain a continuous map for all the forest pixels across the study area. To evaluate the derived map of time since disturbance, we made comparisons with two currently available products. First, density curves of stand age were plotted from maps derived from this study and Pan et al. (2011) for the study area. Another comparison was made between the distribution of forest area with age class from this study and that sampled from the FIA dataset.

## 2.3 Estimating *NEP* and uncertainties across the PNW region

### 2.3.1 Carbon flux trajectories for harvest, fire and bark beetle

Carbon flux trajectories for post-harvest, -fire and -bark beetle outbreaks were derived from our prior work (Ghimire et al., 2012, Williams et al., 2014, Ghimire et al., 2015) involving an inventory-constrained version of the CASA carbon cycle process model with inclusion of disturbance processes. The CASA model used here is based on Randerson et al. (1996) and operates on a monthly time step. It uses a light use efficiency approach to simulating net primary productivity (*NPP*) based on RS-derived absorption of photosynthetically active radiation, biome parameters, and climate data. The model then allocates *NPP* to three live carbon pools (leaves, roots, and wood), and transfers carbon to dead pools (litter and soils) based

on biome-specific rates of tissue turnover. Carbon in dead organic matter pools is transferred between pools (surface structural C, surface metabolic C, soil structural C, soil metabolic C, aboveground coarse woody debris C, belowground coarse woody debris C, surface microbial C, soil microbial C, slow C and passive C) (Fig. S1). Amounts of carbon transferred to microbial pools and carbon emitted to the atmosphere from microbial decomposition of soil and litter, i.e. heterotrophic respiration ($Rh$), depend on the rate and efficiency of heterotrophic consumption which varies between pools in the model, and also depends on biome- and pool-specific chemistry and site-specific climate setting such as soil moisture and temperature. The difference between $NPP$ and summed $Rh$ of the ten detrital pools is then calculated as $NEP$. Aboveground biomass-age curves sampled from FIA database were used as a constraint to adjust wood turnover rate (mortality and shedding) and default output $NPP$ by a scalar parameter, which hence influences accumulation of live biomass and the amounts of carbon allocated to live and deal pools. The adjustment of those CASA parameters was performed for each specific combination of forest type group and site productivity class. Implementation of the model involved a simulation sequence beginning with spin-up to equilibrium carbon pools using FIA-adjusted $NPP$ followed by a pre-disturbance with ensuing regrowth to a set of pre-disturbance ages, and lastly imposition of the disturbance of interest to generate flux and stock dynamics in response to harvest, fire, or beetle outbreak disturbance types at different severities. The fate of carbon influenced by disturbances varied by disturbance type as described below. Disturbance types considered in the model included stand-replacing harvest, fire, and bark beetle outbreak, with full descriptions of the carbon dynamics in pools after disturbances provided in Ghimire et al., 2012, Williams et al., 2014, Ghimire et al., 2015 and described further below.

Model treatments of disturbance impacts to the carbon cycle were as follows. For all disturbance type cases, a post-disturbance decline and ensuing recovery of NPP and fractional allocation to wood were modelled as a negative exponential function of time since disturbance, recovering to the pre-disturbance level within eight years (Williams et al. 2012). The mortality and fate of disturbance killed and/or combusted carbon pools differed by disturbance type. For harvest (Williams et al. 2014), the post-disturbance biomass was set to 50% of the aboveground live wood biomass reported in FIA data for the 0 to 20 year old age class, regardless of the pre-disturbance biomass condition. Harvest-killed live wood, leaves, and roots were calculated from their corresponding fractions of pre-disturbance to post-disturbance conditions. Eighty percent of the disturbance-killed aboveground live wood was assumed to have been removed from the site with the remainder being treated as slash subject to decomposition as coarse woody debris. All leaves of disturbance killed trees are assumed to decompose on-site. All belowground wood that succumbs to mortality enters a belowground coarse woody debris carbon pool. For fire (Ghimire et al. 2012), disturbance kill is portrayed as partial mortality events in which fires reduce pre-fire live biomass pools based on the fractional tree mortality which varies by forest type and fire severity class (high, medium, or low to match the MTBS dataset). The amount of live biomass remaining after a fire is calculated from the fraction of vegetation mortality emerging from an extensive literature survey. Fire-killed material is either directly combusted and released to the atmosphere or transferred to dead carbon pools. The same applies to foliage and root mortality, though roots are not directly combusted. Fire killed trees enter a new standing dead pool in the model with a fast turnover fall rate of 10 years post fire with transfer to the coarse woody debris pool. Litter and soil organic carbon in the upper soil layers were also vulnerable to

combustion. All of these rates were based on literature review (Ghimire et al. 2012). For bark beetle outbreaks (Ghimire et al. 2015), beetle killed biomass was simulated for a wide range of intensities from near zero to 100% mortality to generate a family of curves that could be applied in the mapping stage. Beetle attack caused leaf carbon to enter the surface litter pool, aboveground wood to enter a snag pool, belowground wood and fine root to enter corresponding soil carbon pools. The portions killed for each were based on the percent mortality imposed for each severity level being simulated, derived from the ratio of the biomass at the midpoint of each severity class (of 1680 levels of intensity) to the pre-disturbed aboveground biomass values corresponding to the mean age of a given forest type under attack (Ghimire et al. 2015). Soil carbon pools respond to all of these dynamics according to the model's climate-mediated turnover times for each carbon pool, and the associated carbon flows (Fig. S1).

For a given forest type, site productivity, and prior disturbance, this forest disturbance version of the CASA model simulates *NPP, Rh* and *NEP* as a function of time since disturbance. A family of curves describing carbon stocks in carbon pools (soil organic carbon, litter, slow turnover soil carbon, aboveground coarse woody debris, belowground coarse woody debris, and total live woody biomass), *NPP, Rh* and *NEP* with time since disturbance for each combination were created to represent uncertainties in the amount of biomass killed and left on site after a disturbance, the amount of biomass left live on site post-disturbance, and the rate of biomass accumulation and mortality (Ghimire et al., 2012, Williams et al., 2014, Ghimire et al., 2015). This study emphasized the use of *NEP* curves from our prior works combined with time since disturbance derived from this study to map spatially explicit *NEP* in the PNW region. Fig. 4 provides examples of post-disturbance *NEP* trajectories from our prior work, showing average of 20 simulations of post-harvest *NEP* (Fig. 4a, 4b), the average of 25 simulations of post-fire *NEP* for three different fire severities (Fig. 4c, 4d), and 1 simulation of post-bark beetle *NEP* for three examples of bark beetle disturbances that kill low, medium and high amounts of biomass (Fig. 4e, 4f) in high site productivity Douglas-fir stands in the PNW region. The typical overall pattern of *NEP* following a disturbance involves a large negative value immediately after disturbance, a rise for a number of years to reach a maximum rate of carbon uptake, and then a gradual decline to a steady state. The post-disturbance *NEP* curves across all forest types, productivity classes, and disturbance types are presented in the supplementary figures (Fig. S2, S3, S4, S5, S6), *Rh* trajectories are shown in Fig. S7, S8, S9, S10, S11, and characteristic trajectories of post-disturbance carbon stocks in carbon pools are provided as well (Fig. S12, S13, S14, S15, S16, S17 for harvest as an example).

### 2.3.2 Mapping *NEP* and uncertainties across the PNW region

The characteristic trajectories serve as look-up tables relating carbon fluxes and stocks (here just *NEP*) to years since disturbance within the strata of forest type group, site productivity fraction, disturbance type and severity. For disturbed pixels, the distribution of *NEP* corresponding to the pixel's time since disturbance and forest type was sampled for both high and low productivity classes, and then weighted according to the pixel's fraction of high site productivity ($f_{high}$). Weighting involved a simple repetition of each data population based on the pixel's fraction of high productivity, with $10*f_{high}$ copies for the high productivity estimates and $10*(1 - f_{high})$ copies for the low productivity population. These two populations were

then combined to create a single composite distribution representing the full probability distribution for the pixel's *NEP*. A similar procedure was performed for all remaining undisturbed forest pixels but including the additional uncertainty on the pixel's stand age. We propagated stand age uncertainty by obtaining the *NEP* distribution for each of the 10*th* quantiles of the age distribution corresponding to the pixel's biomass and forest type for both high and low productivity classes, and compositing these into a full probability distribution of the pixel's *NEP* based on the pixel's fraction of high probability ($f_{high}$). Finally, we calculated the mean, standard deviation and quantiles (10*th*, 20*th*, 30*th*, ..., 80*th*, 90*th* quantiles) of *NEP* distribution for each forest pixel across the PNW region.

## 3 Results

### 3.1 Disturbance maps derived from NAFD, MTBS and ADS

Across the $2.1*10^7$ ha of forest in the PNW region, harvest was recorded as having affected the largest area ($5.4*10^6$ ha from 1986-2010) followed by bark beetles ($1.8*10^6$ ha from 1997 - 2010) and then fire ($9.3*10^5$ ha from 1986 - 2010). Their distributions are displayed in Fig. 5. Reported as percentages, harvest, bark beetles, and fire affected 26%, 9%, and 5% of all forestland in the PNW during their respective time intervals. Table 2 provides an additional report of each area by forest type and for high and low productivity class sites. Douglas-fir comprises nearly 50% of all forest in the PNW, with about 70% of it being in high productivity class lands. Ponderosa Pine, Fir-Spruce-Mountain Hemlock, and Hemlock-Sitka Spruce are the next most abundant forest type groups, comprising 17%, 15%, and 7% of the PNW forest, with 16%, 39%, and 85% in high productivity sites, respectively.

About half (52%) of all harvesting occurred in Douglas-fir forests, with 20% in Ponderosa Pine stands and 8% and 7% in Fir-Spruce-Mountain Hemlock, and Hemlock-Sitka Spruce stands. Of all forestland that burned, 37% was in Douglas-fir stands, 27% in Ponderosa Pine, and 21% in Fir-Spruce-Mountain Hemlock. Hemlock-Sitka Spruce was not vulnerable to fire. Though fire affected a larger area of low productivity sites for Ponderosa Pine and Fir-Spruce-Mountain Hemlock forest types, fire occurrence was equally likely across low and high productivity classes. In contrast, Douglas-fir stands had similar burned areas for low and high productivity sites, but low productivity sites were three times as likely to experience fire. Bark beetle outbreaks were most common in Douglas-fir stands, with 40% of all outbreak area, while 30% and 18% occurred in Fir-Spruce-Mountain Hemlock and Ponderosa Pine stands, respectively. As with fire, though a larger proportion of the total bark beetle outbreak area occurred in low productivity Ponderosa Pine and Fir-Spruce-Mountain Hemlock stands, their occurrence was equally likely across low and high productivity sites. Again in contrast, bark beetle outbreak areas for low and high productivity classes are similar in Douglas-fir forests, but beetle outbreak occurrence was about three times more likely in low productivity sites. Of all Douglas-fir stands, 28% were disturbed by harvest, 3% by fire, and 7% by bark beetles. Percentages for Ponderosa Pine stands were 31%, 7%, and 9% for harvest, fire, and bark beetles, and for Fir-Spruce-Mountain Hemlock they were 13%, 6%, and 17%, Hemlock-Sitka Spruce was mainly disturbed by harvesting (28%), with 0% for fire and 5% for bark beetles.

## 3.2 Biomass-age curves by forest types and site productivity classes

The fitted biomass regrowth curves exhibit considerable variations across forest types and site productivity classes (Fig. 6 for Douglas-fir, Ponderosa Pine and Fir-Spruce-Mountain Hemlock). Compared to Douglas-fir forests, Ponderosa Pine forests hold only about 28% to 33% as much biomass, and Fir-Spruce-Mountain Hemlock holds about 59% to 64% as much.

Biomass accumulates more rapidly and to a higher maximum stock for high productivity sites for all forest types according to FIA data and corresponding model fits, achieving about 1.4 to 1.8 times the biomass at low productivity sites. But the biomass-age curves share some common features among different forest types and site productivity classes. Biomass accumulates rapidly at the early ages (~ 0-50 years), slowing down with age until it saturates often around 150-200 years. Besides, variation in biomass increases as a function of stand age both in the FIA data and in the model fits. The fitted

curves provided a range and distribution of biomass at each stand age from 0 to 200. Because of the simple stand-level growth equation that was assumed, these curves yielded a smoothed fit to the inventory data rather than the erratic and fluctuating jumps imposed on the general increase with stand age seen in the field data.

## 3.3 Maps of time since disturbance and uncertainties across the PNW region

The forested landscape is a complicated mosaic of time since last disturbance (Fig. 7a). Overall, a wide range of years are

15 spanned with abrupt discontinuities related to recent stand replacing disturbances, transitions between forest types, and transitions between site productivity classes. One feature that stands out prominently is the prevalence of recent disturbances along the eastern, drier side of the Cascade Range, resulting from both harvesting and bark beetle outbreaks (Fig. 5b). Large fires produce sizable patches with the same time since disturbance. The imprint of segments of relatively old, high-elevation forests is also evident. It should be noted that this map was not used directly in the computation of *NEP* for undisturbed

forest pixels, which relied instead on stand age distributions for high and low site productivity classes, but it is presented here to provide a best estimate of disturbance timing at the pixel scale.

Uncertainty on the time since disturbance for disturbed forest pixels is not currently available from disturbance products and thus was not mapped. For undisturbed forest pixels, the uncertainty of stand age was represented by standard deviation of the full stand age distribution combined from high and low site productivity and reflecting high/low productivity proportion.

The uncertainty map identifies locations where stand age is more tightly constrained by the data and method (Fig. 7b). Across all the stand ages inferred from the biomass data (undisturbed forest pixels), the spatially-averaged mean standard deviation of stand age is around 25 years.

Density curves of stand age were compared between maps derived from this study and from Pan et al. (2011) for the study area. In undisturbed areas, spatial pattern and density distribution of stand age between the two studies are mostly consistent

(Fig. 9a, 9b), but this study has a much higher density at the age class of 0-10 years and a bit lower density at 50-100 years (Fig. 9c). Besides, the distribution of forest area with age class from this study was compared with that sampled from the FIA dataset (Fig. 10 for Douglas-fir and Fir-Spruce-Mountain Hemlock). We provided two age distributions from this study,

one sampled from only undisturbed pixels and another including all forested pixels. Overall, the pattern of FIA-derived age distribution matches well with that derived from our study, but with our study having consistently lower forest areas at age classes larger than 20. This is true except for in the youngest age classes when we include the pixels marked as disturbed in this study, finding a much larger frequency of young-aged forests.

**3.4 Maps of NEP and uncertainties across the PNW region in Year 2010**

Spatial variations in mean annual *NEP* are determined by differences in strata of time since disturbance, forest type group, and site productivity used in the mapping procedure (Fig. 8a). There is a general pattern of weaker carbon sinks in the eastern portion of the study area. Both sink strength and carbon source strength tend to be largest in the western areas of higher biomass. Recent (< 20 years) fire and harvest disturbances tend to create focused carbon sources on the landscape, giving way to sinks as regrowth ensues. For example, one can see a clear imprint of the well-known 2002 Biscuit fire in southwestern Oregon (bottom left of Fig. 8a, also refer to bottom left of Fig. 5a & 5b). Area with very recent, but low severity bark beetle outbreaks have an only muted reduction in *NEP* compared to nearby undisturbed forest, remaining carbon sinks despite the disturbance episode.

At the regional scale, *NEP* is estimated to be 5.9 Tg C $y^{-1}$, or about 28.5 g C $m^{-2}$ $y^{-1}$ averaged for the $2.1*10^7$ ha forest (Table 3). Recently undisturbed forests are the region's main terrestrial carbon sink with *NEP* of $13.6 \pm 3.7$ Tg C $y^{-1}$. In contrast, *NEP* for forests disturbed by harvest, fire and bark beetles within the prior two and a half decades are estimated to be $-6.4 \pm 2.3$ Tg C $y^{-1}$, $-0.5 \pm 0.2$ Tg C $y^{-1}$, and $-0.8 \pm 0.0$ Tg C $y^{-1}$ respectively, serving as significant carbon sources. Table 3 also reports mean *NEP* by forest type groups for all forestland, and also separately for undisturbed and disturbed forests. Fir-Spruce-Mountain Hemlock followed by Douglas-fir and Hemlock-Sitka Spruce were the largest carbon sinks of 1.6 Tg C $y^{-1}$, 1.5 Tg C $y^{-1}$, and 1.5 Tg C $y^{-1}$ respectively. Considering only undisturbed forestlands, Douglas-fir was the largest carbon sink of $7.9 \pm 2.1$ Tg C $y^{-1}$, but this was mostly offset by it having also the region's largest carbon sources from harvest, fire events and bark beetle outbreaks with *NEP* of $-5.7 \pm 1.6$ Tg C $y^{-1}$, $-0.4 \pm 0.1$ Tg C $y^{-1}$, and $-0.2 \pm 0.0$ Tg C $y^{-1}$ respectively. Douglas-fir's relatively large area-integrated carbon fluxes result not only from it being the most abundant forest type in the PNW region, but also its large disturbed areas and large carbon stock potential. Recently disturbed forests tend to aggregate to carbon sources. In some forest type groups we found a net carbon sink even for recently disturbed forests. For example, Lodgepole Pine had net carbon sinks for harvested and burned stands. This results from a large proportion of disturbance events having occurred early in the disturbance record allowing recovery and regrowth to overwhelm the carbon sources from the most recent events.

**4 Discussion**

**4.1 Assumptions of method for mapping time since disturbance and *NEP***

Our method of inferring time since disturbance to estimate carbon flux and biomass accumulation relies on a number of data products and assumptions that need to be critically evaluated. First, the method assumes that field inventory data provide a reliable and well-constrained estimation of forest biomass as a function of stand age for regionally-specific strata of forest type and site productivity class. However, both stand age and biomass are difficult to measure and estimate, especially considering the difficulty of assigning a stand age to uneven-aged forest stands, as well as selecting appropriate species-specific biomass equations (Parresol, 1999). If FIA ages are older than actual stand ages, the associated forest biomass will be underestimated, and stand age inferred from biomass products will be overestimated. Likewise, younger FIA ages than actual ages will result in an overestimation in biomass accumulation, but an underestimation in biomass-inferred stand ages. Though a possible bias in stand ages, our estimates of carbon stocks and fluxes are not likely to be largely adjusted by a stand age bias within 5 years (Williams et al., 2012).

Second, we assume RS-derived NBCD biomass products were well calibrated by field-derived biomass. However, the correlation coefficients between observed and predicted biomass were estimated to be 0.62-0.75 in the PNW region (Kellndorfer et al., 2012). And at 30 m pixel level, NBCD biomass values were biased with a large number of zero biomass values that had predictions in local biomass products (Huang et al., 2015). Discrepancies in biomass values between RS- and field-derived data lead to biased estimates in stand age and associated carbon stocks and fluxes. These were addressed in this study by imposing 20% error to pixel level biomass estimates and replacing zero biomass by the mean biomass of neighboring forest pixels with the same forest type and site productivity.

Third, the approach described here assumes that stand-level biomass is a useful predictor of stand age, biomass accumulation and net carbon flux regardless of how that stand-level biomass was actually achieved (Zhang et al., 2014). However, a particular stand-level biomass may be reached from steady accumulation during a relatively disturbance-free interval of time, or from a decline of biomass accumulation after the biomass reaches the maximum, or also from a previous disturbance that reduced biomass and then accumulated to the current level (Xu et al., 2012). Information is also lacking on how the biomass-age relationship varies depending on the type of stand-replacing disturbance. Such path dependency can have important implications for the true stand age as well as for post-disturbance carbon fluxes and stocks by influencing species composition, stand structure, site fertility, and other relevant factors (Williams et al., 2012).

Next, the carbon cycle model used to estimate carbon fluxes as a function of time since disturbance relies on a simple growth rate equation to characterize biomass accumulation over time with a constant wood turnover time regardless of stand age and a constant rate of carbon allocation to wood. It also assumes that mean annual net primary productivity is constant after an initial rise through stand initiation (assumed 8 years of initialization in the model). These assumptions arise from limited data to describe these dynamics for the range of settings active at a continental-scale but improvement may be possible with

detailed explorations into regional parameterizations. Our prior work indicated some sensitivity of carbon flux estimation to these assumptions, though the impact on continental-scale carbon flux estimation was modest (Williams et al., 2012).

Finally, the method relies on maps of aboveground biomass, forest type group, site productivity class and forest disturbance which are sure to have errors. Accuracies of biomass and forest type group maps were assessed and provided by the data

provider, while the rest of them were not. Accuracy of forest type group map in the PNW region ranges from 61% to 69% (Ruefenacht et al., 2008); besides, forest type groups for some pixels undefined from original data were assigned as the forest types of the nearest pixels. For the same biomass value, inferred stand age and estimated carbon fluxes can vary greatly given difference in forest type group (Fig. 4 & Fig. 6); however, forest type group induced biases in *NEP* were not accounted due to the lack of information on associated errors from the spatial assignment of forest type group. The ADS

dataset is known to be limited by the areas flown in the survey years, and likely underestimate the number of trees killed by bark beetles but likely overestimate the area of affected stands (Meddens et al., 2012). Uncertainties from ADS dataset have important consequences for the carbon balance and flux estimates from bark beetle outbreaks, part of them were accounted for by Ghimire et al. (2015). Incorporation of local high-resolution high-accuracy maps for these strata into national maps can significantly reduce uncertainties in our mapping and interpretation of stand age, carbon accumulation and fluxes at fine

scales (Huang et al., 2015).

Our analyses have sought to incorporate three main sources of uncertainties in input data layers to estimate mean annual *NEP* for a given pixel. The first is uncertainty in the biomass defined at a pixel scale. The second source of uncertainty comes from a range of potential stand ages that could correspond to a given biomass stock. The third source of uncertainty comes from the *NEP* that we estimate for a given stand age, forest type, site productivity, and prior disturbance type and

severity. The first and second uncertainties were propagated to provide a probabilistic, statistical estimation of stand age. The full range distribution of stand age and the third uncertainty were further propagated by probabilistic sampling to obtain an *NEP* distribution for each forest pixel. The potential biases in *NEP* due to those uncertainties were reflected by standard deviation map of *NEP* in Fig. 8b. Though pixel-level accuracies are correspondingly low for many situations, aggregation to larger scales involves spatial cancellation such that regional and continental uncertainties are much reduced relative to what

would be inferred directly from the pixel scale.

## 4.2 Comparing maps of time since disturbance to other studies

We identified disagreements of density curves derived from time since disturbance map in this study and stand age map in Pan et al. (2011) at the age class of 0-10 years and 50-100 years (Fig. 9c). There are a number of likely explanations for these discrepancies. The first is definitional, in this study we estimated time since disturbance including both partial and

stand-replacing disturbances, resulting in assigning a young age to an old-growth forest stand undergoing a light-severity partial disturbance; while Pan et al. (2011) mapped stand age with consideration of only stand-clearing disturbance. The second cause could be related to the years included in each study, with a large percent of forestlands disturbed by harvest, fire or bark beetles between 2000 (mapping year in Pan's study) and 2010 (mapping year in this study) (Fig. 5). Other

factors that may contribute to this discrepancy include different datasets and methodology used for analysis and mapping, and different spatial resolutions between the maps. For example, when mapping at a much coarser resolution (250 m or 1 km), fragmented disturbed forest patches are likely lost due to disturbed areas taking up a small fraction in the coarse-scale pixel, yielding stand age for those areas represented by nearby undisturbed forest stands that are more abundant in that pixel.

The definition bias described above also applies to explain partially inconsistent distribution of forest area with age class from this study from the FIA dataset (Fig. 10). In this study, we included partial, low severity disturbances as stand with ages ranging from 0 to 24, but which are described as undisturbed forests of older stand age from 20-40 years up to 200+ years in FIA dataset. FIA data miss some recent disturbances, partly because FIA remeasurement cycle in the PNW region is about 10 years, with the average time lag of the data being around 5 years. We note that this definitional issue does bias estimates of *NEP* or biomass, which are derived based on severity-specific carbon stock and flux trajectories.

The map of time since disturbance from this study having a spatial resolution of 30 m is able to distinguish finer differences in the stand age structure for persistent forests, but also able to capture abrupt discontinuities related to recent stand replacing disturbances, transitions between forest types, and transitions between site productivity class abundances. This fine spatial detail of the data indicates the information that is lost when stand age is spatially averaged to coarser grids. Such spatial averaging of stand age becomes even more problematic when combined with the nonlinear relationships between forest properties and age, such as with biomass and *NEP*. Maps of time since disturbance and uncertainties from this study may be valuable in and of itself for various ecological applications even if our purpose was generate it as an intermediate variable needed en route to accurate description of, and interpretation of, carbon stocks and fluxes.

**4.3 Comparing maps of *NEP* to other studies**

The PNW-wide forest *NEP* reported here (6 Tg C y$^{-1}$) is lower than in our earlier work (11 Tg C y$^{-1}$) that used similar methods (see RS-based results in Williams et al. 2014). A portion of this difference can be attributed to larger net carbon losses from forestlands (-7 Tg y$^{-1}$ carbon loss in this study vs. -4 Tg y$^{-1}$ carbon loss in Williams et al.) due to recent (1986 to 2010) disturbance by either harvest or fire. Here we also include additional net carbon losses from bark beetle outbreaks (-0.8 Tg C y$^{-1}$) that were not considered in our earlier work. The remaining discrepancy (-1.2 Tg C y$^{-1}$) is necessarily due to other methodological and data source innovations introduced here including: (1) use of the newly available Landsat-derived forest disturbance product that now offers full spatial coverage compared to only about 50% coverage previously, and (2) new use of biomass data to characterize stand age and associated carbon flux patterns.

The net carbon release from recent bark beetle outbreaks (-0.8 Tg C y$^{-1}$ in 2010) is comparable to that reported in our earlier work (Ghimire et al. 2015). In our earlier work we reported that the PNW region contributed about 28% to the net carbon release in western US regions. Applying that percentage to the US west-wide *NEP* reduction of 1.8 to 3.6 Tg C y$^{-1}$ (Ghimire et al. 2015) indicates a *NEP* reduction for just the PNW of about 0.5 to 1.0 Tg C y$^{-1}$, within the range of the net carbon release induced by bark beetles reported here.

Additional points of comparison come from a variety of papers focused on regions of Oregon by Turner et al. (2004, 2007 & 2015). These studies report similar west versus east patterns of *NEP* across the mountain ranges of the region, and similar variation in *NEP* across forest types. However, the work of Turner et al. (2004, 2007 & 2015) tends to estimate higher *NEP* in regenerating forests (e.g. 14 to 99 years since stand clearing) in the Coast Range and West Cascades, reaching 250 to 390 g C m$^{-2}$ y$^{-1}$ whereas our curves peak at around 245 g C m$^{-2}$ y$^{-1}$ for the full PNW region (Fig. S1). This discrepancy could be due to the greater spatial detail on climate patterns included in their modelling work, and also plant productivity, allocation, and turnover rates prescribed at the ecoregion-scale in the work of Turner et al. (2004, 2007 & 2015). Given our work's aim of estimating forest carbon stocks and fluxes across the full conterminous US, it is not currently feasible to assemble the data needed to perform such fine-scale ecoregional calibration even while appreciating its value.

**5 Conclusions**

In this paper, we introduced a new methodology for comprehensively combining RS-based 30 m resolution data on disturbance year, disturbance type, and aboveground biomass with forest inventory data to quantify time since disturbance and associated carbon uptake and release across forested landscapes at a fine scale (30 m). Time since disturbance was an important intermediate variable that aided the assessment of disturbance-driven carbon emissions and removals legacies. We mapped mean, standard deviation and statistical distribution of stand age and *NEP* that were propagated from uncertainties of input data layers by probabilistic sampling. This method was applied to the Pacific Northwest (PNW) region of the US. Region-wide we found a net ecosystem productivity of 5.9 Tg C y$^{-1}$ for forestlands circa 2010, with net uptake in undisturbed forests during 1986-2010 (13.6 Tg C y$^{-1}$) overwhelming net negative *NEP* from tracts that have seen recent harvest (-6.4 Tg C y$^{-1}$), fires (-0.5 Tg C y$^{-1}$), and bark beetle outbreaks (-0.8 Tg C y$^{-1}$). Our proposed approach will be further applied to forestlands in other regions of the conterminous US to advance a more comprehensive monitoring, mapping and reporting the carbon consequences of forest change across the US.

**Acknowledgements**

This study was financially supported by NASA's Carbon Monitoring System program (NNH14ZDA001N-CMS) under award NNX14AR39G. Sincere thanks to Yu Zhou for the suggestions on CASA modelling. Thank you to associate editor and two anonymous reviewers for their helpful comments to greatly improve our paper.

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

**Tables**

**Table 1.** Data sources for inferring time since disturbance for recently disturbed and undisturbed forest pixels.

| Data | Description | Source | Year | Input for recently disturbed or/and undisturbed forests |
|------|-------------|--------|------|-----------------------------------------------|
| NAFD | Forest disturbance | Landsat | 1986-2010 | a, b |
| MTBS | Burned area and severity | Landsat | 1986-2010 | a |
| ADS | Area of insect outbreak and number of trees killed | Aerial survey | 1997-2010 | a |
| NBCD | Aboveground live biomass | Landsat, SRTM, FIA | 2000 | b |
| Forest Type Group | Forest type group | MODIS, NLCD, etc. | 2001 | b |
| Site Productivity | Fraction of high productivity | FIA | 1984-2014 | b |
| Biomass-age Curves | Biomass accumulation as a function of stand age | FIA | 1984-2010 | b |

[a] Data is one of the inputs for inferring time since disturbance for recently disturbed forest pixels.

[b] Data is one of the inputs for inferring time since disturbance for recently undisturbed forest pixels.

**Table 2.** Area (ha) of all forest lands, forests disturbed by harvest (1986-2010), fire (1986-2010) and bark beetle (1997-2010) by forest type groups, and for high and low site productivity classes in the PNW region.

| Forest Type Group* | All Forest | | Harvested | | Burned | | Bark Beetle Infested | |
|---|---|---|---|---|---|---|---|---|
| | High | Low | High | Low | High | Low | High | Low |
| Douglas-fir | 6909151 | 3097083 | 2039661 | 752902 | 161221 | 181147 | 301234 | 416181 |
| Ponderosa Pine | 565633 | 2953701 | 188300 | 888668 | 39135 | 213925 | 53982 | 261530 |
| Fir/Spruce/Mountain Hemlock | 1220916 | 1914562 | 155898 | 250135 | 61140 | 135947 | 193838 | 343326 |
| Hemlock/Sitka Spruce | 1168836 | 211376 | 338263 | 45762 | 255 | 226 | 54352 | 19382 |
| Pinyon/Juniper | 79800 | 664050 | 10561 | 84517 | 2988 | 27601 | 250 | 1378 |
| Alder/Maple | 633369 | 43005 | 278797 | 18128 | 325 | 24 | 3442 | 290 |
| Lodgepole Pine | 135874 | 441717 | 38737 | 167320 | 13064 | 42235 | 22692 | 78753 |
| Western Oak | 52774 | 97472 | 15572 | 25392 | 4371 | 7817 | 1036 | 2026 |
| California Mixed Conifer | 16841 | 73817 | 5017 | 18369 | 550 | 1669 | 836 | 1376 |
| Tanoak/Laurel | 50897 | 26536 | 11291 | 5351 | 6431 | 3509 | 265 | 160 |
| Other Western Hardwoods | 39696 | 33718 | 10542 | 7342 | 535 | 844 | 912 | 777 |
| Elm/Ash/Cottonwood | 34093 | 17945 | 14779 | 7812 | 331 | 284 | 60 | 41 |
| Western Larch | 20464 | 28342 | 3126 | 4611 | 744 | 1436 | 5078 | 6597 |
| Other Western Softwood | 9956 | 24206 | 1441 | 2587 | 1711 | 6037 | 1726 | 5259 |
| Western White Pine | 7877 | 4471 | 204 | 183 | 7360 | 3951 | 35 | 37 |
| Aspen/Birch | 1908 | 2607 | 730 | 894 | 176 | 286 | 124 | 353 |

*Forest type groups are ordered by the forest areas from largest to smallest.

1 **Table 3.** Mean net ecosystem productivity (*NEP*) and total net carbon uptake by forest type group in all forests, recently undisturbed forests,

2 forests disturbed by harvest, fire and bark beetle occurred during time spam of remote sensing disturbance products.

| Forest Type Group* | All Forests | | Recently Undisturbed | | Harvested | | Burned | | Bark Beetle Infested | |
|---|---|---|---|---|---|---|---|---|---|---|
| | Mean NEP | Total NEP | Mean NEP | Total NEP | Mean NEP | Total NEP | Mean NEP | Total NEP | Mean NEP | Total NEP |
| | $(g\ C\ m^{-2}\ y^{-1})$ | $(Gg\ C\ y^{-1})$ | $(g\ C\ m^{-2}\ y^{-1})$ | $(Gg\ C\ y^{-1})$ | $(g\ C\ m^{-2}\ y^{-1})$ | $(Gg\ C\ y^{-1})$ | $(g\ C\ m^{-2}\ y^{-1})$ | $(Gg\ C\ y^{-1})$ | $(g\ C\ m^{-2}\ y^{-1})$ | $(Gg\ C\ y^{-1})$ |
| Douglas-fir | 14.9 | 1489.2 | 128.6 | 7912.6 | -205.7 | -5745.7 | -130.0 | -445.0 | -32.9 | -235.0 |
| Ponderosa Pine | 16.9 | 594.2 | 47.4 | 887.9 | -14.8 | -159.2 | -3.0 | -7.7 | -40.4 | -127.1 |
| Fir/Spruce/Mountain Hemlock | 52.6 | 1649.7 | 108.3 | 2161.6 | -22.0 | -89.5 | -35.0 | -69.0 | -66.2 | -354.7 |
| Hemlock/Sitka Spruce | 107.5 | 1483.5 | 197.0 | 1816.6 | -76.7 | -294.6 | -106.3 | -0.5 | -52.1 | -38.3 |
| Pinyon/Juniper | 5.2 | 39.0 | 9.4 | 58.2 | -18.7 | -17.8 | -1.9 | -0.6 | -48.7 | -0.8 |
| Alder/Maple | 58.4 | 395.3 | 118.0 | 442.8 | -16.7 | -49.7 | 97.6 | 0.3 | | |
| Lodgepole Pine | 9.6 | 55.7 | 36.6 | 78.6 | 4.2 | 8.6 | 28.2 | 15.6 | -46.4 | -47.0 |
| Western Oak | 9.5 | 14.2 | 28.4 | 26.7 | -33.9 | -13.9 | 5.8 | 0.7 | | |
| California Mixed Conifer | 14.4 | 13.1 | 27.0 | 17.0 | -19.5 | -4.6 | 70.3 | 1.6 | -51.4 | -1.0 |
| Tanoak/Laurel | 79.0 | 61.2 | 161.8 | 81.6 | -82.3 | -13.7 | -73.4 | -7.3 | | |
| Other Western Hardwoods | 43.7 | 32.1 | 62.0 | 32.5 | -12.4 | -2.2 | 59.5 | 0.8 | | |
| Elm/Ash/Cottonwood | 29.9 | 15.6 | 119.4 | 34.3 | -81.6 | -18.4 | -61.6 | -0.4 | | |
| Western Larch | 35.9 | 17.5 | 97.2 | 26.5 | -34.7 | -2.7 | -21.1 | -0.5 | -50.0 | -5.8 |
| Other Western Softwood | -12.9 | -4.4 | 26.4 | 4.1 | -21.8 | -0.9 | -39.9 | -3.1 | -64.5 | -4.5 |
| Western White Pine | 35.9 | 4.4 | 4.8 | 0.0 | -14.6 | -0.1 | 39.8 | 4.5 | -53.3 | 0.0 |
| Aspen/Birch | 20.3 | 0.9 | 51.6 | 1.0 | -14.8 | -0.2 | -9.8 | 0.0 | | |
| Total | 28.5 | 5861.2 | 108.8 | 13581.8 | -118.8 | -6404.5 | -55.1 | -510.5 | -46.2 | -814.2 |

3 *Forest type groups are ordered by the forest areas from largest to smallest.

1  **Figures**

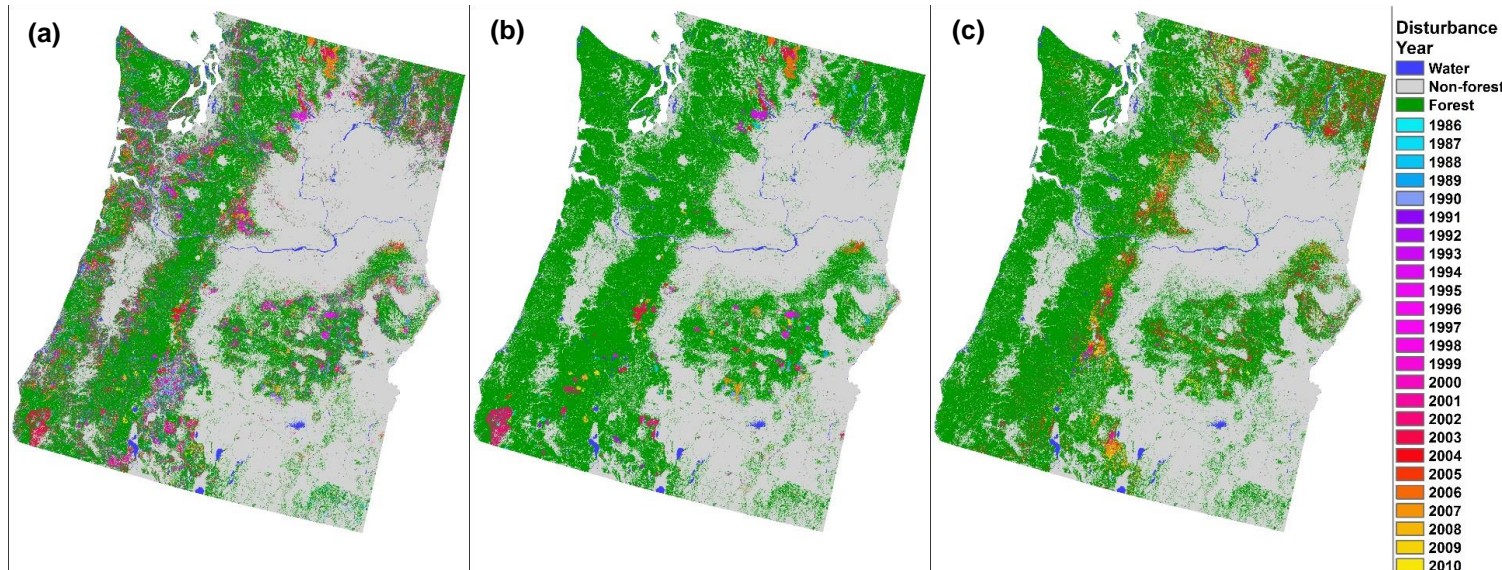

3  **Fig. 1.** Year of last disturbance from (a) NAFD, (b) MTBS and (c) ADS data of the PNW region. The time period for NAFD, MTBS and ADS
4  datasets are 1986-2010, 1986-2010 and 1997-2010 respectively. NAFD: North American Forest Dynamics, MTBS: Monitoring Trends in Burn
5  Severity, ADS: Aerial Detection Surveys.

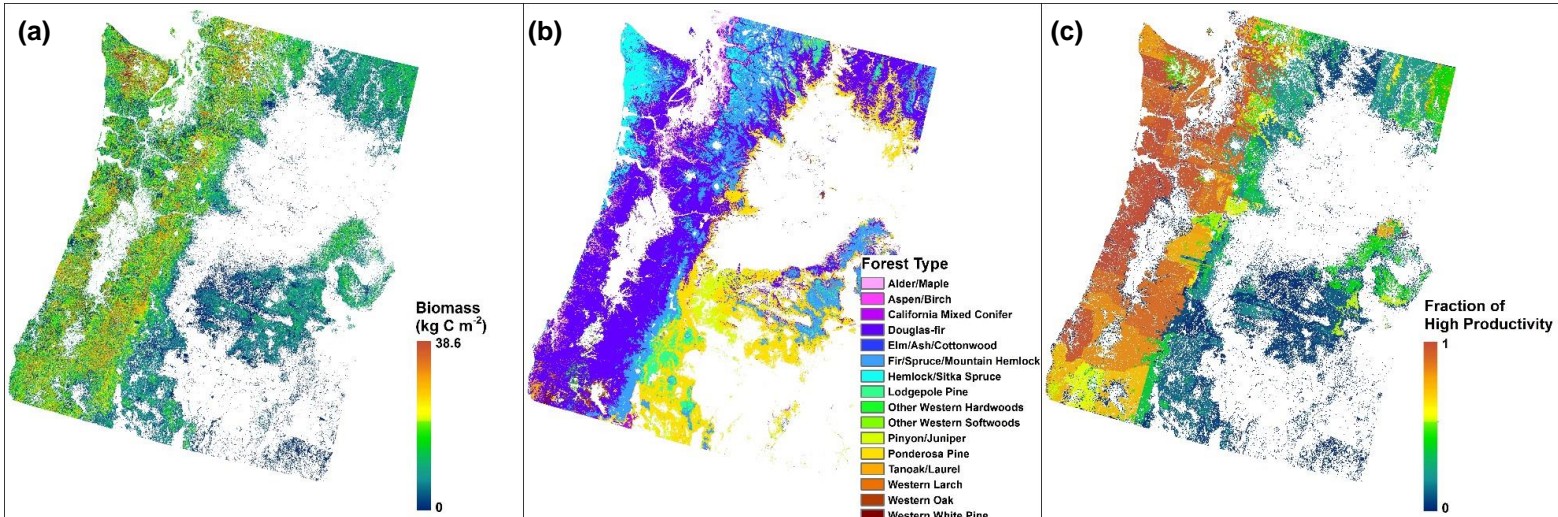

Fig. 2. Maps of (a) NBCD 2000 (National Biomass and Carbon Dataset) aboveground biomass, (b) forest type group and (c) site productivity in the PNW region.

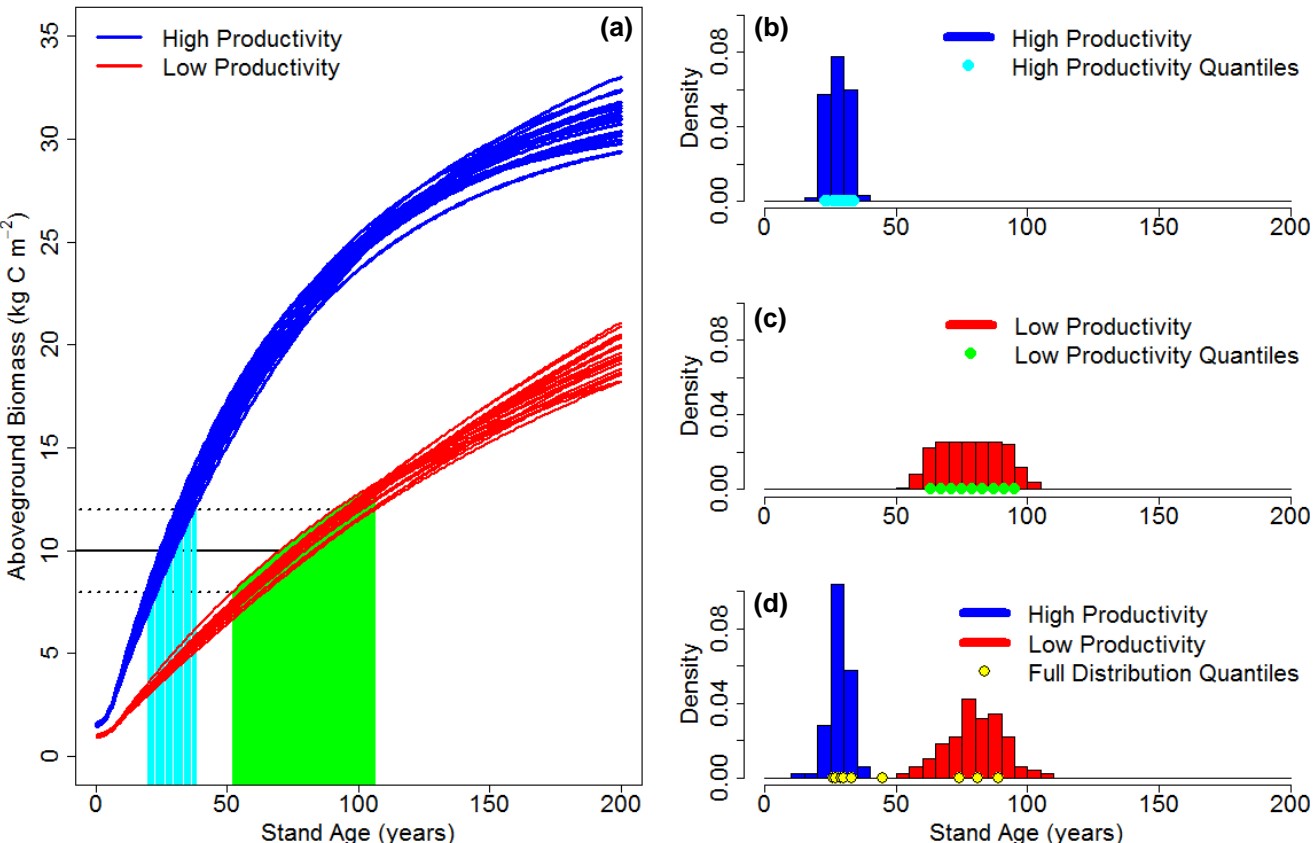

2  **Fig. 3.** Stand age inferred based on field inventory-derived species-specific biomass-age curves (a) for a forest pixel of the
3  same forest type group with aboveground biomass of 10 kg C m$^{-2}$ and having no recent disturbance. Histogram and quantiles
4  of stand ages are shown for three site productivity classes: (b) high productivity site ($f_{high}$ = 1), (c) low productivity site ($f_{high}$
5  = 0), and (d) mixture of high and low productivity sites ($f_{high}$ = 0.6).

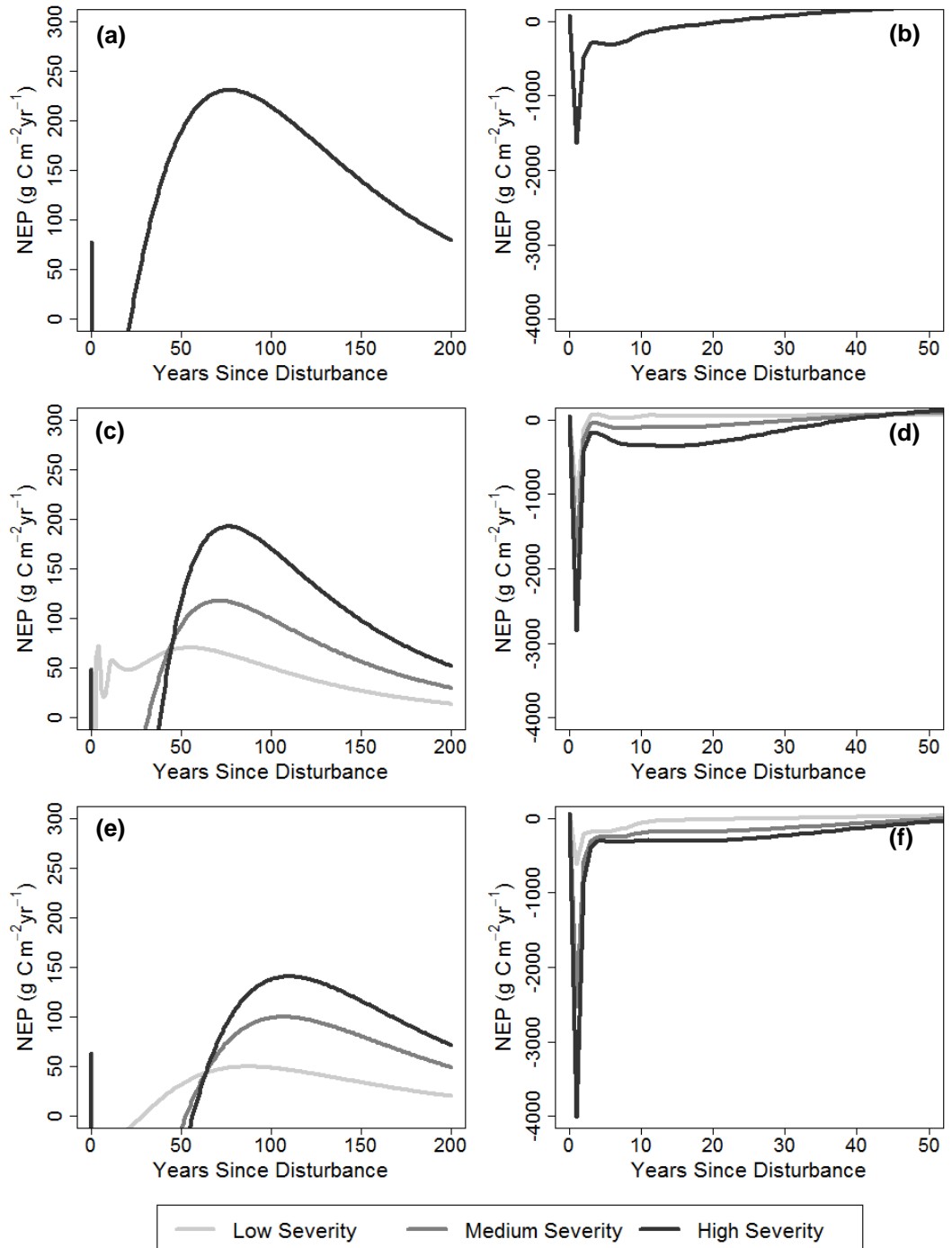

**Fig. 4.** Carbon flux trajectories of (a) (b) post-harvest (Williams et al., 2014), (c) (d) post-fire (Ghimire et al., 2012), and (e) (f) post-beetles (Ghimire et al., 2015) for a range of severities in high site productivity Douglas-fir stands of the PNW region. The typical pattern of *NEP* following a disturbance involves a large negative value immediately after disturbance, a rise for a number of years to reach a maximum rate of carbon uptake, and then a gradual decline to a steady state.

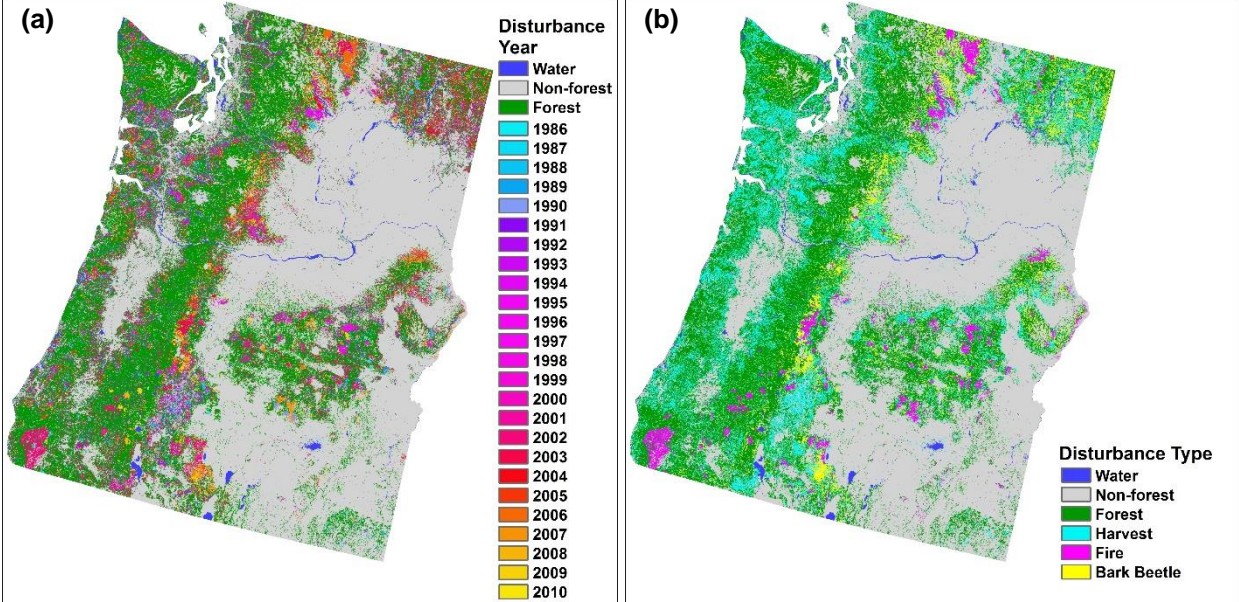

3   **Fig. 5.** Maps of (a) disturbance year and (b) disturbance type integrated from NAFD (Fig. 1a), MTBS (Fig. 1b) and ADS
4   (Fig. 1c) data in the PNW region. NAFD: North American Forest Dynamics, MTBS: Monitoring Trends in Burn Severity,
5   ADS: Aerial Detection Surveys.

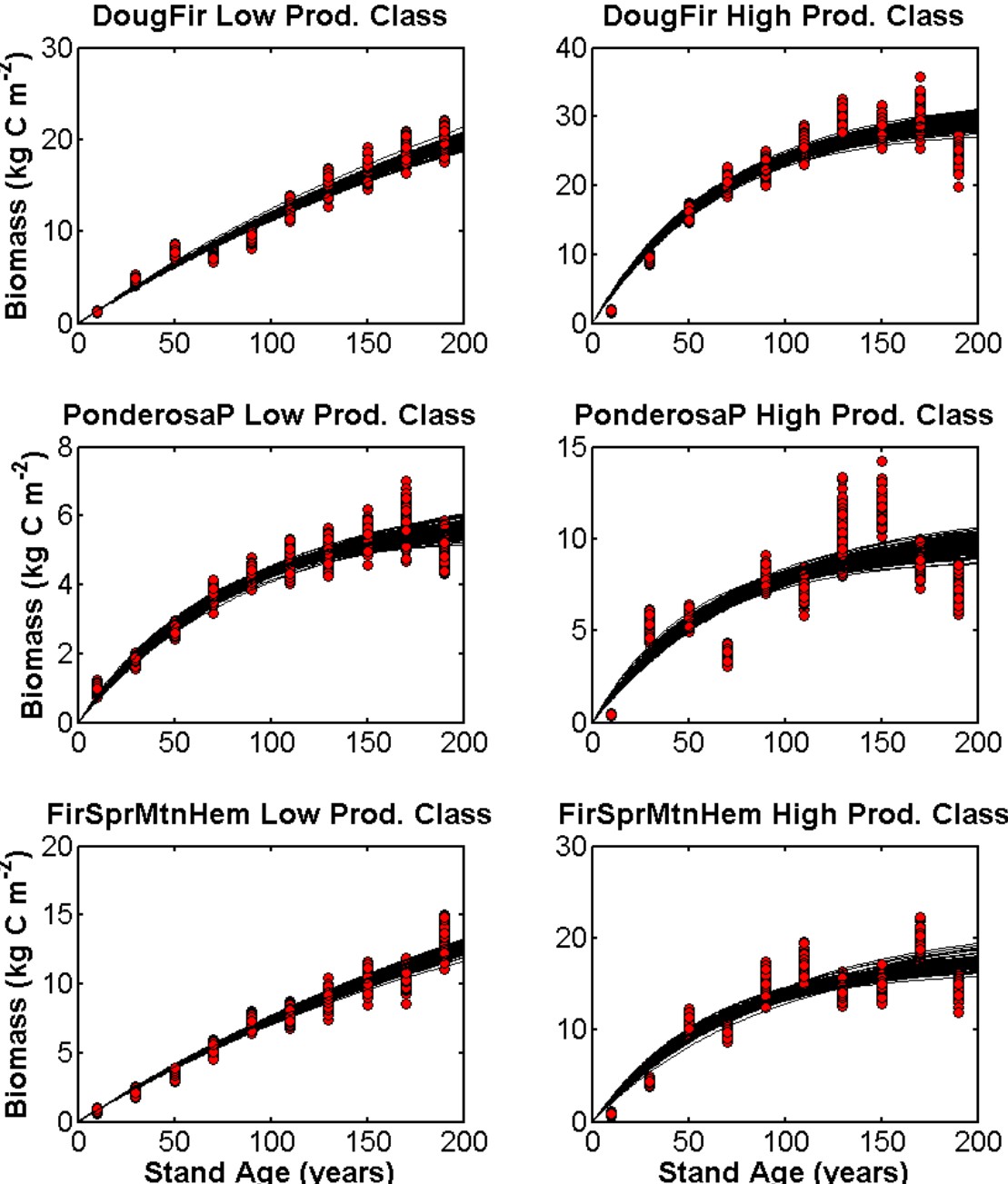

2 **Fig. 6.** Biomass-age curves sampled from FIA data for each forest type group and site productivity class in the PNW region.
3 Curves for the three most abundant forest type groups are shown (DougFir is Douglas-fir, Ponderosa P is ponderosa pine,
4 and FirSprMtnHem is fir/spruce/mountain hemlock). Red dots are independent samples drawn probabilistically from the FIA
5 data.

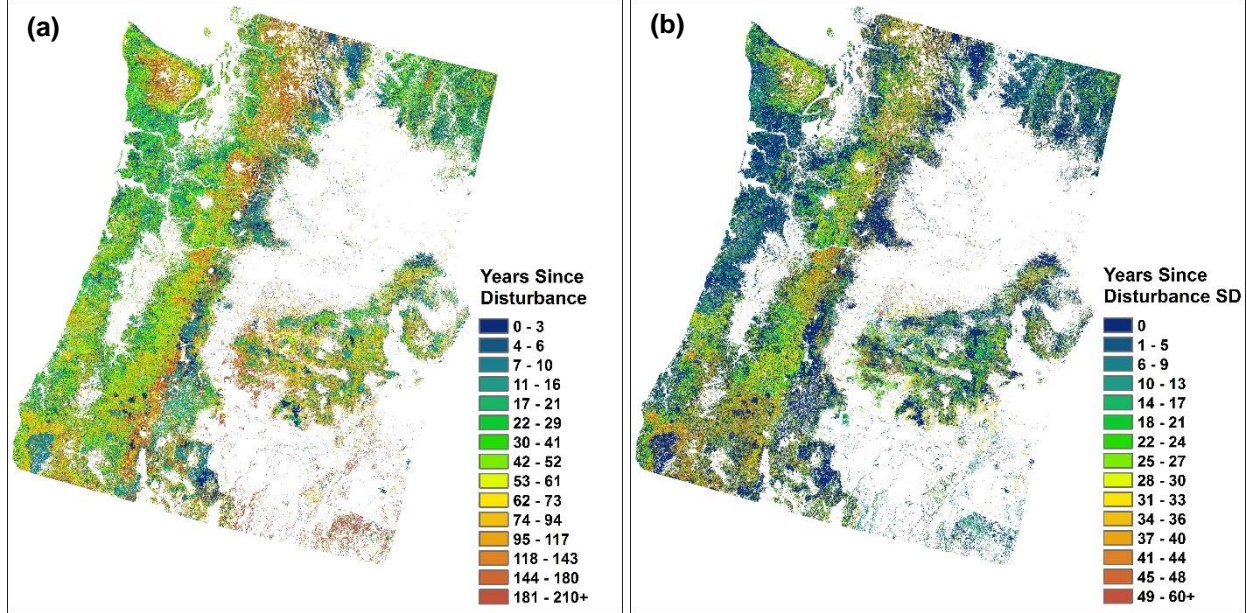

2 **Fig. 7.** Maps of (a) years since disturbance and (b) standard deviation in 2010 in the PNW region.

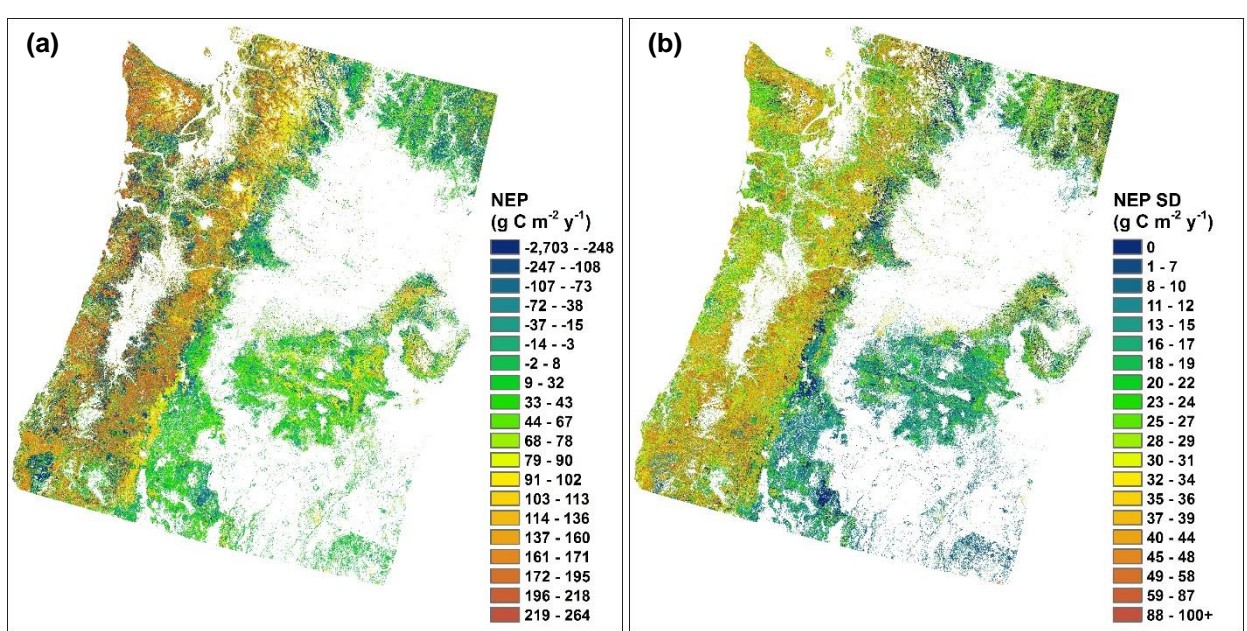

4 **Fig. 8.** Maps of (a) net ecosystem productivity (*NEP*) and (b) standard deviation in 2010 in the PNW region.

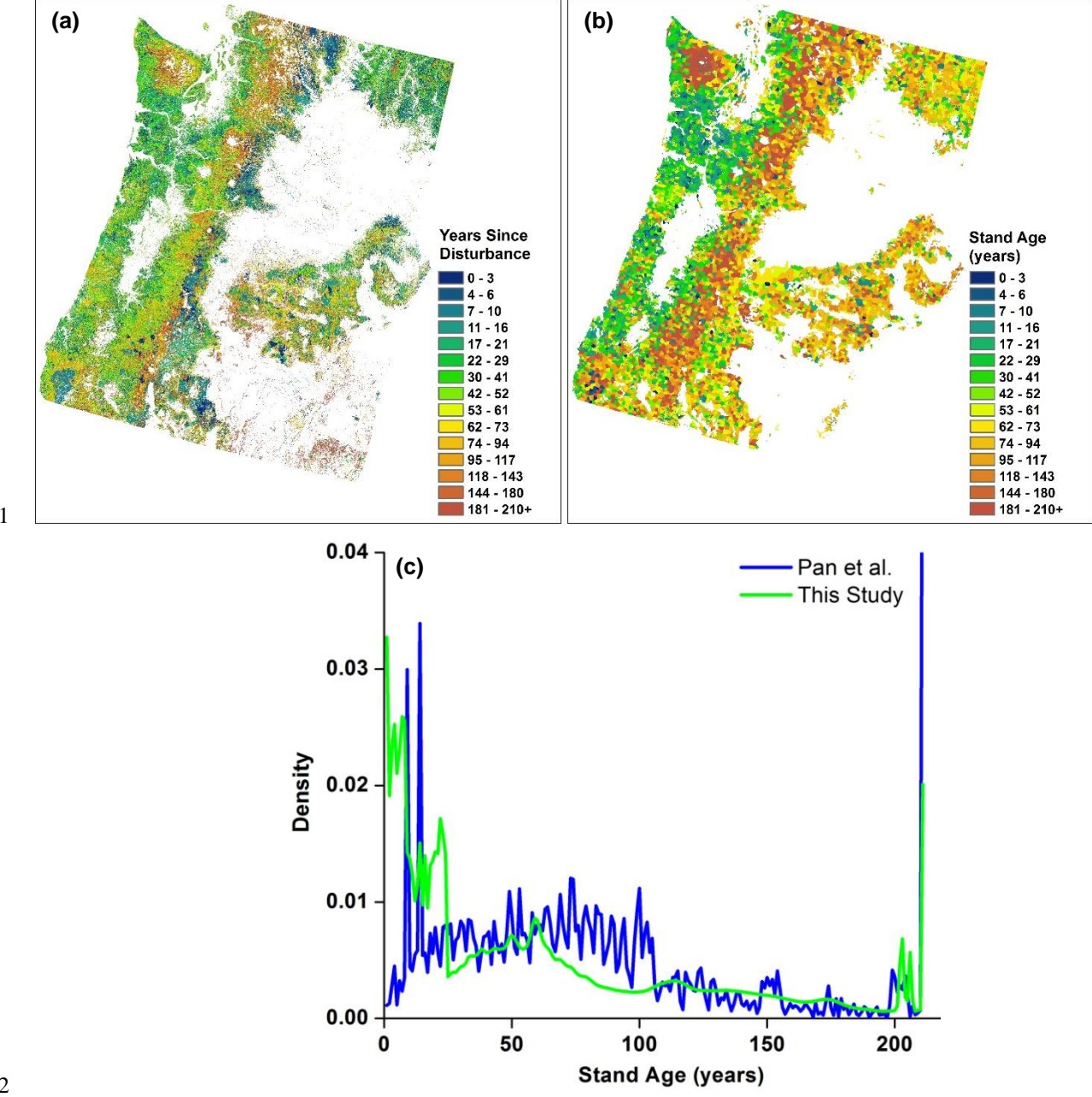

Fig. 9. Maps of time since disturbance from (a) this study (same as Fig. 7a) and stand age from (b) Pan et al. (2011) in the
PNW region, associated density curves of stand age were plotted in (c).

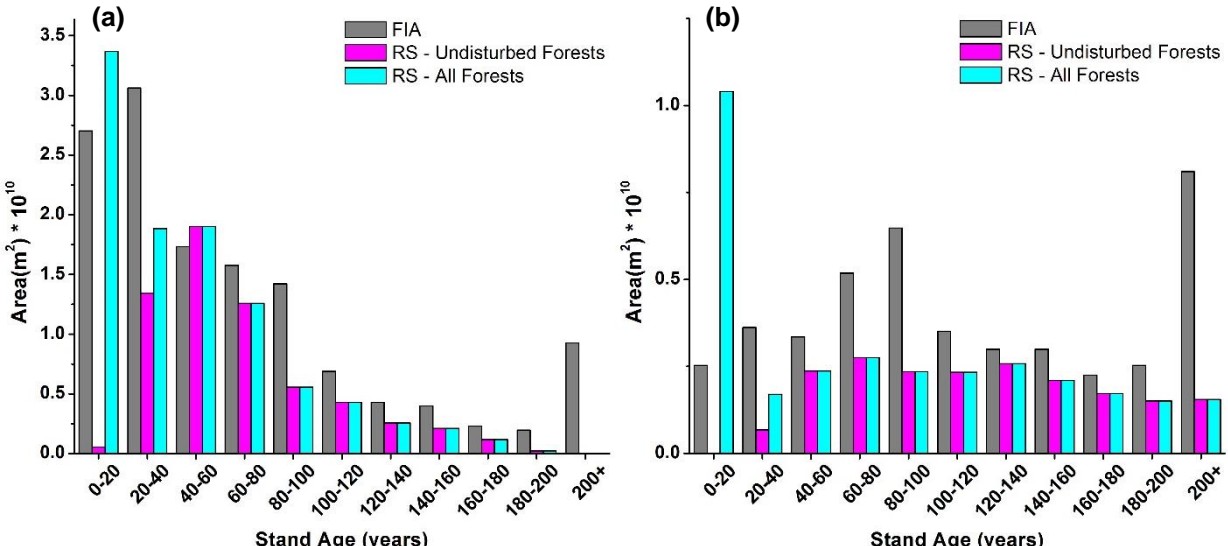

2 **Fig. 10.** Distribution of forest area by age for (a) Douglas-fir and (b) Fir-Spruce-Mountain Hemlock of the PNW region
3 comparing results from FIA data and remote sensing derived (RS-) estimates for undisturbed and all forests including those
4 marked as disturbed in NAFD, ADS, and MTBS datasets and shown here as if they were stand clearing events.