# Peer review of "High-resolution mapping of time since disturbance and forest carbon flux from remote sensing and inventory data to assess harvest, fire, and beetle disturbance legacies in the Pacific Northwest"

_Biogeosciences, 2016_

## Referee Comment (RC1) · Anonymous Referee #1 · 16 Jun 2016

General comments

The manuscript aims to address current issues in constraining forest C dynamics and stocks in relation to multiple different types and intensity of disturbance. The authors combine a range of data including national inventories, management databases, airborne and space-borne remote sensing. These data are then combined / utilized through both statistical (yield curves) and simulation based modelling (CASA) approaches. As such the manuscript is highly relevant and well within scope of Biogeosciences and I believe will ultimately be published in Biogeosciences. However I

believe there is additional scientific value that should be drawn from the current analysis and a substantial re-write to improve readability prior to publication. The following general comments are split broadly between scientific and presentation.

Globally C stored in forests is split roughly equally between woody biomass and soil organic matter (e.g. Pan et al 2011). However the manuscript focuses on estimates of above ground biomass stocks and disturbance to these stocks, lacking any analysis or discussion of soil carbon stocks. I recognize that the authors report net ecosystem productivity (defined as NEP = NPP-Rh), but I would prefer you to distinguish between accumulation and losses between the live biomass and dead organic matter. Or state clearly why not given that you are reporting ecosystem scale values. Also I do not believe that the authors have extracted all relevant information for the above ground biomass stocks. For example in Section 3.4 L29 The authors state "Spatial variations in mean annual NEP are noticeably correlated with the time since disturbance, forest type group, and site productivity strata...". This could be shown more clearly in an x∼y plot and / or this "noticeable" correlation could be explicitly quantified to distinguish the relative importance of the drivers. Further detail follows in the specific comments section.

The overall writing style of the manuscript needs to improved to benefit the flow of reading and in particular clarity. For example the methods overview needs to be clearer as to the overall structure of the analysis and their connections. The authors provide extensive detail on the data sources and what information they provide, however the fact that these data will used as constraints or drivers in the CASA model is not made clear until the final section before the results in Section 2.3.1. This is particularly confusing as all the description of data given is in reference to above ground biomass while at the same time stating that the results from the analysis are the net ecosystem productivity. Moreover the number of words in both the methods and results sections dedicated to the various disturbance maps produced appears disproportionate given that the title and the conclusions imply that C stocks and dynamics are the primary focus. I would

consider way to simplify this information and attempt to move some of it into the supplementary material. Also I note that Figures 10 and 11 do not feature in the results section at all, instead are used to introduce new information in the discussion which is inappropriate. These figures should be introduced in the results section of they could be moved to the supporting information.

Specific comments

The following comments are broken down into Abstract, Introduction, Methods, Results, Discussion and Conclusion sections. General comments on each section will be followed by specific comments with page (P) and line (L) numbers.

Abstract The abstract could be made to flow more easily and make clear that the analysis is feeding into a C-cycling model that represents both the live and dead carbon pools. This will reduce confusion between described / yield curves used which constrain above ground biomass while at the same time reporting a net ecosystem value.

Introduction All relevant information appears to be present in the introduction, however not all of the information is clear. I would recommend the use of topic-sentences to improve clarity of your message for each paragraph. Moreover there are a number of sentences where the wording is awkward to read.

P2L29 "...remote sensing techniques..." should include "...remote sensing (RS) techniques" as RS is used later. P2L30 "...remote sensing techniques provide..."

P2L32 "Such products miss small scale events and extend only so far back in time..." awkward wording. Please reconsider e.g. "However, RS products frequency miss small scale events and only cover the last several decades..." P3L8 "...provide a way forward to capture at least some of the information that is missing but needed..." awkward wording, Please reconsider rewording.

P3L9 – L20 The final paragraph would be a good place to make a clear statement of the studies objective (key questions) and novelty. However the final paragraph here

mixes further introduction and aims. This could be split and made clearer.

Methods The methods are very long (which I accept may be required) and would benefit from an improved overview section. Where possible the methods sections would benefit from moving some material to the supplementary material to improve focus.

P3L25 "...recent disturbance..." how recent? P3L28 "...terms "time since disturbance" and "stand age" " would it be possible to pick one of these terms and use it consistently? P3L29 "It was inferred..." possibly "Stand age was inferred..." would be clearer? P3L30 "The (yield?) curves were sampled from FIA data and specific to forest type and group and site productivity class." Is this information known in all cases? If not, what is assumed in their place?

P4L1 "Net ecosystem productivity (NEP)" prior to this point all data / methods mentioned implies that this study is focusing on above ground biomass only. A link to CASA needs to be made earlier to make this clear. P4L13 It would be useful to have a table with the different data sources listed and state the data and time period they cover.

P5L1 "These assumptions...been reported in the literature." Long sentence, can you break some of the sentences with lists, multiple concepts or conditions down. P5L5-10 Consider making this list in a table P5L11-12 "The target year...was 2010". Possibly make this point earlier say in the overview or introduction aims? P5L23 "age class from ..." how many age classes, are all equal in size? P5L25 All other units are given as SI. Please do so here too. Also it is odd that up until now forest biomass as been discussed, here you have swapped into volume. Can you convert or is there a reason for this?

P6L1 "Differences in forest masks..." which forest masks? Which products you are using? P6L2 "These were replaced by the mean biomass of other undisturbed pixels..." The distributions of stand age in Figure 11 are not Gaussian, would the median be better or is there little difference? P6L12-13 Again SI units please. P6L24 "In reality

fhigh is almost always between 0 and 1." Can you say what the mean value is or distributional information? Something more informative.

P7L4 This is the first mention of the CASA model. Please provide a brief description of the mode and how it works. This is needed given that you make reference to its process representation in the discussion P11L1-5. Also what is the model time step used. Over what period is CASA simulating these forests (prior to 2010)? which meteorological drivers are used (e.g. ERA-Interim, GFS)? How realistic are the spin up pool sizes relative to field estimates in undisturbed pixels. Your estimate of C loss in response to disturbance will partially dependent on soil losses which will also be dependent on their initial magnitude after spin up (e.g. Exbrayat et al., 2014). If this information is available in the cited literature please make this clear. P6L10 "...curves describing carbon fluxes and stocks..." which stocks / fluxes where are they? P6L15 "This study emphasized the use of NEP curves. Fig. 5 ..." Figure 5 seems to show that C losses do no occur whereas losses do occur in the results (Table 2) as presumably soil and litter C is being decomposed and undergoing mineralisation. So where is the C source represented?

Results

What is the primary focus of the manuscript? A large part of the results section is taken up with a description of the various input maps into the analysis. Much of it seems like it should be in the methods sections as a description of the inputs or could be moved to the supporting information. Unless these are actually new numbers derived from the combination of multiple maps. At the moment it is not clear. Possibly an overview could be given to the results as it takes a lot of reading before you get to any information on the estimates of biomass stocks.

P9L11-12 "...these curves yielded a smoothed fit to the inventory data rather than showing a saw-toothed increase with stand age." Here are you referring to saw-toothed due to managed thinning or stem mortality events? P9L23 "Uncertainty on the time since disturbance forest pixels is not currently available from disturbance products and

this was not mapped" Could the uncertainty in the yield curves on growth since distur-bance be included? How strongly do the yield curves constrain CASA? P9L29 "Spatial variations in mean annual NEP are noticeably correlated with..." Why not actually cor-relate them to quantify this? A new x∼y figure might be useful here too. P9L30-31 "...weaker carbon sinks in the eastern, drier portion of the study area..." Again this could be show in an x∼y plotting soil moisture / precipitation against C sink strength to quantify.

P10L6 "Forestlands free of recent disturbance..." could be "Undisturbed forests are..." just trying to be consistent with the terms you use.

Discussion

P10L21-23 Awkward sentence please rephrase / breakdown into smaller parts. P10L25-26 Odd place the begin new paragraph. You appear to be continuing your point from the first paragraph. P10l27 It is not clear what you mean. Are you talk-ing about how the stand-level biomass estimate was calculated or how the real world stand was managed / grew? P10L29 "...or also from a recent disturbance that reduced biomass to the current level." I think you need a reference here. P10L30 "...varies depending on the type of stand-replacing disturbance". Are you referring to e.g. clear felling vs fire?

P11L1-4 Currently you have not described the model used to provide required back-ground for these statements. P11L4 "...initial rise through stand initialization." Are you talking about early phases of forest growth? How long does initialization take? P11L9 "...which are sure to have errors." Are there any estimates of this error? P11L9-13 Should this not be first introduced in the methods section if these describe errors be-tween field information and the maps you have used to constrain your model. Also, is there a bias associated with these errors? If so, how do you expect these biases to impact your analysis. Might a bias here impact the differing conclusions between here and your previous works? P11L27-28 Introducing new information in figures which are

not described in the results. This should not be the case. If the figure and comparison is really needed then it should be included in the results and be part of the experimental design. or could be moved to SI.

P12L12-15 New analysis should not be introduced in the discussion. Also Figure 11 did not appear in the results either. Again, if these comparison and figure is needed then make it part of the experimental design and introduce it in the results section first. P12L14-15 "...distribution agrees well with that for our undisturbed..." poor working rephrase. P12L18, P13L4,L18 Multiple definitions of what is a young forest. Can you reconcile these? P12L22 "A portion of this difference can be attributed to smaller net carbon losses..." If I understand correctly here you mean greater loss / more negative? Comparing between -4 TgC and -7 TgC? Not clear. P12L31-33 Is the PNW region representative of forestry in the US? P13L19 Good to see some comparison with other studies. Are there any more available to broaden the discussion?

Figures

All of the figure captions need to be expanded to make clear where the data / analysis from each figure comes from and any key features. Also there appears to be substantial repetition of the disturbance figure. Can the figures be re-arrange to minimize this / move some of these maps to the SI.

Figure 5. These NEP do not show C loss, even though your analysis does. These figures reinforce the confusion between whether or not you are analysis the C balance of the ecosystem as a whole or just the live biomass. If you are analyzing the whole ecosystem the NEP would surely be negative directly after disturbance due to litter and soil C turnover?

Figure 10. In your analysis are "Years Since Disturbance" and "Stand Age" the same thing? If so why in the same figure are you referring to this by different names. Particularly as in the caption you refer to both as "Stand age".

References

Exbrayat, J.F., Pitman, A.J. and Abramowitz, G. Response of microbial decomposition to spin- up explains cmip5 soil carbon range until 2100. Geoscientific Model Development, 7(6): 2683–2692, 2014. doi: 10.5194/gmd-7-2683-2014

Pan, Y., Birdsey, R.A., Fang, J. et al. A Large and Persistent Carbon Sink in the World's Forests. Science, 333(6045):988–993, 2011. doi: 10.1126/science.1201609.

––––––––––––––––––––––––––––––

---

## Referee Comment (RC2) · Anonymous Referee #2 · 21 Jun 2016

This is a pretty good and potentially useful paper that could be published after some modifications. The main problems I can identify are (1) the introduction is poorly written in places, (2) an important and highly relevant citation is missing, and (3) the discussion needs more work.

Detailed comments below address some problems with the introduction. The missing citation is more troubling since it presents an alternative approach to using the CASA model for estimating growth (or NEP) which is a center piece of this study. The citation is: Raymond, C. L., Healey, S., Peduzzi, A., Patterson, P. 2015. Representative

regional models of post-disturbance forest carbon accumulation: Integrating inventory data and a growth and yield model. Forest Ecology and Management 336: 21-34. This should be referenced in a couple of places (p. 2 line 20 and p. 4 line 1).

The discussion should compare using the CASA model and using the Raymond et al. approach which relies on an FIA driven empirical model, the Forest Vegetation Simulator (FVS). What are the advantages and disadvantages of each approach, and do they yield similar results (the regions are different but still may be able to compare results for one or two forest types). I also suggest that the discussion should explore in more depth the many assumptions and inferences that have to be made to estimate time since disturbance for "undisturbed" pixels (section 2.2.2). For example, the Kellndorfer biomass map used to estimate biomass of "undisturbed" pixels has fairly high uncertainty at the pixel level; some pixels were assigned forest types based on a nearby neighbor pixel, etc. By the way, the title of section 2.2.2 is an oxymoron – if the pixel is "undisturbed" there should not be a time since disturbance. So instead of "undisturbed" the authors should use a different term to identify pixels that had no detected disturbance since 1986, perhaps something like "recently undisturbed".

Specific comments

The title is too long and redundant. Suggest "High-resolution forest carbon flux mapping in the Pacific Northwest with disturbance legacies inferred from remote sensing and inventory data". Could also leave out "in the Pacific Northwest".

p. 1 line 22: delete the second "probabilistic,"

p. 1 line 26: re-word so that it does not appear that tracts of land can somehow "see".

p. 2 line 13: replace "is itself a sort of record of" with "reflects"

p. 2 line 14: replace "general" with "predictable rate of"

p. 2 line 22-23: needs some rewording. The idea is that it is important to include small-scale disturbances down to some minimum threshold, not that disturbances typically

are at this small scale.

p. 2 lines 27-28: add "at smaller scales" to the end since national forests inventories can provide useful guidance only at larger scales. But importantly note, it is possible to conduct field inventories at very small scales, so the statement is not very correct at all, only partially correct with respect to national forest inventories.

p. 3 lines 11-12: One objective is clearly stated. What are the others? The last sentence of this paragraph seems to be another objective, but then, I'm confused as to whether the purpose is to develop a method for large-scale monitoring and management, or small-scale, or both?

p. 3 lines 27-28. Terminology again – "undisturbed" pixels by definition should not have a time since disturbance.

p. 3 line 32: Biomass curves were developed by forest type group and productivity class. How were these 2 classes allocated to the 5 NEP classes described on p. 4 lines 2-4?

p. 3 line 35: add citation after "…varying severity".

p. 6 line 26: replace "stand" with "standard"

p. 8 line 21: sentence that begins with "Again" needs editing.

p. 9 lines 24-25: the imprint is not so clear to me. Maybe need to highlight somehow on the graphic.

p. 10 lines 12-14: One could argue that inventory data does not provide such a reliable estimate of biomass/age. Both of these variables can be rather difficult to measure/estimate especially with respect to the selection of biomass equations, but also the difficulty of assigning a stand age to stands that are uneven-aged.

p. 12 lines 1-7: not stated—FIA does not do a good job of detecting recent disturbances because the remeasurement cycle in the PNW is about 10 years, so the

average time lag of the data at any point in time is at least 5 years.

---

## Author Comment (AC1) · 6 Aug 2016

*We would like to thank the reviewer for giving us very helpful suggestions that help us greatly improve the quality of this manuscript. We provided our responses to all the comments point by point below (italicized typeface).*

General comments

The manuscript aims to address current issues in constraining forest C dynamics and stocks in relation to multiple different types and intensity of disturbance. The authors combine a range of data including national inventories, management databases, airborne and space-borne remote sensing. These data are then combined / utilized through both statistical (yield curves) and simulation based modelling (CASA) approaches. As such the manuscript is highly relevant and well within scope of Biogeosciences and I believe will ultimately be published in Biogeosciences. However I believe there is additional scientific value that should be drawn from the current analysis and a substantial re-write to improve readability prior to publication. The following general comments are split broadly between scientific and presentation.

Globally C stored in forests is split roughly equally between woody biomass and soil organic matter (e.g. Pan et al 2011). However the manuscript focuses on estimates of above ground biomass stocks and disturbance to these stocks, lacking any analysis or discussion of soil carbon stocks. I recognize that the authors report net ecosystem productivity (defined as NEP = NPP-Rh), but I would prefer you to distinguish between accumulation and losses between the live biomass and dead organic matter. Or state clearly why not given that you are reporting ecosystem scale values. Also I do not believe that the authors have extracted all relevant information for the above ground biomass stocks. For example in Section 3.4 L29 The authors state "Spatial variations in mean annual NEP are noticeably correlated with the time since disturbance, forest type group, and site productivity strata...". This could be shown more clearly in an x~y plot and / or this "noticeable" correlation could be explicitly quantified to distinguish the relative importance of the drivers. Further detail follows in the specific comments section.

*Response: We apologize that our title didn't fully convey main objectives and focuses of this manuscript, we edited the title as "High-resolution mapping of time since disturbance and forest carbon flux from remote sensing and inventory data-inferred disturbance legacies in the Pacific Northwest". Quantification and mapping of time since disturbance is an important objective in this paper, while carbon flux is meant to be a relatively minor focus.*

*We rewrote the last paragraph in Introduction, all the objectives were listed now. Aboveground biomass accumulation curves were used for objective 1 to infer stand age from RS-derived biomass data. Carbon flux curves were used in objective 2, which consider carbon accumulation and loss, also live biomass and dead organic matter. Section 2.3.1 in the new version provides a revised and expanded description of the model, including its approach to dead organic carbon cycling and also its inclusion of disturbance processes. Since the processes of the CASA model used to derive carbon flux trajectories were described in detail in our prior papers (Williams et al., 2012, Ghimire et al., 2012, Ghimire et al., 2015), here we mainly focused on the approach of combining these trajectories with the newly mapped time since*

*disturbance derived from objective 1 to simply demonstrate that this method can be used to develop spatial representations of NEP.*

*For section 3.4 L29, we estimated NEP based on carbon flux trajectories, which vary by time since disturbance, forest type group and site productivity. The relationships between NEP and these variables have been presented in the trajectory curves in Fig. 4. Because these trajectories are directly applied to map NEP, there is limited new information, if any, in the fact that NEP is correlated with time since disturbance, forest type group, and site productivity strata. Therefore, we think x~y plot is unnecessary. Furthermore, our intention is not to explore relationships between NEP and input data. Correspondingly, we revised the sentence in question to read "Spatial variations in mean annual NEP are determined by differences in strata of ..."*

The overall writing style of the manuscript needs to improved to benefit the flow of reading and in particular clarity. For example the methods overview needs to be clearer as to the overall structure of the analysis and their connections. The authors provide extensive detail on the data sources and what information they provide, however the fact that these data will used as constraints or drivers in the CASA model is not made clear until the final section before the results in Section 2.3.1. This is particularly confusing as all the description of data given is in reference to above ground biomass while at the same time stating that the results from the analysis are the net ecosystem productivity. Moreover the number of words in both the methods and results sections dedicated to the various disturbance maps produced appears disproportionate given that the title and the conclusions imply that C stocks and dynamics are the primary focus. I would consider way to simplify this information and attempt to move some of it into the supplementary material. Also I note that Figures 10 and 11 do not feature in the results section at all, instead are used to introduce new information in the discussion which is inappropriate. These figures should be introduced in the results section of they could be moved to the supporting information.

**Response:** *Again, we apologize that our title didn't fully convey main objectives and focuses of this manuscript, we edited the title as "High-resolution mapping of time since disturbance and forest carbon flux from remote sensing and inventory data-inferred disturbance legacies in the Pacific Northwest". Quantification and mapping of time since disturbance is an important objective in this paper, and while it is being generated for the purpose of carbon flux, the characterization of carbon fluxes was the focus of the group's prior work and is thus of less emphasis here. We did revise the present paper to include a more comprehensive overview of the carbon modeling performed in our prior work.*

*We rewrote the last paragraph in Introduction where all of the objectives are now listed. One of the main objectives is to introduce a new method for inferring a pixel's representative time since disturbance by using a number of currently available data sources. All the input data sources can be used directly except for a disturbance map with disturbance type/severity and biomass~age growth curves. So we provided explicit description of how to derive these two important components, and the resulting maps, figures and numbers. Another objective is to make use of the time since disturbance map that we derived for objective 1 with carbon flux trajectory curves from our previous work to map NEP. Since the processes of the CASA model and the methods for deriving carbon flux trajectories were each described in detail in our prior papers (Williams et al., 2012, Ghimire et al., 2012, Ghimire et al., 2015), here we emphasize only the approach of applying these trajectories and resulting NEP maps.*

*For Fig. 10 and Fig. 11, we divided the two relevant paragraphs in "Discussion" into three parts. The sentence introducing the comparison between the results from this study and those of previous work was moved to "Materials and Methods" and the comparison of those results (formerly Fig. 10 and Fig. 11 ,*

*now Fig. 9 and Fig. 10) were moved to "Results". We keep associated discussion about likely sources of the discrepancy in the "Discussion" section.*

Specific comments

The following comments are broken down into Abstract, Introduction, Methods, Results, Discussion and Conclusion sections. General comments on each section will be followed by specific comments with page (P) and line (L) numbers.

Abstract The abstract could be made to flow more easily and make clear that the analysis is feeding into a C-cycling model that represents both the live and dead carbon pools. This will reduce confusion between described / yield curves used which constrain above ground biomass while at the same time reporting a net ecosystem value.

***Response:*** *We edited the abstract to reflect objectives of this manuscript. We clarified biomass growth curves and carbon flux trajectories were used for different objectives. The CASA carbon cycle model was not mentioned in the abstract, since this is not the focus of this paper.*

Introduction All relevant information appears to be present in the introduction, however not all of the information is clear. I would recommend the use of topic-sentences to improve clarity of your message for each paragraph. Moreover there are a number of sentences where the wording is awkward to read.

***Response:*** *We rewrote most of the "Introduction", making the objectives much clearer and the content proportional to the objectives.*

P2L29 "...remote sensing techniques..." should include "...remote sensing (RS) techniques" as RS is used later.

***Response:*** *Acronym of remote sensing has been included in brackets.*

P2L30 "...remote sensing techniques provide..."

***Response:*** *"... are providing ..." has been changed to "... generate ..." to avoid repeat use of the word "provide" in the same sentence.*

P2L32 "Such products miss small scale events and extend only so far back in time..." awkward wording. Please reconsider e.g. "However, RS products frequency miss small scale events and only cover the last several decades..."

***Response:*** *This sentence was edited as "However, such disturbance products miss small-scale events and only provide a record of events that occurred during the last several decades, ..."*

P3L8 "...provide a way forward to capture at least some of the information that is missing but needed..." awkward wording, Please reconsider rewording.

***Response:*** *This sentence was edited as "Nonetheless, RS-derived forest biomass still provides a valuable way of characterizing the legacy effects of disturbance that occurred prior to RS observations, which is required for quantifying carbon stock recovery and carbon uptake and release rates over large areas."*

P3L9 – L20 The final paragraph would be a good place to make a clear statement of the studies objective (key questions) and novelty. However the final paragraph here mixes further introduction and aims. This could be split and made clearer.

*Response: We rewrote the last paragraph in Introduction, all the objectives were listed. Now it reads as "This study estimates and maps time since disturbance at a fine scale of 30 m from RS-derived products and FIA-derived biomass growth curves, and then maps net ecosystem productivity (NEP) based on disturbance history, time since disturbance and carbon flux legacy. The specific objectives in this study are to: (1) introduce a method for inferring a pixel's representative time since disturbance from RS-derived biomass and disturbance products at the 30 m resolution; (2) map NEP based on model-derived carbon stock and flux trajectories that describe how NEP changes with time following harvest, fire, or bark beetle disturbances of varying severity; (3) propagate uncertainties from RS-derived biomass products and FIA into uncertainty quantification of stand age and NEP. Our research represents an approach to map carbon stocks and fluxes at a high resolution across the conterminous US in support of national carbon monitoring, reporting, and management."*

Methods The methods are very long (which I accept may be required) and would benefit from an improved overview section. Where possible the methods sections would benefit from moving some material to the supplementary material to improve focus.

*Response: We made several edits in "2 Materials and Methods", especially making the "2.1 Overview" more clarified.*

P3L25 "...recent disturbance..." how recent?

*Response: "...recent disturbance…" refers to disturbance since the starting year of disturbance product as noted in the manuscript.*

P3L28 "...terms "time since disturbance" and "stand age" " would it be possible to pick one of these terms and use it consistently?

*Response: We prefer to use term "time since disturbance" based on context of this paper, while in FIA and other papers, such as Pan et al. (2011), "stand age" was mainly used for undisturbed forests. To make the connection between two terms, here we wrote: terms "time since disturbance" and "stand age" are used interchangeably for recently undisturbed forest pixels thereafter.*

P3L29 "It was inferred..." possibly "Stand age was inferred..." would be clearer?

*Response: The sentence was edited as "Stand age was inferred …"*

P3L30 "The (yield?) curves were sampled from FIA data and specific to forest type and group and site productivity class." Is this information known in all cases? If not, what is assumed in their place?

*Response: Two out of 16 forest type groups did not have FIA-derived biomass-age curves available, they are "Pinyon/Juniper" and "California Mixed Conifer", so we used curves of "Other Western Softwood" instead.*

P4L1 "Net ecosystem productivity (NEP)" prior to this point all data / methods mentioned implies that this study is focusing on above ground biomass only. A link to CASA needs to be made earlier to make this clear.

*Response: Two main objectives in this paper are to map time since disturbance and NEP. In section "2.1 Overview", we spent the first paragraph on methods to infer time since disturbance, and the second paragraph on mapping NEP. So we think it's appropriate to mention NEP at the beginning of the second paragraph. Besides, CASA model was mentioned here now.*

P4L13 It would be useful to have a table with the different data sources listed and state the data and time period they cover.

*Response: We added a new table (Table 1, shown below) in the manuscript to summarize the data sources.*

**Table 1.** Data sources for inferring time since disturbance for recently disturbed and undisturbed forest pixels.

| Data | Description | Source | Year | Input for recently disturbed or/and undisturbed forests |
|---|---|---|---|---|
| NAFD | Forest disturbance | Landsat | 1986-2010 | a, b |
| MTBS | Burned area and severity | Landsat | 1986-2010 | a |
| ADS | Area of insect outbreak and number of trees killed | Aerial survey | 1997-2010 | a |
| NBCD | Aboveground live biomass | Landsat, SRTM, FIA | 2000 | b |
| Forest Type Group | Forest type group | MODIS, NLCD, etc. | 2001 | b |
| Site Productivity | Fraction of high productivity | FIA | 1984-2014 | b |
| Biomass-age Curves | Biomass accumulation as a function of stand age | FIA | 1984-2010 | b |

[a] Data is one of the inputs for inferring time since disturbance for recently disturbed forest pixels.
[b] Data is one of the inputs for inferring time since disturbance for recently undisturbed forest pixels.

P5L1 "These assumptions...been reported in the literature." Long sentence, can you break some of the sentences with lists, multiple concepts or conditions down.

*Response: This sentence has been broken into a short sentence with three lists. It reads as "These assumptions are based on the rationales: (1) MTBS records most of the notable fire events in the region, (2) harvest events are one of the most ubiquitous stand replacing disturbance types active in the region, (3) ADS-mapped polygons of bark beetle infestations often include unaffected stands as has been reported in the literature".*

P5L5-10 Consider making this list in a table

*Response: We keep four rules in lists.*

P5L11-12 "The target year...was 2010". Possibly make this point earlier say in the overview or introduction aims?

*Response: Target mapping year of 2010 was now mentioned in "2.1 Overview".*

P5L23 "age class from ..." how many age classes, are all equal in size?

*Response: There are 11 age classes in total: 0-20, 20-40, 40-60, 60-80, 80-100, 100-120, 120-140, 140-160, 160-180, 180-200, 200+. This information has now been included in the revised manuscript.*

P5L25 All other units are given as SI. Please do so here too. Also it is odd that up until now forest biomass as been discussed, here you have swapped into volume. Can you convert or is there a reason for this?

*Response: The unit "cubic feet/acre/year" here is used by FIA and is the output unit from FIA EVALIDator. We didn't convert it to SI, since readers will know how we combine 7 site productivity classes (0-19, 20-49, 50-84, 85-119, 120-164, 165-224, 224+ cubic feet/acre/year) into two classes.*

P6L1 "Differences in forest masks..." which forest masks? Which products you are using?

*Response: Different forest masks were used in NAFD disturbance products and NBCD biomass products. This sentence was edited as "Differences in forest masks between NAFD disturbance and NBCD biomass products led to". In our study, NAFD-based forest mask was used for the analyses and mapping.*

P6L2 "These were replaced by the mean biomass of other undisturbed pixels..." The distributions of stand age in Figure 11 are not Gaussian, would the median be better or is there little difference?

*Response: This is a great point. We do compare mean and median biomass of other undisturbed pixels with the same forest type and site productivity class (10 classes here from 0 to 1 with equal interval), and we found there is little difference between mean and median values.*

P6L12-13 Again SI units please.

*Response: Same reason as previous comment on SI. That is, we would like to keep the unit consistent with outputs from FIA EVALIDator, so readers would know how we combine 7 site productivity classes (0-19, 20-49, 50-84, 85-119, 120-164, 165-224, 224+ cubic feet/acre/year) into two classes.*

P6L24 "In reality fhigh is almost always between 0 and 1." Can you say what the mean value is or distributional information? Something more informative.

*Response: maximal $f_{high}$ = 0.996, minimal $f_{high}$ = 0.015 and mean $f_{high}$ = 0.530 in PNW. We included maximal $f_{high}$ and minimal $f_{high}$ values in the manuscript.*

P7L4 This is the first mention of the CASA model. Please provide a brief description of the mode and how it works. This is needed given that you make reference to its process representation in the discussion P11L1-5. Also what is the model time step used. Over what period is CASA simulating these forests (prior to 2010)? which meteorological drivers are used (e.g. ERA-Interim, GFS)? How realistic are the spin up pool sizes relative to field estimates in undisturbed pixels. Your estimate of C loss in response to disturbance will partially dependent on soil losses which will also be dependent on their initial magnitude after spin up (e.g. Exbrayat et al., 2014). If this information is available in the cited literature please make this clear.

*Response: We now include the following description: "The CASA model operates on a monthly time step. It uses a light use efficiency approach to simulating net primary productivity (NPP) based on RS-derived absorption of photosynthetically active radiation, biome parameters, and climate data. The model then allocates NPP to three live carbon pools (leaves, roots, and wood), and transfers carbon to dead pools (litter and soils) based on biome-specific rates of tissue turnover. Carbon in dead organic matter pools is transferred between pools depending on the rate and efficiency of heterotroph consumption which varies in the model based on biome-specific litter chemistry and site-specific climate setting. The default model parameters that influence NPP and wood turnover (mortality and shedding), and hence accumulation of live biomass, were adjusted by forest type based on fits to yield data from the forest inventory and analysis dataset. Aboveground live biomass per unit area versus stand age was sampled from the forest inventory and analysis data for individual forest type and site productivity class strata." Meteorological and satellite based drivers, soil type, biome (forest) type, and other parameter and driver datasets are all described in the original papers from which the corresponding results were directly taken.*

P7L10 "...curves describing carbon fluxes and stocks..." which stocks / fluxes where are they?

*Response: They include aboveground forest stock, NPP, heterotrophic respiration and NEP. This sentence was edited as "...curves describing aboveground forest stock, NPP, Rh and NEP with time since*

*disturbance ...". In fact, we have estimates of carbon stocks for all live and dead pools in the model (about 10 more) but here we restrict our use to those mentioned.*

*All these curves were derived and shown in our prior work (Williams et al., 2012, Ghimire et al., 2012, Ghimire et al., 2015), citations were added at the end of this sentences.*

P7L15 "This study emphasized the use of NEP curves. Fig. 5 ..." Figure 5 seems to show that C losses do no occur whereas losses do occur in the results (Table 2) as presumably soil and litter C is being decomposed and undergoing mineralisation. So where is the C source represented?

*Response: Fig. 5 only displays positive NEP trajectories after disturbance to support a scale that enhances the ability to compare curves for different forest types and productivity classes. More texts were added in caption of Fig. 5 (now Fig. 4), "The typical pattern of NEP following a disturbance involves a large negative value immediately after disturbance, a rise for a number of years to reach a maximum rate of carbon uptake, and then a gradual decline. Only the positive part of NEP trajectories were displayed."*

Results

What is the primary focus of the manuscript? A large part of the results section is taken up with a description of the various input maps into the analysis. Much of it seems like it should be in the methods sections as a description of the inputs or could be moved to the supporting information. Unless these are actually new numbers derived from the combination of multiple maps. At the moment it is not clear. Possibly an overview could be given to the results as it takes a lot of reading before you get to any information on the estimates of biomass stocks.

*Response: We included four sections in "Results", the first is about a disturbance map, and the second is on biomass~age growth curves. We provided the resulting maps, figures and numbers of a disturbance map with disturbance type/severity and biomass growth curves, because all the input data for inferring time since disturbance can be readily used except for these two important components, besides these results are new. The third and fourth sections are maps of time since disturbance and NEP, which are the main objectives of this paper. We hope that the substantial revisions we have made clarify the corresponding focus of this manuscript.*

P9L11-12 "...these curves yielded a smoothed fit to the inventory data rather than showing a saw-toothed increase with stand age." Here are you referring to saw-toothed due to managed thinning or stem mortality events?

*Response: FIA samples were compiled from FIA plot measurements in Oregon and Washington. Managed thinning or stem mortality events could be part of reasons, but there are also some other possible reasons causing the erratic and fluctuating jumps, such as vagueries of plot-to-plot variability that span climate, soils, topographic and other variations across sampling plots.*

P9L23 "Uncertainty on the time since disturbance forest pixels is not currently available from disturbance products and this was not mapped" Could the uncertainty in the yield curves on growth since disturbance be included? How strongly do the yield curves constrain CASA?

*Response: No, it would not be logically sound or feasible to use uncertainty in the yield curves to characterize uncertainty in the time since disturbance mapped from remote sensing (Landsat spectral reflectances).*

*The yield curves provide very important adjustments to the default NPP and wood turnover rates in the biome-scale parameters in the CASA model. Without this adjustment, the CASA model, which is designed*

*for global scale applications, would not provide an accurate and fitting representation of forest biomass for the fine-scale and diverse settings of the US where we are applying the model.*

P9L29 "Spatial variations in mean annual NEP are noticeably correlated with..." Why not actually correlate them to quantify this? A new x~y figure might be useful here too.

*Response: We estimated NEP based on carbon flux trajectories, which vary by time since disturbance, forest type group and site productivity. The relationships between NEP and these variables have been presented by trajectory curves in Fig. 4, and are directly applied in the mapping exercise so there is no new information derived in the mapping aside from the spatial allocation. The patterns are indeed interesting and can be important for some applications but an x~y plot would not be particularly instructive (it would simply recover the trajectories applied in the mapping). Besides, we didn't mean to explore relationships between NEP and input data. So we edited this sentence as "Spatial variations in mean annual NEP are determined by differences in strata of ..."*

P9L30-31 "...weaker carbon sinks in the eastern, drier portion of the study area..." Again this could be show in an x~y plotting soil moisture / precipitation against C sink strength to quantify.

*Response: Here we meant to describe a spatial pattern in NEP map, which has lower NEP values in the east side of the study area. We mentioned "drier" just because the eastern part of study area is less humid from our knowledge. We do not intend to emphasize the relationship between NEP and soil moisture/precipitation. We removed "drier" from the text.*

P10L6 "Forestlands free of recent disturbance..." could be "Undisturbed forests are..." just trying to be consistent with the terms you use.

*Response: Edited as "Recently undisturbed forests are ..."*

Discussion

P10L21-23 Awkward sentence please rephrase / breakdown into smaller parts.

*Response: This sentence was shortened as "Our method of inferring time since disturbance to estimate carbon flux and biomass accumulation relies on a number of data products and assumptions that need to be critically evaluated."*

P10L25-26 Odd place the begin new paragraph. You appear to be continuing your point from the first paragraph.

*Response: We would like to keep each assumption as one paragraph, we also added more discussion on the first assumption.*

P10l27 It is not clear what you mean. Are you talking about how the stand-level biomass estimate was calculated or how the real world stand was managed / grew?

*Response: We mean how the real world stand was accumulated to the current amount of biomass. The sentence was edited as "... how that stand-level biomass was actually achieved ..."*

P10L29 "...or also from a recent disturbance that reduced biomass to the current level." I think you need a reference here.

*Response: Reference "Xu et al., 2012" was added at the end of this sentence.*

P10L30 "...varies depending on the type of stand-replacing disturbance". Are you referring to e.g. clear felling vs fire?

*Response: Correct.*

P11L1-4 Currently you have not described the model used to provide required background for these statements.

*Response: This has now been added as noted above.*

P11L4 "...initial rise through stand initialization." Are you talking about early phases of forest growth? How long does initialization take?

*Response: Establishment of stands can of course take a variable amount of time depending on many factors. In the modeling (as described in our prior work) we assumed that NPP post disturbance rises to a steady rate over the course of 8 years post-disturbance, and that allocation of NPP to woody biomass also increases over that interval.*

P11L9 "...which are sure to have errors." Are there any estimates of this error?

*Response: We meant to say input maps (maps of biomass, forest type group, site productivity and forest disturbance) have errors. The accuracies of biomass and forest type group were assessed and provided by the data provider, however site productivity and forest disturbance not.*

P11L9-13 Should this not be first introduced in the methods section if these describe errors between field information and the maps you have used to constrain your model. Also, is there a bias associated with these errors? If so, how do you expect these biases to impact your analysis. Might a bias here impact the differing conclusions between here and your previous works?

*Response: We intend to discuss that there are errors associated with these input maps, and how we partly account for these uncertainties in this study. We developed more discussion on these, including:*

*(1) Adding a new discussion paragraph on Kellndorfer NBCD biomass products. It reads as "Second, we assume remote sensing-derived NBCD biomass products were well calibrated by field-derived biomass. However, the correlation coefficients between observed and predicted biomass were estimated to be 0.62-0.75 in the PNW region (Kellndorfer et al., 2012). And at 30 m pixel level, NBCD biomass values were biased with a large number of zero biomass values that had predictions in local biomass products (Huang et al., 2015). Discrepancies in biomass values between remote sensing- and field-derived data lead to biased stand age, as well as associated carbon stocks and fluxes. These were addressed in this study by imposing 20% error to pixel level biomass estimates and replacing zero biomass by the mean biomass of forest pixels with the same forest type and site productivity within this region."*

*(2) Adding more discussion on forest type group, it reads as "It was reported that accuracy of forest type group map in the PNW region ranges from 61% to 69% (Ruefenacht et al., 2008); besides, forest type groups for some pixels undefined from original data were assigned as the forest types of the nearest pixels. For the same biomass value, inferred stand age and estimated carbon fluxes can vary greatly given difference in forest type group (Fig. 4 & Fig. 6)."*

*(3) Developing more discussion on ADS data, and now it reads as "The ADS dataset is known to be limited by the areas flown in the survey years, and likely underestimate the number of trees killed by bark beetles but likely overestimate the area of affected stands (Meddens et al., 2012). These uncertainties*

*have important consequences for the carbon balance and flux estimates from bark beetle outbreaks, part of them were accounted for by adopting several approaches (Ghimire et al., 2015)."*

*We expect errors from input map will bring bias in stand age and NEP estimates, so we propagated part of these uncertainties into our estimates, and provided a distribution of estimates, including quantiles and standard deviation, instead of a single value.*

P11L27-28 Introducing new information in figures which are not described in the results. This should not be the case. If the figure and comparison is really needed then it should be included in the results and be part of the experimental design. or could be moved to SI.

*Response: We divided this paragraph into three parts. The sentence of introducing comparison was moved to "Materials and Methods", comparing results and Fig. 10 (now Fig. 9) was moved to "Results", and we keep the discussion on discrepancy from comparison in "Discussion" section.*

P12L12-15 New analysis should not be introduced in the discussion. Also Figure 11 did not appear in the results either. Again, if these comparison and figure is needed then make it part of the experimental design and introduce it in the results section first.

*Response: Similar to response to last comment, we divided this paragraph into three parts. The sentence of introducing comparison was moved to "Materials and Methods", comparing results and Fig. 11 (now Fig. 10) was moved to "Results", and we keep the discussion on discrepancy from comparison in "Discussion" section.*

P12L14-15 "...distribution agrees well with that for our undisturbed..." poor working rephrase.

*Response: This sentence was edited as "Overall, the pattern of FIA-derived age distribution matches well with that derived from our study, but with our study having consistently lower forest areas at age classes larger than 20."*

P12L18, P13L4,L18 Multiple definitions of what is a young forest. Can you reconcile these?

*Response: P12L18: We replaced "stand of young ages" by "stand with ages ranging from 0 to 24". P13L4: We deleted "young" from "young regenerating forests". P13L18: We replaced "relatively mature (>24 year old) forests" by "recently undisturbed forests".*

P12L22 "A portion of this difference can be attributed to smaller net carbon losses..." If I understand correctly here you mean greater loss / more negative? Comparing between -4 TgC and -7 TgC? Not clear.

*Response: Correct. This study reports more net carbon losses (-7 Tg C $y^{-1}$) than -4 Tg C $y^{-1}$ reported in our previous work (Williams et al., 2014). The sentence was edited as "A portion of this difference can be attributed to larger net carbon losses from forestlands (-7 Tg y-1 carbon loss in this study vs. -4 Tg y-1 carbon loss in Williams et al.) due to recent (1986 to 2010) disturbance by either harvest or fire."*

P12L31-33 Is the PNW region representative of forestry in the US?

*Response: We meant western US, not US at P12L31-33. Bark beetle outbreak in PNW is not representative of forests in western US, where rocky mountain north (RMN) and rocky mountain south (RMS) regions have higher mortality rates (37%, 35%), while a lower rate (1%) in Pacific Southwest (PSW) region (Ghimire et al. 2015). So we use PNW mortality percentage in western US to obtain NEP reduction due to bark beetle outbreak in PNW.*

P13L19 Good to see some comparison with other studies. Are there any more available to broaden the discussion?

*Response: Thank you. We have current discussions on NEP comparison.*

Figures

All of the figure captions need to be expanded to make clear where the data / analysis from each figure comes from and any key features. Also there appears to be substantial repetition of the disturbance figure. Can the figures be re-arrange to minimize this / move some of these maps to the SI.

*Response: We edited all the figure captions to include more detailed description, and deleted Fig. 1.*

*We provided two figures on forest disturbance, one is year of last disturbance map from NAFD, MTBS and ADS, and the other one is integrated disturbance map (disturbance year and type), which is one of the resulting figures and new contribution to current disturbance maps.*

Figure 5. These NEP do not show C loss, even though your analysis does. These figures reinforce the confusion between whether or not you are analysis the C balance of the ecosystem as a whole or just the live biomass. If you are analyzing the whole ecosystem the NEP would surely be negative directly after disturbance due to litter and soil C turnover?

*Response: We analyzed the whole ecosystem, and Fig. 5 (now Fig. 4) only displays positive NEP trajectories after disturbance. More texts were added in caption of Fig. 5 (now Fig. 4), "The typical pattern of NEP following a disturbance involves a large negative value immediately after disturbance, a rise for a number of years to reach a maximum rate of carbon uptake, and then a gradual decline. Only the positive part of NEP trajectories were displayed."*

Figure 10. In your analysis are "Years Since Disturbance" and "Stand Age" the same thing? If so why in the same figure are you referring to this by different names. Particularly as in the caption you refer to both as "Stand age".

*Response: In our study, for undisturbed or stand-replacing disturbed forests, "years since disturbance" and "stand age" are the same thing; while for partial disturbed forest, they are not. We used term "years since disturbance" in this paper, and Pan et al. (2011) used term "stand age", so we keep both to be consistent with original paper. We edited the figure caption of Fig. 10 (now Fig. 9).*

**References**

Ghimire, B., Williams, C.A., Collatz, G.J., and Vanderhoof, M.: Fire-induced carbon emissions and regrowth uptake in western US forests: documenting variation across forest types, fire severity, and climate regions, J. Geophys. Res.-Biogeo., 117, doi:10.1029/2011JG001935, 2012.

Ghimire, B., Williams, C.A., Collatz, G.J., Vanderhoof, M., Rogan, J., Kulakowski, D., and Masek, J.G.: Large carbon release legacy from bark beetle outbreaks across Western United States, Glob. Change Biol., 21(8), 3087-3101, doi: 10.1111/gcb.12933, 2015.

Huang, W., Swatantran, A., Johnson, K., Duncanson, L., Tang, H., O'Neil-Dunne, J., Hurtt, G., and Dubayah, R.: Local discrepancies in continental scale biomass maps: a case study over forested and non-forested landscapes in Maryland, USA, Carbon Balance and Management, 10:19, doi: 10.1186/s13021-015-0030-9, 2015.

Kellndorfer, J., Walker, W., Kirsch, K., Fiske, G., Bishop, J., LaPoint, L., Hoppus, M., and Westfall, J.: NACP Aboveground Biomass and Carbon Baseline Data, V. 2 (NBCD 2000), U.S.A., 2000. Dataset Available on-line [http://daac.ornl.gov] from ORNL DAAC, Oak Ridge, Tennessee, U.S.A. http://dx.doi.org/10.3334/ORNLDAAC/1161, 2013.

Meddens, A.J.H., Hicke, J.A., and Ferguson, C.A.: Spatial and temporal patterns of observed bark beetle-caused tree mortality in British Columbia and western US, Ecol. Appl., 22, 1876–1891, 2012.

Pan, Y., Chen, J.M., Birdsey, R.A., McCullough, K., He, L., and Deng, F.: Age Structure and Disturbance Legacy of North American Forests, Biogeosciences, 8, 715-732, 2011.

Ruefenacht, B., Finco, M.V., Nelson, M.D., Czaplewski, R., Helmer, E.H., Blackard, J. A., Holden, G.R., Lister, A.J., Salajanu, D., Weyermann, D., and Winterberger, K.: Conterminous U.S. and Alaska Forest Type Mapping Using Forest Inventory and Analysis Data. Photogramm, Eng. Rem. S., 74(11), 1379-1388, 2008.

Williams, C.A., Collatz, G.J., Masek, J., and Goward, S.N.: Carbon consequences of forest disturbance and recovery across the conterminous United States, Global Biogeochem. Cy., 26, doi:10.1029/2010GB003947, 2012.

Williams, C.A., Collatz, G.J., Masek, J., Huang, C., and Goward, S.N.: Impacts of disturbance history on forest carbon stocks and fluxes: merging satellite disturbance mapping with forest inventory data in a carbon cycle model framework, Remote Sens. Environ., 151, 57–71, 2014.

Xu, C., Turnbull, M.H., Tissue, D.T., Lewis, J.D., Carson, R., Schuster, W.S.F., Whitehead, D., Walcroft, A.S., Li, J., and Griffin, K.L.: Age-related decline of stand biomass accumulation is primarily due to mortality and not to reduction in NPP associated with individual tree physiology, tree growth or stand structure in a *Quercus*-dominated forest, J. Ecol., 100, 428-440, doi: 10.1111/j.1365-2745.2011.01933.x, 2012.

---

## Author Comment (AC2) · 6 Aug 2016

*We would like to thank the reviewer for giving us very helpful suggestions that help us greatly improve the quality of this manuscript. We provided our responses to all the comments point by point below (italicized typeface).*

This is a pretty good and potentially useful paper that could be published after some modifications. The main problems I can identify are (1) the introduction is poorly written in places, (2) an important and highly relevant citation is missing, and (3) the discussion needs more work.

*Response: (1) We rewrote most of the "Introduction", making the content proportional to the objectives. (2) The suggested citation was included in our introduction. (3) We expanded more in discussion.*

Detailed comments below address some problems with the introduction. The missing citation is more troubling since it presents an alternative approach to using the CASA model for estimating growth (or NEP) which is a center piece of this study. The citation is: Raymond, C. L., Healey, S., Peduzzi, A., Patterson, P. 2015. Representative regional models of post-disturbance forest carbon accumulation: Integrating inventory data and a growth and yield model. Forest Ecology and Management 336: 21-34. This should be referenced in a couple of places (p. 2 line 20 and p. 4 line 1).

*Response: Thanks for suggested citation, we have included it in our introduction.*

The discussion should compare using the CASA model and using the Raymond et al. approach which relies on an FIA driven empirical model, the Forest Vegetation Simulator (FVS). What are the advantages and disadvantages of each approach, and do they yield similar results (the regions are different but still may be able to compare results for one or two forest types). I also suggest that the discussion should explore in more depth the many assumptions and inferences that have to be made to estimate time since disturbance for "undisturbed" pixels (section 2.2.2). For example, the Kellndorfer biomass map used to estimate biomass of "undisturbed" pixels has fairly high uncertainty at the pixel level; some pixels were assigned forest types based on a nearby neighbor pixel, etc. By the way, the title of section 2.2.2 is an oxymoron – if the pixel is "undisturbed" there should not be a time since disturbance. So instead of "undisturbed" the authors should use a different term to identify pixels that had no detected disturbance since 1986, perhaps something like "recently undisturbed".

*Response: We didn't compare CASA and FVS models in discussion for two reasons: (1) the process of carbon cycle model is not the main focus in this paper, which has been described and discussed in our prior work (Williams et al., 2012, Ghimire et al., 2012, Ghimire et al., 2015). Our objective related to CASA is to use CASA-derived carbon stock and flux trajectories to do mapping; (2) it's not appropriate to compare carbon trajectories developed for different study area. Both CASA and FVS models heavily depend on FIA data, without same/similar FIA input, the comparison won't make a solid point. But when we moved to Rocky Mountain region in our future work, it sounds good to make such comparison.*

*We added a new discussion paragraph on Kellndorfer NBCD biomass products. It reads as "Second, we assume remote sensing-derived NBCD biomass products were well calibrated by field-derived biomass. However, the correlation coefficients between observed and predicted biomass were estimated to be 0.62-*

*0.75 in the PNW region (Kellndorfer et al., 2012). And at 30 m pixel level, NBCD biomass values were biased with a large number of zero biomass values that had predictions in local biomass products (Huang et al., 2015). Discrepancies in biomass values between remote sensing- and field-derived data lead to biased stand age, as well as associated carbon stocks and fluxes. These were addressed in this study by imposing 20% error to pixel level biomass estimates and replacing zero biomass by the mean biomass of forest pixels with the same forest type and site productivity within this region."*

*We also added more discussion on forest type group, "It was reported that accuracy of forest type group map in the PNW region ranges from 61% to 69% (Ruefenacht et al., 2008); besides, forest type groups for some pixels undefined from original data were assigned as the forest types of the nearest pixels. For the same biomass value, inferred stand age and estimated carbon fluxes can vary greatly given difference in forest type group (Fig. 4 & Fig. 6)."*

*The title of section 2.2.2 was edited as "Time since disturbance for recently undisturbed forest pixels".*

Specific comments

The title is too long and redundant. Suggest "High-resolution forest carbon flux mapping in the Pacific Northwest with disturbance legacies inferred from remote sensing and inventory data". Could also leave out "in the Pacific Northwest".

**Response:** *We apologize that our title didn't fully convey main objectives and focuses of this manuscript, we edited the title as "High-resolution mapping of time since disturbance and forest carbon flux from remote sensing and inventory data-inferred disturbance legacies in the Pacific Northwest".*

p. 1 line 22: delete the second "probabilistic,"

**Response:** *Second "probabilistic" was deleted as suggested.*

p. 1 line 26: re-word so that it does not appear that tracts of land can somehow "see".

**Response:** *"seen" was deleted from the sentence.*

p. 2 line 13: replace "is itself a sort of record of" with "reflects"

**Response:** *Replaced as suggested.*

p. 2 line 14: replace "general" with "predictable rate of"

**Response:** *Replaced as suggested.*

p. 2 line 22-23: needs some rewording. The idea is that it is important to include smallscale disturbances down to some minimum threshold, not that disturbances typically are at this small scale.

**Response:** *The sentence was edited as "characterization of time since disturbance across landscapes at a scale of being able to detect small-scale disturbance events, typically around 100 m or less."*

p. 2 lines 27-28: add "at smaller scales" to the end since national forests inventories can provide useful guidance only at larger scales. But importantly note, it is possible to conduct field inventories at very small scales, so the statement is not very correct at all, only partially correct with respect to national forest inventories.

**Response:** *"at smaller scales" was added at the end of this sentence.*

p. 3 lines 11-12: One objective is clearly stated. What are the others? The last sentence of this paragraph seems to be another objective, but then, I'm confused as to whether the purpose is to develop a method for large-scale monitoring and management, or small-scale, or both?

*Response: We rewrote the last paragraph in Introduction, all the objectives were listed. Now it reads as "This study estimates and maps time since disturbance at a fine scale of 30 m from RS-derived products and FIA-derived biomass growth curves, and then maps net ecosystem productivity (NEP) based on disturbance history, time since disturbance and carbon flux legacy. The specific objectives in this study are to: (1) introduce a method for inferring a pixel's representative time since disturbance from RS-derived biomass and disturbance products at the 30 m resolution; (2) map NEP based on model-derived carbon stock and flux trajectories that describe how NEP changes with time following harvest, fire, or bark beetle disturbances of varying severity; (3) propagate uncertainties from RS-derived biomass products and FIA into uncertainty quantification of stand age and NEP. Our research represents an approach to map carbon stocks and fluxes at a high resolution across the conterminous US in support of national carbon monitoring, reporting, and management."*

*The last sentence of this paragraph is misleading, we have edited it as shown above. Our method will be used to map carbon stock and fluxes at a fine scale in the conterminous US.*

p. 3 lines 27-28. Terminology again – "undisturbed" pixels by definition should not have a time since disturbance.

*Response: "undisturbed forest pixels" has been edited as "recently undisturbed forest pixels".*

p. 3 line 32: Biomass curves were developed by forest type group and productivity class. How were these 2 classes allocated to the 5 NEP classes described on p. 4 lines 2-4?

*Response: Biomass curves by forest type group and productivity class were used to infer stand age for forest pixels undisturbed during remote sensing observation period.*

*NEP curves were derived from CASA carbon cycle process model with inclusion of disturbance processes. Biomass curves were used to adjust model's rates of NPP and wood turnover for each forest type group and productivity class. At the final stage of the modeling, the disturbance processes imposed stand-replacing harvest, fire or insect-induced partial disturbance to generate carbon stock and flux curves as a function of time since disturbance, and are specific to forest type group, site productivity class, disturbance type and disturbance severity.*

p. 3 line 35: add citation after ". . .varying severity".

*Response: Citations "Williams et al. 2012, Ghimire et al., 2012, Ghimire et al., 2015" were added after "... varying severity".*

p. 6 line 26: replace "stand" with "standard"

*Response: Replaced as suggested.*

p. 8 line 21: sentence that begins with "Again" needs editing.

*Response: This sentence was edited as "Again in contrast, bark beetle outbreak areas for low and high productivity classes are similar in Douglas-fir forests, but beetle outbreak occurrence was about three times more likely in low productivity sites."*

p. 9 lines 24-25: the imprint is not so clear to me. Maybe need to highlight somehow on the graphic.

*Response: We included the position description of Biscuit fire in the bracket, "bottom left of Fig. 8a, also refer to bottom left of Fig. 5a & 5b".*

p. 10 lines 12-14: One could argue that inventory data does not provide such a reliable estimate of biomass/age. Both of these variables can be rather difficult to measure/estimate especially with respect to the selection of biomass equations, but also the difficulty of assigning a stand age to stands that are uneven-aged.

*Response: We added more discussion on the first assumption as suggested. "However, both stand age and biomass are difficult to measure and estimate, especially considering the difficulty of assigning a stand age to uneven-aged forest stands, as well as selecting appropriate species-specific biomass equations (Parresol, 1999). If FIA ages are older than actual stand ages, the associated forest biomass will be underestimated, and stand age inferred from biomass products will be overestimated. And younger FIA ages than actual ages will result in an overestimation in biomass accumulation, but an underestimation in biomass-inferred stand ages. Though a possible bias in stand ages, our estimates of carbon stocks and fluxes are not likely to be largely adjusted by a stand age bias within 5 years (Williams et al., 2012)."*

p. 12 lines 1-7: not stated TFIA does not do a good job of detecting recent dis- ˘ turbances because the remeasurement cycle in the PNW is about 10 years, so the average time lag of the data at any point in time is at least 5 years.

*Response: We added the suggested point in this paragraph, now it reads as "FIA data miss some recent disturbances, partly because FIA remeasurement cycle in the PNW region is about 10 years, with the average time lag of the data being around 5 years."*

**References**

Ghimire, B., Williams, C.A., Collatz, G.J., and Vanderhoof, M.: Fire-induced carbon emissions and regrowth uptake in western US forests: documenting variation across forest types, fire severity, and climate regions, J. Geophys. Res.-Biogeo., 117, doi:10.1029/2011JG001935, 2012.

Ghimire, B., Williams, C.A., Collatz, G.J., Vanderhoof, M., Rogan, J., Kulakowski, D., and Masek, J.G.: Large carbon release legacy from bark beetle outbreaks across Western United States, Glob. Change Biol., 21(8), 3087-3101, doi: 10.1111/gcb.12933, 2015.

Huang, W., Swatantran, A., Johnson, K., Duncanson, L., Tang, H., O'Neil-Dunne, J., Hurtt, G., and Dubayah, R.: Local discrepancies in continental scale biomass maps: a case study over forested and non-forested landscapes in Maryland, USA, Carbon Balance and Management, 10:19, doi: 10.1186/s13021-015-0030-9, 2015.

Kellndorfer, J., Walker, W., Kirsch, K., Fiske, G., Bishop, J., LaPoint, L., Hoppus, M., and Westfall, J.: NACP Aboveground Biomass and Carbon Baseline Data, V. 2 (NBCD 2000), U.S.A., 2000. Dataset Available on-line [http://daac.ornl.gov] from ORNL DAAC, Oak Ridge, Tennessee, U.S.A. http://dx.doi.org/10.3334/ORNLDAAC/1161, 2013.

Parresol, B.R.: Assessing tree and stand biomass: A review with examples and critical comparisons, Forest Sci., 45(4), 573-593, 1999.

Raymond, C.L., Healey, S., Peduzzi, A., and Patterson, P.: Representative regional models of post-disturbance forest carbon accumulation: Integrating inventory data and a growth and yield model. Forest Ecol. Manag., 336, 21-34, 2015.

Ruefenacht, B., Finco, M.V., Nelson, M.D., Czaplewski, R., Helmer, E.H., Blackard, J. A., Holden, G.R., Lister, A.J., Salajanu, D., Weyermann, D., and Winterberger, K.: Conterminous U.S. and Alaska Forest Type Mapping Using Forest Inventory and Analysis Data. Photogramm, Eng. Rem. S., 74(11), 1379-1388, 2008.

Williams, C.A., Collatz, G.J., Masek, J., and Goward, S.N.: Carbon consequences of forest disturbance and recovery across the conterminous United States, Global Biogeochem. Cy., 26, doi:10.1029/2010GB003947, 2012.

---

## Author Response (AR1)

**Summary of changes:**

We have extensively revised the manuscript to address the concerns of the editor and two reviewers, including rewriting, editing and adding more text in several sections based on the reviews. We also clarified a number of confusing sentences throughout the manuscript, updated tables, figures, references and the supplementary document, and performed an overall grammar check. Please refer to "response to reviewers" for detailed information. Here are the **major** revisions we have made in the manuscript:

- 1. We edited the title as "High-resolution mapping of time since disturbance and forest carbon flux from remote sensing and inventory data on harvest, fire, and beetle disturbance legacies in the Pacific Northwest". Quantification and mapping of time since disturbance is an important objective in this paper, while carbon flux is meant to be a relatively minor focus.
- 2. We rewrote many paragraphs in the "Introduction", particularly the last paragraph where all of the objectives are listed. Now it reads as "This study estimates and maps time since disturbance at a fine scale of 30 m from RS-derived products and FIA-derived biomass growth curves, and then maps net ecosystem productivity (NEP) based on disturbance history, time since disturbance and carbon flux legacy. The specific objectives in this study are to: (1) introduce a method for inferring a pixel's representative time since disturbance from RS-derived biomass and disturbance products at the 30 m resolution; (2) map NEP based on pre-existing, model-derived carbon stock and flux trajectories that describe how NEP changes with time following harvest, fire, or bark beetle disturbances of varying severity; (3) propagate uncertainties from RS-derived biomass products and FIA into uncertainty quantification of stand age and NEP. Our research represents an approach to map carbon stocks and fluxes at a high resolution across the conterminous US in support of national carbon monitoring, reporting, and management."
- 3. We have added more descriptions on the process of CASA model in section 2.3.1. Since the processes of the CASA model used to derive carbon flux trajectories were described in detail in our prior papers (Williams et al., 2012, Ghimire et al., 2012, Ghimire et al., 2015), here we mainly focused on the approach of combining these trajectories with the newly mapped time since disturbance derived from objective 1 to simply demonstrate that this method can be used to develop spatial representations of NEP.
- 4. We have added more discussion of assumptions made in this study in section 4.1. We present a mature characterization and propagation of errors from biomass and disturbance year input data sources, and formally assess how these can bias stand age and NEP estimates. These major error sources are fully propagated as part of the uncertainties that we provide as a distribution of estimates, including quantiles and standard deviations, instead of a single value. However, we are unable to assess potential biases from some of the input datasets/parameters due to limited/no information on those errors.
We would like to thank the reviewer for giving us very helpful suggestions that help us greatly improve the quality of this manuscript. We provided our responses to all the comments point by point below (italicized typeface) with page (P) and line (L) numbers referred if necessary.

**General comments**

The manuscript aims to address current issues in constraining forest C dynamics and stocks in relation to multiple different types and intensity of disturbance. The authors combine a range of data including national inventories, management databases, airborne and space-borne remote sensing. These data are then combined / utilized through both statistical (yield curves) and simulation based modelling (CASA) approaches. As such the manuscript is highly relevant and well within scope of Biogeosciences and I believe will ultimately be published in Biogeosciences. However I believe there is additional scientific value that should be drawn from the current analysis and a substantial re-write to improve readability prior to publication. The following general comments are split broadly between scientific and presentation.

Globally C stored in forests is split roughly equally between woody biomass and soil organic matter (e.g. Pan et al 2011). However the manuscript focuses on estimates of above ground biomass stocks and disturbance to these stocks, lacking any analysis or discussion of soil carbon stocks. I recognize that the authors report net ecosystem productivity (defined as NEP = NPP-Rh), but I would prefer you to distinguish between accumulation and losses between the live biomass and dead organic matter. Or state clearly why not given that you are reporting ecosystem scale values. Also I do not believe that the authors have extracted all relevant information for the above ground biomass stocks. For example in Section 3.4 L29 The authors state "Spatial variations in mean annual NEP are noticeably correlated with the time since disturbance, forest type group, and site productivity strata...". This could be shown more clearly in an  $x \sim y$  plot and / or this "noticeable" correlation could be explicitly quantified to distinguish the relative importance of the drivers. Further detail follows in the specific comments section.

**Response:** We apologize that our title didn't fully convey main objectives and focuses of this manuscript, we edited the title as "High-resolution mapping of time since disturbance and forest carbon flux from remote sensing and inventory data on harvest, fire, and beetle disturbance legacies in the Pacific Northwest". Quantification and mapping of time since disturbance is an important objective in this paper, while carbon flux is meant to be a relatively minor focus.

We rewrote the last paragraph in Introduction at lines P3L23 to P3L31, where now all of the objectives are listed. Aboveground biomass accumulation curves were used for objective 1 to infer stand age from RS-derived biomass data. Carbon flux curves were used in objective 2, which considers carbon accumulation and loss, also live biomass and dead organic matter. Section 2.3.1 in the new version at lines P7L21 to P8L2 provides a revised and expanded description of the model, including its approach to dead organic carbon cycling and also its inclusion as part of disturbance processes. Since the processes of the base CASA model (Randerson et al. 1996), and our specific use to derive post-disturbance carbon flux trajectories, were all described in detail in prior papers (Williams et al., 2012, Ghimire et al., 2012,

Ghimire et al., 2015), here we mainly focused on the approach of combining these trajectories with the newly mapped time since disturbance derived from objective 1 to simply demonstrate that this method can be used to develop spatial representations of NEP.

For section 3.4 L29 (now P10L31 to P10L32), we estimated NEP based on carbon flux trajectories, which vary by time since disturbance, forest type group and site productivity. The relationships between NEP and these variables have been presented in the trajectory curves in Fig. 4. Because these trajectories are directly applied to map NEP, there is limited new information, if any, in the fact that NEP is correlated with time since disturbance, forest type group, and site productivity strata. Therefore, we think  $x \sim y$  plot is unnecessary. Furthermore, our intention is not to explore relationships between NEP and input data. Correspondingly, we revised the sentence in question to read "Spatial variations in mean annual NEP are determined by differences in strata of ..."

The overall writing style of the manuscript needs to improved to benefit the flow of reading and in particular clarity. For example the methods overview needs to be clearer as to the overall structure of the analysis and their connections. The authors provide extensive detail on the data sources and what information they provide, however the fact that these data will used as constraints or drivers in the CASA model is not made clear until the final section before the results in Section 2.3.1. This is particularly confusing as all the description of data given is in reference to above ground biomass while at the same time stating that the results from the analysis are the net ecosystem productivity. Moreover the number of words in both the methods and results sections dedicated to the various disturbance maps produced appears disproportionate given that the title and the conclusions imply that C stocks and dynamics are the primary focus. I would consider way to simplify this information and attempt to move some of it into the supplementary material. Also I note that Figures 10 and 11 do not feature in the results section at all, instead are used to introduce new information in the discussion which is inappropriate. These figures should be introduced in the results section of they could be moved to the supporting information.

**Response:** Again, we apologize that our title didn't fully convey main objectives and focuses of this manuscript, we edited the title as "High-resolution mapping of time since disturbance and forest carbon flux from remote sensing and inventory data on harvest, fire, and beetle disturbance legacies in the Pacific Northwest". We also rewrote the last paragraph in Introduction at lines P3L23 to P3L31, where now all of the objectives are listed. Quantification and mapping of time since disturbance is an important objective in this paper, and while it is being generated for the purpose of carbon flux, the characterization of carbon fluxes was the focus of the group's prior work and is thus of less emphasis here. We did revise the present paper to include a more comprehensive overview of the carbon modeling performed in our prior work.

The purpose behind why we introduce a number of data sources is not to describe how they are used as constraints or drivers of the CASA modelling, but rather to introduce the methodology of how we make use of those currently available data sources to infer a pixel's representative time since disturbance, which is one of the main objectives in this paper. Since all the input data sources can be used directly except for a disturbance map with disturbance type/severity and biomass~age growth curves, so we provided explicit description of how to derive these two important components, and the resulting maps, figures and numbers. Another objective is to make use of the time since disturbance map that we derived for objective 1 with carbon flux trajectory curves from our previous work to map NEP. Since the processes of the CASA model and the methods for deriving carbon flux trajectories were each described in detail in our prior papers (Williams et al., 2012, Ghimire et al., 2012, Ghimire et al., 2015), here we emphasize only the approach of applying these trajectories and resulting NEP maps.

For Fig. 10 and Fig. 11, we divided the two relevant paragraphs in "Discussion" into three parts. The sentence introducing the comparison between the results from this study and those of previous work was moved to "Materials and Methods" and the comparison of those results (formerly Fig. 10 and Fig. 11, now Fig. 9 and Fig. 10) were moved to "Results". We keep associated discussion about likely sources of the discrepancy in the "Discussion" section.

**Specific comments**

The following comments are broken down into Abstract, Introduction, Methods, Results, Discussion and Conclusion sections. General comments on each section will be followed by specific comments with page (P) and line (L) numbers.

Abstract The abstract could be made to flow more easily and make clear that the analysis is feeding into a C-cycling model that represents both the live and dead carbon pools. This will reduce confusion between described / yield curves used which constrain above ground biomass while at the same time reporting a net ecosystem value.

**Response:** We edited the abstract to reflect objectives of this manuscript. We clarified biomass growth curves and carbon flux trajectories were used for different objectives. The CASA carbon cycle model was not mentioned in the abstract, since this is not the focus of this paper.

Introduction All relevant information appears to be present in the introduction, however not all of the information is clear. I would recommend the use of topic-sentences to improve clarity of your message for each paragraph. Moreover there are a number of sentences where the wording is awkward to read.

**Response:** We rewrote most of the "Introduction", making the objectives much clearer and the content proportional to the objectives.

P2L29 "...remote sensing techniques..." should include "...remote sensing (RS) techniques" as RS is used later.

Response: Acronym of remote sensing has been included in brackets at line P2L29.

P2L30 "...remote sensing techniques provide..."

**Response:** "... are providing ..." has been changed to "... generate ..." at line P2L30 to avoid repeat use of the word "provide" in the same sentence.

P2L32 "Such products miss small scale events and extend only so far back in time..." awkward wording. Please reconsider e.g. "However, RS products frequency miss small scale events and only cover the last several decades..."

**Response:** This sentence was edited as "However, such disturbance products only record events that occurred within the last several decades, ..." at lines P2L32 to P2L33.

P3L8 "...provide a way forward to capture at least some of the information that is missing but needed..." awkward wording, Please reconsider rewording.

**Response:** This sentence was edited as "Nonetheless, RS-derived forest biomass still provides a valuable way of characterizing the pixel-scale (e.g. 30m or 250 m) legacy effects of disturbance that occurred prior to RS observations, which is required for quantifying carbon stock recovery and carbon uptake and release rates over large areas." at lines P3L9 to P3L11.

P3L9 - L20 The final paragraph would be a good place to make a clear statement of the studies objective (key questions) and novelty. However the final paragraph here mixes further introduction and aims. This could be split and made clearer.

**Response:** We rewrote the last paragraph in Introduction at lines P3L23 to P3L31, where now all of the objectives are listed. Now it reads as "This study estimates and maps time since disturbance at a fine scale of 30 m from RS-derived products and FIA-derived biomass growth curves, and then maps net ecosystem productivity (NEP) based on disturbance history, time since disturbance and carbon flux legacy. The specific objectives in this study are to: (1) introduce a method for inferring a pixel's representative time since disturbance from RS-derived biomass and disturbance products at the 30 m resolution; (2) map NEP based on pre-existing, model-derived carbon stock and flux trajectories that describe how NEP changes with time following harvest, fire, or bark beetle disturbances of varying severity; (3) propagate uncertainties from RS-derived biomass products and FIA into uncertainty quantification of stand age and NEP. Our research represents an approach to map carbon stocks and fluxes at a high resolution across the conterminous US in support of national carbon monitoring, reporting, and management."

Methods The methods are very long (which I accept may be required) and would benefit from an improved overview section. Where possible the methods sections would benefit from moving some material to the supplementary material to improve focus.

**Response:** We made several edits in "2 Materials and Methods", especially making the "2.1 Overview" more clarified.

P3L25 "...recent disturbance..." how recent?

**Response:** "...recent disturbance..." at line P4L6 refers to disturbance since the starting year of disturbance product as noted in the manuscript.

P3L28 "...terms "time since disturbance" and "stand age" " would it be possible to pick one of these terms and use it consistently?

**Response:** We prefer to use term "time since disturbance" based on context of this paper, while in FIA and other papers, such as Pan et al. (2011), "stand age" was mainly used for undisturbed forests. To make the connection between two terms, here we wrote: terms "time since disturbance" and "stand age" are used interchangeably for recently undisturbed forest pixels thereafter at lines P4L9 to P4L10.

P3L29 "It was inferred..." possibly "Stand age was inferred..." would be clearer?

Response: The sentence was edited as "Stand age was inferred ..." at line P4L10.

P3L30 "The (yield?) curves were sampled from FIA data and specific to forest type and group and site productivity class." Is this information known in all cases? If not, what is assumed in their place?

**Response:** Two out of 16 forest type groups did not have FIA-derived biomass-age curves available, they are "Pinyon/Juniper" and "California Mixed Conifer", so we used curves of "Other Western Softwood" instead.

P4L1 "Net ecosystem productivity (NEP)" prior to this point all data / methods mentioned implies that this study is focusing on above ground biomass only. A link to CASA needs to be made earlier to make this clear.

**Response:** Two main objectives in this paper are to map time since disturbance and NEP. In section "2.1 Overview", we spent the first paragraph on methods to infer time since disturbance, and the second paragraph on mapping NEP. So we think it's appropriate to mention NEP at the beginning of the second paragraph. Besides, CASA model was mentioned here at lines P4L14 to P4L15 now.

P4L13 It would be useful to have a table with the different data sources listed and state the data and time period they cover.

**Response:** We added a new table (Table 1, shown below) in the manuscript to summarize the data sources.

| Data               | Description                 | Source             | Year      | Input for recently disturbed |
|--------------------|-----------------------------|--------------------|-----------|------------------------------|
|                    | 1                           |                    |           | or/and undisturbed forests   |
|                    |                             |                    |           | 01/alia ullaistuibea loiests |
| NAFD               | Forest disturbance          | Landsat            | 1986-2010 | a, b                         |
| MTBS               | Burned area and severity    | Landsat            | 1986-2010 | a                            |
| ADS                | Area of insect outbreak and | Aerial survey      | 1997-2010 | а                            |
|                    | number of trees killed      | ·                  |           |                              |
| NBCD               | Aboveground live biomass    | Landsat, SRTM, FIA | 2000      | b                            |
| Forest Type Group  | Forest type group           | MODIS, NLCD, etc.  | 2001      | b                            |
| Site Productivity  | Fraction of high            | FIA                | 1984-2014 | b                            |
|                    | productivity                |                    |           |                              |
| Biomass-age Curves | Biomass accumulation as a   | FIA                | 1984-2010 | b                            |
| -                  | function of stand age       |                    |           |                              |

Table 1. Data sources for inferring time since disturbance for recently disturbed and undisturbed forest pixels.

a Data is one of the inputs for inferring time since disturbance for recently disturbed forest pixels.

b Data is one of the inputs for inferring time since disturbance for recently undisturbed forest pixels.

P5L1 "These assumptions...been reported in the literature." Long sentence, can you break some of the sentences with lists, multiple concepts or conditions down.

**Response:** This sentence has been broken into a short sentence with three lists at lines P5L15 to P5L18. It reads as "These assumptions are based on the rationales: (1) MTBS records most of the notable fire events in the region, (2) harvest events are one of the most ubiquitous stand replacing disturbance types active in the region, (3) ADS-mapped polygons of bark beetle infestations often include unaffected stands as has been reported in the literature".

P5L5-10 Consider making this list in a table

Response: We keep four rules in lists.

P5L11-12 "The target year...was 2010". Possibly make this point earlier say in the overview or introduction aims?

Response: Target mapping year of 2010 was now mentioned in "2.1 Overview" at line P4L14.

P5L23 "age class from ..." how many age classes, are all equal in size?

*Response:* There are 11 age classes in total: 0-20, 20-40, 40-60, 60-80, 80-100, 100-120, 120-140, 140-160, 160-180, 180-200, 200+. This information has now been included in Table S1.

P5L25 All other units are given as SI. Please do so here too. Also it is odd that up until now forest biomass as been discussed, here you have swapped into volume. Can you convert or is there a reason for this?

**Response:** The unit "cubic feet/acre/year" here is used by FIA and is the output unit from FIA EVALIDator. We didn't convert it to SI, since readers will know how we combine 7 site productivity classes (0-19, 20-49, 50-84, 85-119, 120-164, 165-224, 224+ cubic feet/acre/year) into two classes.

P6L1 "Differences in forest masks..." which forest masks? Which products you are using?

**Response:** Different forest masks were used in NAFD disturbance products and NBCD biomass products. This sentence was edited as "Differences in forest masks between NAFD disturbance and NBCD biomass products led to" at line P6L14. In our study, NAFD-based forest mask was used for the analyses and mapping.

P6L2 "These were replaced by the mean biomass of other undisturbed pixels..." The distributions of stand age in Figure 11 are not Gaussian, would the median be better or is there little difference?

**Response:** This is a great point. We do compare mean and median biomass of other undisturbed pixels with the same forest type and site productivity class (10 classes here from 0 to 1 with equal interval), and we found there is little difference between mean and median values.

P6L12-13 Again SI units please.

**Response:** Same reason as previous comment on SI. That is, we would like to keep the unit consistent with outputs from FIA EVALIDator, so readers would know how we combine 7 site productivity classes (0-19, 20-49, 50-84, 85-119, 120-164, 165-224, 224+ cubic feet/acre/year) into two classes.

P6L24 "In reality fhigh is almost always between 0 and 1." Can you say what the mean value is or distributional information? Something more informative.

**Response:** maximal  $f_{high} = 0.996$ , minimal  $f_{high} = 0.015$  and mean  $f_{high} = 0.530$  in PNW. We included maximal  $f_{high}$  and minimal  $f_{high}$  values in the manuscript at lines P7L4 to P7L5.

P7L4 This is the first mention of the CASA model. Please provide a brief description of the mode and how it works. This is needed given that you make reference to its process representation in the discussion P11L1-5. Also what is the model time step used. Over what period is CASA simulating these forests (prior to 2010)? which meteorological drivers are used (e.g. ERA-Interim, GFS)? How realistic are the spin up pool sizes relative to field estimates in undisturbed pixels. Your estimate of C loss in response to disturbance will partially dependent on soil losses which will also be dependent on their initial magnitude after spin up (e.g. Exbrayat et al., 2014). If this information is available in the cited literature please make this clear.

**Response:** We now include the following description at lines P7L21 to P8L2: "The CASA model used here is based on Randerson et al. (1996) and operates on a monthly time step. It uses a light use efficiency approach to simulating net primary productivity (NPP) based on RS-derived absorption of photosynthetically active radiation, biome parameters, and climate data. The model then allocates NPP to three live carbon pools (leaves, roots, and wood), and transfers carbon to dead pools (litter and soils) based on biome-specific rates of tissue turnover. Carbon in dead organic matter pools is transferred between pools, of which there are 10, depending on the rate and efficiency of heterotrophic consumption which varies between pools in the model and also depends on biome- and pool-specific chemistry and site-specific climate setting. The default model parameters that influence NPP and wood turnover (mortality and shedding), and hence accumulation of live biomass, were adjusted by forest type based on fits to yield data from the forest inventory and analysis dataset. Aboveground live biomass per unit area versus stand age was sampled from the forest inventory and analysis data for individual forest type and

site productivity class strata. The disturbances imposed in our version of the CASA model included standreplacing harvest, fire and bark beetle outbreak (Williams et al., 2012, Ghimire et al., 2012, Ghimire et al., 2015). Disturbance processes were imposed at the final stage of the modelling after a spin-up to equilibrium carbon pools followed by a prior disturbance with ensuing regrowth to a set pre-disturbance ages." Meteorological and satellite based drivers, soil type, biome (forest) type, and other parameter and driver datasets are all described in the original papers from which the corresponding results were directly taken.

P7L10 "...curves describing carbon fluxes and stocks..." which stocks / fluxes where are they?

**Response:** They include aboveground forest stock, NPP, heterotrophic respiration and NEP. This sentence was edited as "...curves describing aboveground forest stock, NPP, Rh and NEP with time since disturbance ..." at lines P8L4 to P8L5. In fact, we have estimates of carbon stocks for all live and dead pools in the model (about 10 more) but here we restrict our use to those mentioned.

All these curves were derived and shown in our prior work (Williams et al., 2012, Ghimire et al., 2012, Ghimire et al., 2015), citations were added at the end of this sentences.

P7L15 "This study emphasized the use of NEP curves. Fig. 5 …" Figure 5 seems to show that C losses do no occur whereas losses do occur in the results (Table 2) as presumably soil and litter C is being decomposed and undergoing mineralisation. So where is the C source represented?

**Response:** Fig. 5 (now Fig. 4) only displays positive NEP trajectories after disturbance to support a scale that enhances the ability to compare curves for different forest types and productivity classes. More texts were added in caption of Fig. 5 (now Fig. 4), "The typical pattern of NEP following a disturbance involves a large negative value immediately after disturbance, a rise for a number of years to reach a maximum rate of carbon uptake, and then a gradual decline. Only the positive part of NEP trajectories were displayed, the full range of post-disturbance NEP curves across forest types, productivity classes, and disturbance types are presented in the supplementary figures (Fig. S1, S2, S3, S4)."

**Results**

What is the primary focus of the manuscript? A large part of the results section is taken up with a description of the various input maps into the analysis. Much of it seems like it should be in the methods sections as a description of the inputs or could be moved to the supporting information. Unless these are actually new numbers derived from the combination of multiple maps. At the moment it is not clear. Possibly an overview could be given to the results as it takes a lot of reading before you get to any information on the estimates of biomass stocks.

**Response:** We included four sections in "Results", the first is about a disturbance map, and the second is on biomass~age growth curves. We provided the resulting maps, figures and numbers of a disturbance map with disturbance type/severity and biomass growth curves, because all the input data for inferring time since disturbance can be readily used except for these two important components, besides these results are new. The third and fourth sections are maps of time since disturbance and NEP, which are the main objectives of this paper. We hope that the substantial revisions we have made clarify the corresponding focus of this manuscript.

P9L11-12 "...these curves yielded a smoothed fit to the inventory data rather than showing a saw-toothed increase with stand age." Here are you referring to saw-toothed due to managed thinning or stem mortality events?

**Response:** FIA samples were compiled from FIA plot measurements in Oregon and Washington. Managed thinning or stem mortality events could be part of reasons, but there are also some other possible reasons causing the erratic and fluctuating jumps, such as vagueries of plot-to-plot variability that span climate, soils, topographic and other variations across sampling plots.

P9L23 "Uncertainty on the time since disturbance forest pixels is not currently available from disturbance products and this was not mapped" Could the uncertainty in the yield curves on growth since disturbance be included? How strongly do the yield curves constrain CASA?

**Response:** No, it would not be logically sound or feasible to use uncertainty in the yield curves to characterize uncertainty in the time since disturbance mapped from remote sensing (Landsat spectral reflectances).

The yield curves provide very important adjustments to the default NPP and wood turnover rates in the biome-scale parameters in the CASA model. Without this adjustment, the CASA model, which is designed for global scale applications, would not provide an accurate and fitting representation of forest biomass for the fine-scale and diverse settings of the US where we are applying the model.

P9L29 "Spatial variations in mean annual NEP are noticeably correlated with..." Why not actually correlate them to quantify this? A new  $x \sim y$  figure might be useful here too.

**Response:** We estimated NEP based on carbon flux trajectories, which vary by time since disturbance, forest type group and site productivity. The relationships between NEP and these variables have been presented by trajectory curves in Fig. 4, and are directly applied in the mapping exercise so there is no new information derived in the mapping aside from the spatial allocation. The patterns are indeed interesting and can be important for some applications but an x-y plot would not be particularly instructive (it would simply recover the trajectories applied in the mapping). Besides, we didn't mean to explore relationships between NEP and input data. So we edited this sentence as "Spatial variations in mean annual NEP are determined by differences in strata of ..." at line P10L31.

P9L30-31 "...weaker carbon sinks in the eastern, drier portion of the study area..." Again this could be show in an  $x \sim y$  plotting soil moisture / precipitation against C sink strength to quantify.

**Response:** Here we meant to describe a spatial pattern in NEP map, which has lower NEP values in the east side of the study area. We mentioned "drier" just because the eastern part of study area is less humid from our knowledge. We do not intend to emphasize the relationship between NEP and soil moisture/precipitation. We removed "drier" from the text at line P11L1.

P10L6 "Forestlands free of recent disturbance..." could be "Undisturbed forests are..." just trying to be consistent with the terms you use.

Response: Edited as "Recently undisturbed forests are ..." at line P11L8.

Discussion

P10L21-23 Awkward sentence please rephrase / breakdown into smaller parts.

**Response:** This sentence was shortened as "Our method of inferring time since disturbance to estimate carbon flux and biomass accumulation relies on a number of data products and assumptions that need to be critically evaluated." at lines P11L24 to P11L25.

P10L25-26 Odd place the begin new paragraph. You appear to be continuing your point from the first paragraph.

**Response:** We would like to keep each assumption as one paragraph, we also added more discussion on the first assumption at lines P11L27 to P12L2.

P10127 It is not clear what you mean. Are you talking about how the stand-level biomass estimate was calculated or how the real world stand was managed / grew?

**Response:** We mean how the real world stand was accumulated to the current amount of biomass. The sentence was edited as "... how that stand-level biomass was actually achieved ..." at line P12L11.

P10L29 "...or also from a recent disturbance that reduced biomass to the current level." I think you need a reference here.

Response: Reference "Xu et al., 2012" was added at the end of this sentence at line P12L14.

P10L30 "...varies depending on the type of stand-replacing disturbance". Are you referring to e.g. clear felling vs fire?

**Response: Correct.**

P11L1-4 Currently you have not described the model used to provide required background for these statements.

Response: This has now been added as noted above (responses to P7L4 comment above).

P11L4 "...initial rise through stand initialization." Are you talking about early phases of forest growth? How long does initialization take?

**Response:** Establishment of stands can of course take a variable amount of time depending on many factors. In the modeling (as described in our prior work) we assumed that NPP post disturbance rises to a steady rate over the course of 8 years post-disturbance, and that allocation of NPP to woody biomass also increases over that interval.

P11L9 "...which are sure to have errors." Are there any estimates of this error?

**Response:** We meant to say input maps (maps of biomass, forest type group, site productivity and forest disturbance) have errors. The accuracies of biomass and forest type group were assessed and provided by the data provider, however site productivity and forest disturbance not.

P11L9-13 Should this not be first introduced in the methods section if these describe errors between field information and the maps you have used to constrain your model. Also, is there a bias associated with these errors? If so, how do you expect these biases to impact your analysis. Might a bias here impact the differing conclusions between here and your previous works?

**Response:** We intend to discuss that there are errors associated with these input maps, and how we partly account for these uncertainties in this study. We developed more discussion on these, including:

(1) Adding a new discussion paragraph on Kellndorfer NBCD biomass products. It reads as "Second, we assume remote sensing-derived NBCD biomass products were well calibrated by field-derived biomass. However, the correlation coefficients between observed and predicted biomass were estimated to be 0.62-0.75 in the PNW region (Kellndorfer et al., 2012). And at 30 m pixel level, NBCD biomass values were biased with a large number of zero biomass values that had predictions in local biomass products (Huang

et al., 2015). Discrepancies in biomass values between remote sensing- and field-derived data lead to biased stand age, as well as associated carbon stocks and fluxes. These were addressed in this study by imposing 20% error to pixel level biomass estimates and replacing zero biomass by the mean biomass of neighboring forest pixels with the same forest type and site productivity. " at lines P12L3 to P12L9.

(2) Adding more discussion on forest type group, it reads as "Accuracy of forest type group map in the PNW region ranges from 61% to 69% (Ruefenacht et al., 2008); besides, forest type groups for some pixels undefined from original data were assigned as the forest types of the nearest pixels. For the same biomass value, inferred stand age and estimated carbon fluxes can vary greatly given difference in forest type group (Fig. 4 & Fig. 6); however, forest type group induced biases in NEP were not accounted due to the lack of information on associated errors from the spatial assignment of forest type group." at lines P12L27 to P12L31. Uncertainties and associated errors from forest type group were not accounted due to the lack of information on those uncertainties. At the very least a confusion matrix would be required to characterize the probability mis-assignment and the probable alternative assignment of forest type group. Unfortunately such a confusion matrix was not made available for this mapped data product.

(3) Developing more discussion on ADS data, and now it reads as "The ADS dataset is known to be limited by the areas flown in the survey years, and likely underestimate the number of trees killed by bark beetles but likely overestimate the area of affected stands (Meddens et al., 2012). Uncertainties from ADS dataset have important consequences for the carbon balance and flux estimates from bark beetle outbreaks, part of them were accounted for by Ghimire et al. (2015)." at lines P12L31 to P13L1.

We expect errors from input map will bring bias in stand age and NEP estimates, so we propagated part of these uncertainties into our estimates, and provided a distribution of estimates, including quantiles and standard deviation, instead of a single value.

P11L27-28 Introducing new information in figures which are not described in the results. This should not be the case. If the figure and comparison is really needed then it should be included in the results and be part of the experimental design. or could be moved to SI.

**Response:** We divided this paragraph into three parts. The sentence of introducing comparison was moved to "Materials and Methods" at lines P7L13 to P7L15, comparing results and Fig. 10 (now Fig. 9) was moved to "Results" at lines P10L21 to P10L24, and we keep the discussion on discrepancy from comparison in "Discussion" section at lines P13L15 to P13L25.

P12L12-15 New analysis should not be introduced in the discussion. Also Figure 11 did not appear in the results either. Again, if these comparison and figure is needed then make it part of the experimental design and introduce it in the results section first.

**Response:** Similar to response to previous comment, we divided this paragraph into three parts. The sentence of introducing comparison was moved to "Materials and Methods" at lines P7L15 to P7L16, comparing results and Fig. 11 (now Fig. 10) was moved to "Results" at lines P10L24 to P10L29, and we keep the discussion on discrepancy from comparison in "Discussion" section at lines P13L26 to P13L31.

P12L14-15 "...distribution agrees well with that for our undisturbed..." poor working rephrase.

**Response:** This sentence was edited as "Overall, the pattern of FIA-derived age distribution matches well with that derived from our study, but with our study having consistently lower forest areas at age classes larger than 20." at lines P10L26 to P10L28.

P12L18, P13L4,L18 Multiple definitions of what is a young forest. Can you reconcile these?

**Response:** P12L18: We replaced "stand of young ages" by "stand with ages ranging from 0 to 24" at lines P13L27 to P13L28.

*P13L4: We deleted "young" from "young regenerating forests" at line P14L24. P13L18: We replaced "relatively mature (>24 year old) forests" by "recently undisturbed forests" at line P15L8 to P15L9.*

P12L22 "A portion of this difference can be attributed to smaller net carbon losses..." If I understand correctly here you mean greater loss / more negative? Comparing between -4 TgC and -7 TgC? Not clear.

**Response:** Correct. This study reports more net carbon losses  $(-7 \text{ Tg C y}^{-1})$  than  $-4 \text{ Tg C y}^{-1}$  reported in our previous work (Williams et al., 2014). The sentence was edited as "A portion of this difference can be attributed to larger net carbon losses from forestlands (-7 Tg y-1 carbon loss in this study vs. -4 Tg y-1 carbon loss in Williams et al.) due to recent (1986 to 2010) disturbance by either harvest or fire." at lines P14L9 to P14L11.

P12L31-33 Is the PNW region representative of forestry in the US?

**Response:** We meant western US, not US at P12L31-33 (now P14L17 to P14L20). Bark beetle outbreak in PNW is not representative of forests in western US, where rocky mountain north (RMN) and rocky mountain south (RMS) regions have higher mortality rates (37%, 35%), while a lower rate (1%) in Pacific Southwest (PSW) region (Ghimire et al. 2015). So we use PNW mortality percentage in western US to obtain NEP reduction due to bark beetle outbreak in PNW.

P13L19 Good to see some comparison with other studies. Are there any more available to broaden the discussion?

Response: Thank you. We have current discussions on NEP comparisons.

Figures

All of the figure captions need to be expanded to make clear where the data / analysis from each figure comes from and any key features. Also there appears to be substantial repetition of the disturbance figure. Can the figures be re-arrange to minimize this / move some of these maps to the SI.

**Response:** We edited all the figure captions to include more detailed description, and deleted Fig. 1.

We provided two figures on forest disturbance, one is year of last disturbance map from NAFD, MTBS and ADS, and the other one is integrated disturbance map (disturbance year and type), which is one of the resulting figures and new contribution to current disturbance maps.

Figure 5. These NEP do not show C loss, even though your analysis does. These figures reinforce the confusion between whether or not you are analysis the C balance of the ecosystem as a whole or just the live biomass. If you are analyzing the whole ecosystem the NEP would surely be negative directly after disturbance due to litter and soil C turnover?

**Response:** We analyzed the whole ecosystem, and Fig. 5 (now Fig. 4) only displays positive NEP trajectories after disturbance. More texts were added in caption of Fig. 5 (now Fig. 4), "The typical pattern of NEP following a disturbance involves a large negative value immediately after disturbance, a rise for a number of years to reach a maximum rate of carbon uptake, and then a gradual decline. Only the positive part of NEP trajectories were displayed, the full range of post-disturbance NEP curves across forest types, productivity classes, and disturbance types are presented in the supplementary figures (Fig. S1, S2, S3, S4)."

Figure 10. In your analysis are "Years Since Disturbance" and "Stand Age" the same thing? If so why in the same figure are you referring to this by different names. Particularly as in the caption you refer to both as "Stand age".

**Response:** In our study, for undisturbed or stand-replacing disturbed forests, "years since disturbance" and "stand age" are the same thing; while for partial disturbed forest, they are not. We used term "years since disturbance" in this paper, and Pan et al. (2011) used term "stand age", so we keep both to be consistent with original paper. We edited the figure caption of Fig. 10 (now Fig. 9).

We would like to thank the reviewer for giving us very helpful suggestions that help us greatly improve the quality of this manuscript. We provided our responses to all the comments point by point below (italicized typeface) with page (P) and line (L) numbers referred if necessary.

This is a pretty good and potentially useful paper that could be published after some modifications. The main problems I can identify are (1) the introduction is poorly written in places, (2) an important and highly relevant citation is missing, and (3) the discussion needs more work.

**Response:** (1) We rewrote most of the "Introduction", making the content proportional to the objectives. (2) The suggested citation was included in our introduction. (3) We expanded more in discussion. Please refer to point-by-point responses for detailed information. Please refer to point-by-point responses for detailed information.

Detailed comments below address some problems with the introduction. The missing citation is more troubling since it presents an alternative approach to using the CASA model for estimating growth (or NEP) which is a center piece of this study. The citation is: Raymond, C. L., Healey, S., Peduzzi, A., Patterson, P. 2015. Representative regional models of post-disturbance forest carbon accumulation: Integrating inventory data and a growth and yield model. Forest Ecology and Management 336: 21-34. This should be referenced in a couple of places (p. 2 line 20 and p. 4 line 1).

**Response: Thanks for suggested citation, we have included it in our introduction at line P3L20.**

The discussion should compare using the CASA model and using the Raymond et al. approach which relies on an FIA driven empirical model, the Forest Vegetation Simulator (FVS). What are the advantages and disadvantages of each approach, and do they yield similar results (the regions are different but still may be able to compare results for one or two forest types). I also suggest that the discussion should explore in more depth the many assumptions and inferences that have to be made to estimate time since disturbance for "undisturbed" pixels (section 2.2.2). For example, the Kellndorfer biomass map used to estimate biomass of "undisturbed" pixels has fairly high uncertainty at the pixel level; some pixels were assigned forest types based on a nearby neighbor pixel, etc. By the way, the title of section 2.2.2 is an oxymoron – if the pixel is "undisturbed" there should not be a time since disturbance. So instead of "undisturbed" the authors should use a different term to identify pixels that had no detected disturbance since 1986, perhaps something like "recently undisturbed".

**Response:** We didn't compare CASA and FVS models in discussion for two reasons: (1) the process of carbon cycle model is not the main focus in this paper, which has been described and discussed in our prior work (Williams et al., 2012, Ghimire et al., 2012, Ghimire et al., 2015). Our objective related to CASA is to use CASA-derived carbon stock and flux trajectories to do mapping; (2) it's not appropriate to compare carbon trajectories developed for different study area. Both CASA and FVS models heavily depend on FIA data, without same/similar FIA input, the comparison won't make a solid point. But when we moved to Rocky Mountain region in our future work, it sounds good to make such comparison.

We added a new discussion paragraph on Kellndorfer NBCD biomass products at lines P12L3 to P12L9. It reads as "Second, we assume RS-derived NBCD biomass products were well calibrated by fieldderived biomass. However, the correlation coefficients between observed and predicted biomass were estimated to be 0.62-0.75 in the PNW region (Kellndorfer et al., 2012). And at 30 m pixel level, NBCD biomass values were biased with a large number of zero biomass values that had predictions in local biomass products (Huang et al., 2015). Discrepancies in biomass values between RS- and field-derived data lead to biased estimates in stand age and associated carbon stocks and fluxes. These were addressed in this study by imposing 20% error to pixel level biomass estimates and replacing zero biomass by the mean biomass of neighboring forest pixels with the same forest type and site productivity."

We also added more discussion on forest type group at lines P12L27 to P12L31. It reads as "Accuracy of forest type group map in the PNW region ranges from 61% to 69% (Ruefenacht et al., 2008); besides, forest type groups for some pixels undefined from original data were assigned as the forest types of the nearest pixels. For the same biomass value, inferred stand age and estimated carbon fluxes can vary greatly given difference in forest type group (Fig. 4 & Fig. 6); however, forest type group induced biases in NEP were not accounted due to the lack of information on associated errors from the spatial assignment of forest type group."

The title of section 2.2.2 was edited as "Time since disturbance for recently undisturbed forest pixels".

Specific comments

The title is too long and redundant. Suggest "High-resolution forest carbon flux mapping in the Pacific Northwest with disturbance legacies inferred from remote sensing and inventory data". Could also leave out "in the Pacific Northwest".

**Response:** We apologize that our title didn't fully convey main objectives and focuses of this manuscript, we edited the title as "High-resolution mapping of time since disturbance and forest carbon flux from remote sensing and inventory data on harvest, fire, and beetle disturbance legacies in the Pacific Northwest".

p. 1 line 22: delete the second "probabilistic,"

Response: Second "probabilistic" was deleted as suggested at line P1L20.

p. 1 line 26: re-word so that it does not appear that tracts of land can somehow "see".

**Response:** "seen" was deleted from the sentence at line P1L24.

p. 2 line 13: replace "is itself a sort of record of" with "reflects"

Response: Replaced as suggested at line P2L13.

p. 2 line 14: replace "general" with "predictable rate of"

Response: Replaced as suggested at line P2L13.

p. 2 line 22-23: needs some rewording. The idea is that it is important to include smallscale disturbances down to some minimum threshold, not that disturbances typically are at this small scale.

**Response:** The sentence was edited as "characterization of time since disturbance across landscapes at a scale of being able to detect small-scale disturbance events, typically around 100 m or less." at lines P2L22 to P2L23.

p. 2 lines 27-28: add "at smaller scales" to the end since national forests inventories can provide useful guidance only at larger scales. But importantly note, it is possible to conduct field inventories at very small scales, so the statement is not very correct at all, only partially correct with respect to national forest inventories.

**Response:* "at smaller scales" was added at the end of this sentence at line P2L28.**

p. 3 lines 11-12: One objective is clearly stated. What are the others? The last sentence of this paragraph seems to be another objective, but then, I'm confused as to whether the purpose is to develop a method for large-scale monitoring and management, or small-scale, or both?

**Response:** We rewrote the last paragraph in Introduction at lines P3L23 to P3L31, where now all of the objectives are listed. It reads as "This study estimates and maps time since disturbance at a fine scale of 30 m from RS-derived products and FIA-derived biomass growth curves, and then maps net ecosystem productivity (NEP) based on disturbance history, time since disturbance and carbon flux legacy. The specific objectives in this study are to: (1) introduce a method for inferring a pixel's representative time since disturbance from RS-derived biomass and disturbance products at the 30 m resolution; (2) map NEP based on pre-existing, model-derived carbon stock and flux trajectories that describe how NEP changes with time following harvest, fire, or bark beetle disturbances of varying severity; (3) propagate uncertainties from RS-derived biomass products and FIA into uncertainty quantification of stand age and NEP. Our research represents an approach to map carbon stocks and fluxes at a high resolution across the conterminous US in support of national carbon monitoring, reporting, and management."

The last sentence of this paragraph is misleading, we have edited it as shown above. Our method will be used to map carbon stock and fluxes at a fine scale in the conterminous US.

p. 3 lines 27-28. Terminology again – "undisturbed" pixels by definition should not have a time since disturbance.

*Response:* "undisturbed forest pixels" has been edited as "recently undisturbed forest pixels" at line *P4L9.*

p. 3 line 32: Biomass curves were developed by forest type group and productivity class. How were these 2 classes allocated to the 5 NEP classes described on p. 4 lines 2-4?

**Response:** Biomass curves by forest type group and productivity class were derived from FIA and used to infer stand age for forest pixels undisturbed during remote sensing observation period.

NEP curves were derived from CASA carbon cycle process model with inclusion of disturbance processes. Biomass curves were used to adjust model's rates of NPP and wood turnover for each forest type group and productivity class. At the final stage of the modeling, the disturbance processes imposed stand-replacing harvest, fire or insect-induced partial disturbance to generate carbon stock and flux curves as a function of time since disturbance, and are specific to forest type group, site productivity class, disturbance type and disturbance severity.

p. 3 line 35: add citation after ". . .varying severity".

**Response:** Citations "Williams et al. 2012, Ghimire et al., 2012, Ghimire et al., 2015" were added after "... varying severity" at line P4L16.

p. 6 line 26: replace "stand" with "standard"

Response: Replaced as suggested at line P7L11.

p. 8 line 21: sentence that begins with "Again" needs editing.

**Response:** This sentence was edited as "Again in contrast, bark beetle outbreak areas for low and high productivity classes are similar in Douglas-fir forests, but beetle outbreak occurrence was about three times more likely in low productivity sites." at lines P9L20 to P9L22.

p. 9 lines 24-25: the imprint is not so clear to me. Maybe need to highlight somehow on the graphic.

**Response:** We included the position description of Biscuit fire in the bracket at line P11L4, "bottom left of Fig. 8a, also refer to bottom left of Fig. 5a & 5b".

p. 10 lines 12-14: One could argue that inventory data does not provide such a reliable estimate of biomass/age. Both of these variables can be rather difficult to measure/estimate especially with respect to the selection of biomass equations, but also the difficulty of assigning a stand age to stands that are uneven-aged.

**Response:** We added more discussion on the first assumption as suggested at lines P11L27 to P12L2. "However, both stand age and biomass are difficult to measure and estimate, especially considering the difficulty of assigning a stand age to uneven-aged forest stands, as well as selecting appropriate speciesspecific biomass equations (Parresol, 1999). If FIA ages are older than actual stand ages, the associated forest biomass will be underestimated, and stand age inferred from biomass products will be overestimated. And younger FIA ages than actual ages will result in an overestimation in biomass accumulation, but an underestimation in biomass-inferred stand ages. Though a possible bias in stand ages, our estimates of carbon stocks and fluxes are not likely to be largely adjusted by a stand age bias within 5 years (Williams et al., 2012)."

p. 12 lines 1-7: not statedâA TFIA does not do a good job of detecting recent disturbances because the remeasurement cycle in the PNW is about 10 years, so the average time lag of the data at any point in time is at least 5 years.

**Response:** We added the suggested point in this paragraph at lines P13L29 to P13L30, now it reads as "
[revised manuscript text omitted]

---

## Author Response (AR2)

*Summary of changes:*

*We have extensively revised the manuscript to address the concerns of the editor and two reviewers, including rewriting, editing and adding more text in several sections based on the reviews. We also clarified a number of confusing sentences throughout the manuscript, updated tables, figures, references and the supplementary document, and performed an overall grammar check. Please refer to "response to reviewers" for detailed information. Here are the **major** revisions we have made in the manuscript:*

1. *We edited the title as "High-resolution mapping of time since disturbance and forest carbon flux from remote sensing and inventory data to assess harvest, fire, and beetle disturbance legacies in the Pacific Northwest". Quantification and mapping of time since disturbance is an important objective in this paper, while applying carbon flux trajectories to newly derived time since disturbance map is another focus.*

2. *We rewrote many paragraphs in the "Introduction", particularly the last paragraph where all of the objectives are listed. Now it reads as "This study estimates and maps time since disturbance at a fine scale of 30 m from RS-derived products and FIA-derived biomass growth curves, and then maps net ecosystem productivity (NEP) based on disturbance history, time since disturbance and carbon flux legacy. The specific objectives in this study are to: (1) introduce a method for inferring a pixel's representative time since disturbance from RS-derived biomass and disturbance products at the 30 m resolution; (2) map NEP based on pre-existing, model-derived carbon stock and flux trajectories that describe how NEP changes with time following harvest, fire, or bark beetle disturbances of varying severity; (3) propagate uncertainties from RS-derived biomass products and FIA into uncertainty quantification of stand age and NEP. Our research represents an approach to map carbon stocks and fluxes at a high resolution across the conterminous US in support of national carbon monitoring, reporting, and management."*

3. *We have added more explicit descriptions on the CASA modelling, carbon stocks, allocations and transfers in live pools and dead pools, and carbon emissions from dead carbon pools in section 2.3.1. In this study, we mainly focused on the approach of combining CASA-derived trajectories with the newly mapped time since disturbance derived from objective 1 to demonstrate that this method can be used to develop spatial representations of NEP.*

4. *We have added more discussion of assumptions made in this study in section 4.1. We present a mature characterization and propagation of errors from biomass and disturbance year input data sources, and formally assess how these can bias stand age and NEP estimates. These major error sources are fully propagated as part of the uncertainties that we provide as a distribution of estimates, including quantiles and standard deviations, instead of a single value. However, we are unable to assess potential biases from some of the input datasets/parameters due to limited/no information on those errors.*

5. *We replaced Fig. 5 (now Fig. 4) by a figure displaying full range of NEP after harvest (Fig. 4a, 4b), fire (Fig. 4c, 4d) and bark beetle (Fig. 4e, 4f). More texts were added in caption of Fig. 5 (now Fig. 4).*

6. *We included a carbon allocation and transfer flowchart of CASA model (Fig. S1) and post-disturbance trajectories on NEP, heterotrophic respiration (Rh), carbon stocks in carbon pools (soil organic carbon, litter, slow turnover soil carbon, aboveground coarse woody debris,*

*belowground coarse woody debris, and total live woody biomass) in the supplementary document. Fig. S2, S3, S4, S5, S6 present post-disturbance NEP curves across forest types, productivity classes, and disturbance types, and Fig. S7, S8, S9, S10, S11 are shown for Rh trajectories. Characteristic trajectories of post-disturbance carbon stocks in carbon pools are provided as well (Fig. S12, S13, S14, S15, S16, S17 for harvest as an example).*
*We would like to thank the reviewer for giving us very helpful suggestions that help us greatly improve the quality of this manuscript. We provided our responses to all the comments point by point below (italicized typeface) with page (P) and line (L) numbers referred if necessary.*

General comments

The manuscript aims to address current issues in constraining forest C dynamics and stocks in relation to multiple different types and intensity of disturbance. The authors combine a range of data including national inventories, management databases, airborne and space-borne remote sensing. These data are then combined / utilized through both statistical (yield curves) and simulation based modelling (CASA) approaches. As such the manuscript is highly relevant and well within scope of Biogeosciences and I believe will ultimately be published in Biogeosciences. However I believe there is additional scientific value that should be drawn from the current analysis and a substantial re-write to improve readability prior to publication. The following general comments are split broadly between scientific and presentation.

Globally C stored in forests is split roughly equally between woody biomass and soil organic matter (e.g. Pan et al 2011). However the manuscript focuses on estimates of above ground biomass stocks and disturbance to these stocks, lacking any analysis or discussion of soil carbon stocks. I recognize that the authors report net ecosystem productivity (defined as NEP = NPP-Rh), but I would prefer you to distinguish between accumulation and losses between the live biomass and dead organic matter. Or state clearly why not given that you are reporting ecosystem scale values. Also I do not believe that the authors have extracted all relevant information for the above ground biomass stocks. For example in Section 3.4 L29 The authors state "Spatial variations in mean annual NEP are noticeably correlated with the time since disturbance, forest type group, and site productivity strata...". This could be shown more clearly in an x~y plot and / or this "noticeable" correlation could be explicitly quantified to distinguish the relative importance of the drivers. Further detail follows in the specific comments section.

*Response: In this study, aboveground biomass accumulation curves were used for two purposes: (1) to infer stand age from RS-derived biomass data for objective 1; (2) to be a constraint to adjust wood turnover rate (mortality and shedding) and default output NPP by a scalar parameter NPPscalar in CASA modeling, which hence influences accumulation of live biomass and amounts of carbon allocated to live and deal pools, this was clarified at lines P8L8 to P8L10.*

*More details on CASA modeling were provided, including CASA modeling, carbon stocks, allocations and transfers in live pools and dead carbon pools, and emissions from dead carbon pools at lines P7L29 to P9L9. It reads as:*

*"The CASA model used here is based on Randerson et al. (1996) and operates on a monthly time step. It uses a light use efficiency approach to simulating net primary productivity (NPP) based on RS-derived absorption of photosynthetically active radiation, biome parameters, and climate data. The model then allocates NPP to three live carbon pools (leaves, roots, and wood), and transfers carbon to dead pools (litter and soils) based on biome-specific rates of tissue turnover. Carbon in dead organic matter pools is*

*transferred between pools (surface structural C, surface metabolic C, soil structural C, soil metabolic C, aboveground coarse woody debris C, belowground coarse woody debris C, surface microbial C, soil microbial C, slow C and passive C) (Fig. S1). Amounts of carbon transferred to microbial pools and carbon emitted to the atmosphere from microbial decomposition of soil and litter, i.e. heterotrophic respiration (Rh), depend on the rate and efficiency of heterotrophic consumption which varies between pools in the model, and also depends on biome- and pool-specific chemistry and site-specific climate setting such as soil moisture and temperature. The difference between NPP and summed Rh of the ten detrital pools is then calculated as NEP. Aboveground biomass-age curves sampled from FIA database were used as a constraint to adjust wood turnover rate (mortality and shedding) and default output NPP by a scalar parameter, which hence influences accumulation of live biomass and the amounts of carbon allocated to live and deal pools. The adjustment of those CASA parameters was performed for each specific combination of forest type group and site productivity class. Implementation of the model involved a simulation sequence beginning with spin-up to equilibrium carbon pools using FIA-adjusted NPP followed by a pre-disturbance with ensuing regrowth to a set of pre-disturbance ages, and lastly imposition of the disturbance of interest to generate flux and stock dynamics in response to harvest, fire, or beetle outbreak disturbance types at different severities. The fate of carbon influenced by disturbances varied by disturbance type as described below. Disturbance types considered in the model included stand-replacing harvest, fire, and bark beetle outbreak, with full descriptions of the carbon dynamics in pools after disturbances provided in Ghimire et al., 2012, Williams et al., 2014, Ghimire et al., 2015 and described further below.*

*Model treatments of disturbance impacts to the carbon cycle were as follows. For all disturbance type cases, a post-disturbance decline and ensuing recovery of NPP and fractional allocation to wood were modelled as a negative exponential function of time since disturbance, recovering to the pre-disturbance level within eight years (Williams et al. 2012). The mortality and fate of disturbance killed and/or combusted carbon pools differed by disturbance type. For harvest (Williams et al. 2014), the post-disturbance biomass was set to 50% of the aboveground live wood biomass reported in FIA data for the 0 to 20 year old age class, regardless of the pre-disturbance biomass condition. Harvest-killed live wood, leaves, and roots were calculated from their corresponding fractions of pre-disturbance to post-disturbance conditions. Eighty percent of the disturbance-killed aboveground live wood was assumed to have been removed from the site with the remainder being treated as slash subject to decomposition as coarse woody debris. All leaves of disturbance killed trees are assumed to decompose on-site. All belowground wood that succumbs to mortality enters a belowground coarse woody debris carbon pool. For fire (Ghimire et al. 2012), disturbance kill is portrayed as partial mortality events in which fires reduce pre-fire live biomass pools based on the fractional tree mortality which varies by forest type and fire severity class (high, medium, or low to match the Monitoring Trends in Burn Severity dataset). The amount of live biomass remaining after a fire is calculated from the fraction of vegetation mortality emerging from an extensive literature survey. Fire-killed material is either directly combusted and released to the atmosphere or transferred to dead carbon pools. The same applies to foliage and root mortality, though roots are not directly combusted. Fire killed trees enter a new standing dead pool in the model with a fast turnover fall rate of 10 years post fire with transfer to the CWD pool. Litter and soil organic carbon in the upper soil layers were also vulnerable to combustion. All of these rates were based on literature review (Ghimire et al. 2012). For bark beetle outbreaks (Ghimire et al. 2015), beetle killed biomass was simulated for a wide range of intensities from near zero to 100% mortality to generate a family of curves that could be applied in the mapping stage. Beetle attack caused leaf carbon to enter the surface litter pool, aboveground wood to enter a snag pool, belowground wood and fine root to enter corresponding soil carbon pools. The portions killed for each were based on the percent mortality*

*imposed for each severity level being simulated, derived from the ratio of the biomass at the midpoint of each severity class (of 1680 levels of intensity) to the pre-disturbed aboveground biomass values corresponding to the mean age of a given forest type under attack (Ghimire et al. 2015). Soil carbon pools respond to all of these dynamics according to the model's climate-mediated turnover times for each carbon pool, and the associated carbon flows (Fig S1)."*

*In addition, we included a carbon allocation and transfer flowchart of CASA model (Fig. S1) and post-disturbance trajectories on NEP, heterotrophic respiration (Rh), carbon stocks in carbon pools (live and dead) in the supplementary document. Fig. S2, S3, S4, S5, S6 presents post-disturbance NEP curves across forest types, productivity classes, and disturbance types, and Fig. S7, S8, S9, S10, S11 are shown for Rh trajectories. Characteristic trajectories of post-disturbance carbon stocks in carbon pools (soil organic carbon, litter, slow turnover soil carbon, aboveground coarse woody debris, belowground coarse woody debris, and total live woody biomass) are provided as well (S12, S13, S14, S15, S16, S17 for harvest as an example). In this study, we mainly focused on the approach of combining NEP trajectories with the newly mapped time since disturbance derived from objective 1 to simply demonstrate that this method can be used to develop spatial representations of NEP.*

*For section 3.4 L29 (now P12L6 to P12L7), we estimated NEP based on carbon flux trajectories, which vary by time since disturbance, forest type group and site productivity. The relationships between NEP and these variables have been presented in the trajectory curves in Fig. 4 and Fig. S2, S3, S4, S5, S6 in supplementary document. Because these trajectories are directly applied to map NEP, there is limited new information, if any, in the fact that NEP is correlated with time since disturbance, forest type group, and site productivity strata. Therefore, we think x~y plot is unnecessary. Furthermore, our intention is not to explore relationships between NEP and input data. Correspondingly, we revised the sentence in question to read "Spatial variations in mean annual NEP are determined by differences in strata of ..."*

The overall writing style of the manuscript needs to improved to benefit the flow of reading and in particular clarity. For example the methods overview needs to be clearer as to the overall structure of the analysis and their connections. The authors provide extensive detail on the data sources and what information they provide, however the fact that these data will used as constraints or drivers in the CASA model is not made clear until the final section before the results in Section 2.3.1. This is particularly confusing as all the description of data given is in reference to above ground biomass while at the same time stating that the results from the analysis are the net ecosystem productivity. Moreover the number of words in both the methods and results sections dedicated to the various disturbance maps produced appears disproportionate given that the title and the conclusions imply that C stocks and dynamics are the primary focus. I would consider way to simplify this information and attempt to move some of it into the supplementary material. Also I note that Figures 10 and 11 do not feature in the results section at all, instead are used to introduce new information in the discussion which is inappropriate. These figures should be introduced in the results section of they could be moved to the supporting information.

**Response:** *We apologize that our title didn't fully convey main objectives and focuses of this manuscript, we edited the title as "High-resolution mapping of time since disturbance and forest carbon flux from remote sensing and inventory data to assess harvest, fire, and beetle disturbance legacies in the Pacific Northwest". We also rewrote the last paragraph in Introduction at lines P3L27 to P4L2, where now all of the objectives are listed. Quantification and mapping of time since disturbance is an important objective in this paper, and while it is being generated for the purpose of carbon flux, the characterization of carbon fluxes was the focus of the group's prior work and is thus of less emphasis here. We did revise the present paper to include a more comprehensive overview of the carbon modeling performed in our prior work.*

*The purpose behind why we introduce a number of data sources is **not** to describe how they are used as constraints or drivers of the CASA modelling, though biomass~age curves is a constraint to adjust CASA model; **but rather to** introduce the methodology of how we make use of those currently available data sources to infer a pixel's representative time since disturbance, which is objective 1 in this paper. Since all the input data sources can be used directly except for a disturbance map with disturbance type/severity and biomass~age growth curves, so we provided explicit description of how to derive these two important components, and the resulting maps, figures and numbers. Another objective is to make use of the time since disturbance map that we derived for objective 1 with carbon flux trajectory curves from our previous work to map NEP. Since the processes of the CASA model and the methods for deriving carbon flux trajectories were each described in detail in our prior papers (Ghimire et al., 2012, Williams et al., 2014, Ghimire et al., 2015), here we emphasize the approach of applying these trajectories and resulting NEP maps.*

*For Fig. 10 and Fig. 11, we divided the two relevant paragraphs in "Discussion" into three parts. The sentence introducing the comparison between the results from this study and those of previous work was moved to "Materials and Methods" and the comparison of those results (formerly Fig. 10 and Fig. 11 , now Fig. 9 and Fig. 10) were moved to "Results". We keep associated discussion about likely sources of the discrepancy in the "Discussion" section.*

Specific comments

The following comments are broken down into Abstract, Introduction, Methods, Results, Discussion and Conclusion sections. General comments on each section will be followed by specific comments with page (P) and line (L) numbers.

Abstract The abstract could be made to flow more easily and make clear that the analysis is feeding into a C-cycling model that represents both the live and dead carbon pools. This will reduce confusion between described / yield curves used which constrain above ground biomass while at the same time reporting a net ecosystem value.

***Response:*** *We edited the abstract to reflect objectives of this manuscript. We clarified biomass growth curves and carbon flux trajectories were used for different objectives. The CASA carbon cycle model was now mentioned in the abstract at lines P1L18 to P1L19.*

Introduction All relevant information appears to be present in the introduction, however not all of the information is clear. I would recommend the use of topic-sentences to improve clarity of your message for each paragraph. Moreover there are a number of sentences where the wording is awkward to read.

***Response:*** *We rewrote most of the "Introduction", making the objectives much clearer and the content proportional to the objectives.*

P2L29 "...remote sensing techniques..." should include "...remote sensing (RS) techniques" as RS is used later.

***Response:*** *Acronym of remote sensing has been included in brackets at line P2L32.*

P2L30 "...remote sensing techniques provide..."

***Response:*** *"... are providing ..." has been changed to "... generate ..." at line P2L33 to avoid repeat use of the word "provide" in the same sentence.*

P2L32 "Such products miss small scale events and extend only so far back in time..." awkward wording. Please reconsider e.g. "However, RS products frequency miss small scale events and only cover the last several decades..."

*Response: This sentence was edited as "However, such disturbance products only record events that occurred within the last several decades, ..." at lines P3L2 to P3L3.*

P3L8 "...provide a way forward to capture at least some of the information that is missing but needed..." awkward wording, Please reconsider rewording.

*Response: This sentence was edited as "Nonetheless, RS-derived forest biomass still provides a valuable way of characterizing the pixel-scale (e.g. 30m or 250 m) legacy effects of disturbance that occurred prior to RS observations, which is required for quantifying carbon stock recovery and carbon uptake and release rates over large areas." at lines P3L13 to P3L15.*

P3L9 – L20 The final paragraph would be a good place to make a clear statement of the studies objective (key questions) and novelty. However the final paragraph here mixes further introduction and aims. This could be split and made clearer.

*Response: We rewrote the last paragraph in Introduction at lines P3L27 to P4L2, where now all of the objectives are listed. Now it reads as "This study estimates and maps time since disturbance at a fine scale of 30 m from RS-derived products and FIA-derived biomass growth curves, and then maps net ecosystem productivity (NEP) based on disturbance history, time since disturbance and carbon flux legacy. The specific objectives in this study are to: (1) introduce a method for inferring a pixel's representative time since disturbance from RS-derived biomass and disturbance products at the 30 m resolution; (2) map NEP based on pre-existing, model-derived carbon stock and flux trajectories that describe how NEP changes with time following harvest, fire, or bark beetle disturbances of varying severity; (3) propagate uncertainties from RS-derived biomass products and FIA into uncertainty quantification of stand age and NEP. Our research represents an approach to map carbon stocks and fluxes at a high resolution across the conterminous US in support of national carbon monitoring, reporting, and management."*

Methods The methods are very long (which I accept may be required) and would benefit from an improved overview section. Where possible the methods sections would benefit from moving some material to the supplementary material to improve focus.

*Response: We made several edits in "2 Materials and Methods", especially making the "2.1 Overview" more clarified.*

P3L25 "...recent disturbance..." how recent?

*Response: "...recent disturbance..." at line P4L8 refers to disturbance since the starting year of disturbance products as noted in the manuscript. It was edited as "a recent disturbance since the beginning of the relevant RS observations".*

P3L28 "...terms "time since disturbance" and "stand age" " would it be possible to pick one of these terms and use it consistently?

*Response: We prefer to use term "time since disturbance" based on context of this paper, while in FIA and other papers, such as Pan et al. (2011), "stand age" was mainly used for undisturbed forests. To make the connection between two terms, here we wrote: terms "time since disturbance" and "stand age" are used interchangeably for recently undisturbed forest pixels thereafter at lines P4L11 to P4L12.*

P3L29 "It was inferred..." possibly "Stand age was inferred..." would be clearer?

*Response: The sentence was edited as "Stand age was inferred ..." at line P4L12.*

P3L30 "The (yield?) curves were sampled from FIA data and specific to forest type and group and site productivity class." Is this information known in all cases? If not, what is assumed in their place?

*Response: Two out of 16 forest type groups did not have FIA-derived biomass-age curves available, they are "Pinyon/Juniper" and "California Mixed Conifer", so we used curves of "Other Western Softwood" instead.*

P4L1 "Net ecosystem productivity (NEP)" prior to this point all data / methods mentioned implies that this study is focusing on above ground biomass only. A link to CASA needs to be made earlier to make this clear.

*Response: Two main objectives in this paper are to map time since disturbance and NEP. In section "2.1 Overview", we spent the first paragraph on methods to infer time since disturbance, and the second paragraph on mapping NEP. So we think it's appropriate to mention NEP at the beginning of the second paragraph. Besides, CASA model was mentioned here at lines P4L17 to P4L18 now.*

P4L13 It would be useful to have a table with the different data sources listed and state the data and time period they cover.

*Response: We added a new table (Table 1, shown below) in the manuscript to summarize the data sources.*

**Table 1.** Data sources for inferring time since disturbance for recently disturbed and undisturbed forest pixels.

| Data | Description | Source | Year | Input for recently disturbed or/and undisturbed forests |
|---|---|---|---|---|
| NAFD | Forest disturbance | Landsat | 1986-2010 | a, b |
| MTBS | Burned area and severity | Landsat | 1986-2010 | a |
| ADS | Area of insect outbreak and number of trees killed | Aerial survey | 1997-2010 | a |
| NBCD | Aboveground live biomass | Landsat, SRTM, FIA | 2000 | b |
| Forest Type Group | Forest type group | MODIS, NLCD, etc. | 2001 | b |
| Site Productivity | Fraction of high productivity | FIA | 1984-2014 | b |
| Biomass-age Curves | Biomass accumulation as a function of stand age | FIA | 1984-2010 | b |

[a] Data is one of the inputs for inferring time since disturbance for recently disturbed forest pixels.
[b] Data is one of the inputs for inferring time since disturbance for recently undisturbed forest pixels.

P5L1 "These assumptions...been reported in the literature." Long sentence, can you break some of the sentences with lists, multiple concepts or conditions down.

*Response: This sentence has been broken into a short sentence with three lists at lines P5L20 to P5L23. It reads as "These assumptions are based on the rationales: (1) MTBS records most of the notable fire events in the region, (2) harvest events are one of the most ubiquitous stand replacing disturbance types active in the region, (3) ADS-mapped polygons of bark beetle infestations often include unaffected stands as has been reported in the literature".*

P5L5-10 Consider making this list in a table

*Response: We keep four rules in lists.*

P5L11-12 "The target year...was 2010". Possibly make this point earlier say in the overview or introduction aims?

*Response: Target mapping year of 2010 was now mentioned in "2.1 Overview" at line P4L9.*

P5L23 "age class from ..." how many age classes, are all equal in size?

*Response: There are 11 age classes in total: 0-20, 20-40, 40-60, 60-80, 80-100, 100-120, 120-140, 140-160, 160-180, 180-200, 200+. This information has now been included in Table S1.*

P5L25 All other units are given as SI. Please do so here too. Also it is odd that up until now forest biomass as been discussed, here you have swapped into volume. Can you convert or is there a reason for this?

*Response: The unit "cubic feet/acre/year" here is used by FIA and is the output unit from FIA EVALIDator. We didn't convert it to SI, since readers will know how we combine 7 site productivity classes (0-19, 20-49, 50-84, 85-119, 120-164, 165-224, 224+ cubic feet/acre/year) into two classes.*

P6L1 "Differences in forest masks..." which forest masks? Which products you are using?

*Response: Different forest masks were used in NAFD disturbance products and NBCD biomass products. This sentence was edited as "Differences in forest masks between NAFD disturbance and NBCD biomass products led to" at line P6L21. In our study, NAFD-based forest mask was used for the analyses and mapping.*

P6L2 "These were replaced by the mean biomass of other undisturbed pixels..." The distributions of stand age in Figure 11 are not Gaussian, would the median be better or is there little difference?

*Response: This is a great point. We do compare mean and median biomass of other undisturbed pixels with the same forest type and site productivity class (10 classes here from 0 to 1 with equal interval), and we found there is little difference between mean and median values.*

P6L12-13 Again SI units please.

*Response: Same reason as previous comment on SI. That is, we would like to keep the unit consistent with outputs from FIA EVALIDator, so readers would know how we combine 7 site productivity classes (0-19, 20-49, 50-84, 85-119, 120-164, 165-224, 224+ cubic feet/acre/year) into two classes.*

P6L24 "In reality fhigh is almost always between 0 and 1." Can you say what the mean value is or distributional information? Something more informative.

*Response: maximal $f_{high}$ = 0.996, minimal $f_{high}$ = 0.015 and mean $f_{high}$ = 0.530 in PNW. We included maximal $f_{high}$ and minimal $f_{high}$ values in the manuscript at lines P7L12 to P7L13.*

P7L4 This is the first mention of the CASA model. Please provide a brief description of the mode and how it works. This is needed given that you make reference to its process representation in the discussion P11L1-5. Also what is the model time step used. Over what period is CASA simulating these forests (prior to 2010)? which meteorological drivers are used (e.g. ERA-Interim, GFS)? How realistic are the spin up pool sizes relative to field estimates in undisturbed pixels. Your estimate of C loss in response to disturbance will partially dependent on soil losses which will also be dependent on their initial magnitude

after spin up (e.g. Exbrayat et al., 2014). If this information is available in the cited literature please make this clear.

*Response:* *We now include the following descriptions of CASA model at lines P7L29 to P9L9. It reads as:*

*"The CASA model used here is based on Randerson et al. (1996) and operates on a monthly time step. It uses a light use efficiency approach to simulating net primary productivity (NPP) based on RS-derived absorption of photosynthetically active radiation, biome parameters, and climate data. The model then allocates NPP to three live carbon pools (leaves, roots, and wood), and transfers carbon to dead pools (litter and soils) based on biome-specific rates of tissue turnover. Carbon in dead organic matter pools is transferred between pools (surface structural C, surface metabolic C, soil structural C, soil metabolic C, aboveground coarse woody debris C, belowground coarse woody debris C, surface microbial C, soil microbial C, slow C and passive C) (Fig. S1). Amounts of carbon transferred to microbial pools and carbon emitted to the atmosphere from microbial decomposition of soil and litter, i.e. heterotrophic respiration (Rh), depend on the rate and efficiency of heterotrophic consumption which varies between pools in the model, and also depends on biome- and pool-specific chemistry and site-specific climate setting such as soil moisture and temperature. The difference between NPP and summed Rh of the ten detrital pools is then calculated as NEP. Aboveground biomass-age curves sampled from FIA database were used as a constraint to adjust wood turnover rate (mortality and shedding) and default output NPP by a scalar parameter, which hence influences accumulation of live biomass and the amounts of carbon allocated to live and deal pools. The adjustment of those CASA parameters was performed for each specific combination of forest type group and site productivity class. Implementation of the model involved a simulation sequence beginning with spin-up to equilibrium carbon pools using FIA-adjusted NPP followed by a pre-disturbance with ensuing regrowth to a set of pre-disturbance ages, and lastly imposition of the disturbance of interest to generate flux and stock dynamics in response to harvest, fire, or beetle outbreak disturbance types at different severities. The fate of carbon influenced by disturbances varied by disturbance type as described below. Disturbance types considered in the model included stand-replacing harvest, fire, and bark beetle outbreak, with full descriptions of the carbon dynamics in pools after disturbances provided in Ghimire et al., 2012, Williams et al., 2014, Ghimire et al., 2015 and described further below.*

*Model treatments of disturbance impacts to the carbon cycle were as follows. For all disturbance type cases, a post-disturbance decline and ensuing recovery of NPP and fractional allocation to wood were modelled as a negative exponential function of time since disturbance, recovering to the pre-disturbance level within eight years (Williams et al. 2012). The mortality and fate of disturbance killed and/or combusted carbon pools differed by disturbance type. For harvest (Williams et al. 2014), the post-disturbance biomass was set to 50% of the aboveground live wood biomass reported in FIA data for the 0 to 20 year old age class, regardless of the pre-disturbance biomass condition. Harvest-killed live wood, leaves, and roots were calculated from their corresponding fractions of pre-disturbance to post-disturbance conditions. Eighty percent of the disturbance-killed aboveground live wood was assumed to have been removed from the site with the remainder being treated as slash subject to decomposition as coarse woody debris. All leaves of disturbance killed trees are assumed to decompose on-site. All belowground wood that succumbs to mortality enters a belowground coarse woody debris carbon pool. For fire (Ghimire et al. 2012), disturbance kill is portrayed as partial mortality events in which fires reduce pre-fire live biomass pools based on the fractional tree mortality which varies by forest type and fire severity class (high, medium, or low to match the Monitoring Trends in Burn Severity dataset). The amount of live biomass remaining after a fire is calculated from the fraction of vegetation mortality emerging from an extensive literature survey. Fire-killed material is either directly combusted and*

*released to the atmosphere or transferred to dead carbon pools. The same applies to foliage and root mortality, though roots are not directly combusted. Fire killed trees enter a new standing dead pool in the model with a fast turnover fall rate of 10 years post fire with transfer to the CWD pool. Litter and soil organic carbon in the upper soil layers were also vulnerable to combustion. All of these rates were based on literature review (Ghimire et al. 2012). For bark beetle outbreaks (Ghimire et al. 2015), beetle killed biomass was simulated for a wide range of intensities from near zero to 100% mortality to generate a family of curves that could be applied in the mapping stage. Beetle attack caused leaf carbon to enter the surface litter pool, aboveground wood to enter a snag pool, belowground wood and fine root to enter corresponding soil carbon pools. The portions killed for each were based on the percent mortality imposed for each severity level being simulated, derived from the ratio of the biomass at the midpoint of each severity class (of 1680 levels of intensity) to the pre-disturbed aboveground biomass values corresponding to the mean age of a given forest type under attack (Ghimire et al. 2015). Soil carbon pools respond to all of these dynamics according to the model's climate-mediated turnover times for each carbon pool, and the associated carbon flows (Fig S1)."*

*Meteorological and satellite based drivers, soil type, biome (forest) type, and other parameter and driver datasets are all described in the original papers (Ghimire et al., 2012, Williams et al., 2014, Ghimire et al., 2015), from which the corresponding results were directly taken.*

P7L10 "...curves describing carbon fluxes and stocks..." which stocks / fluxes where are they?

***Response:*** *They include carbon stocks in carbon pools (soil organic carbon, litter, slow turnover soil carbon, aboveground coarse woody debris, belowground coarse woody debris, and total live woody biomass), NPP, Rh and NEP. This sentence was edited as "...curves describing carbon stocks in carbon pools (soil organic carbon, litter, slow turnover soil carbon, aboveground coarse woody debris, belowground coarse woody debris, and total live woody biomass), NPP, Rh and NEP with time since disturbance ..." at lines P9L11 to P9L13. The complete set of these stocks and fluxes trajectories are shown in the supplementary document.*

*All these curves were derived in our prior work (Ghimire et al., 2012, Williams et al., 2014, Ghimire et al., 2015), citations were added at the end of this sentences.*

P7L15 "This study emphasized the use of NEP curves. Fig. 5 ..." Figure 5 seems to show that C losses do no occur whereas losses do occur in the results (Table 2) as presumably soil and litter C is being decomposed and undergoing mineralisation. So where is the C source represented?

***Response:*** *C losses occur after disturbances, Fig. 5 (now Fig. 4) was replaced by a figure displaying full range of NEP after harvest (Fig. 4a, 4b), fire (Fig. 4c, 4d) and bark beetle (Fig. 4e, 4f). More texts were added in caption of Fig. 5 (now Fig. 4), "The typical pattern of NEP following a disturbance involves a large negative value immediately after disturbance, a rise for a number of years to reach a maximum rate of carbon uptake, and then a gradual decline to a steady state."*

Results

What is the primary focus of the manuscript? A large part of the results section is taken up with a description of the various input maps into the analysis. Much of it seems like it should be in the methods sections as a description of the inputs or could be moved to the supporting information. Unless these are actually new numbers derived from the combination of multiple maps. At the moment it is not clear. Possibly an overview could be given to the results as it takes a lot of reading before you get to any information on the estimates of biomass stocks.

*Response:* We included four sections in "Results", the first is about a disturbance map, and the second is on biomass~age growth curves. We provided the resulting maps, figures and numbers of a disturbance map with disturbance type/severity and biomass growth curves, because all the input data for inferring time since disturbance can be readily used except for these two important components, besides these results are new. The third and fourth sections are maps of time since disturbance and NEP, which are the main objectives of this paper. We hope that the substantial revisions we have made clarify the corresponding focus of this manuscript.

P9L11-12 "...these curves yielded a smoothed fit to the inventory data rather than showing a saw-toothed increase with stand age." Here are you referring to saw-toothed due to managed thinning or stem mortality events?

*Response:* FIA samples were compiled from FIA plot measurements in Oregon and Washington. Managed thinning or stem mortality events could be part of reasons, but there are also some other possible reasons causing the erratic and fluctuating jumps, such as vagueries of plot-to-plot variability that span climate, soils, topographic and other variations across sampling plots.

P9L23 "Uncertainty on the time since disturbance forest pixels is not currently available from disturbance products and this was not mapped" Could the uncertainty in the yield curves on growth since disturbance be included? How strongly do the yield curves constrain CASA?

*Response:* No, it would not be logically sound or feasible to use uncertainty in the yield curves to characterize uncertainty in the time since disturbance mapped from remote sensing (Landsat spectral reflectances).

*The yield curves provide very important adjustments to the default NPP and wood turnover rates in the biome-scale parameters in the CASA model. Without this adjustment, the CASA model, which is designed for global scale applications, would not provide an accurate and fitting representation of forest biomass for the fine-scale and diverse settings of the US where we are applying the model.*

P9L29 "Spatial variations in mean annual NEP are noticeably correlated with..." Why not actually correlate them to quantify this? A new x~y figure might be useful here too.

*Response:* We estimated NEP based on carbon flux trajectories, which vary by time since disturbance, forest type group and site productivity. The relationships between NEP and these variables have been presented by trajectory curves in Fig. 4 and Fig. S2, S3, S4, S5, S6 in supplementary document, and are directly applied in the mapping exercise so there is no new information derived in the mapping aside from the spatial allocation. The patterns are indeed interesting and can be important for some applications but an x~y plot would not be particularly instructive (it would simply recover the trajectories applied in the mapping). Besides, we didn't mean to explore relationships between NEP and input data. So we edited this sentence as "Spatial variations in mean annual NEP are determined by differences in strata of ..." at line P12L6.

P9L30-31 "...weaker carbon sinks in the eastern, drier portion of the study area..." Again this could be show in an x~y plotting soil moisture / precipitation against C sink strength to quantify.

*Response:* Here we meant to describe a spatial pattern in NEP map, which has lower NEP values in the east side of the study area. We mentioned "drier" just because the eastern part of study area is less humid from our knowledge. We do not intend to emphasize the relationship between NEP and soil moisture/precipitation. We removed "drier" from the text at line P12L8.

P10L6 "Forestlands free of recent disturbance..." could be "Undisturbed forests are..." just trying to be consistent with the terms you use.

*Response: Edited as "Recently undisturbed forests are ..." at line P12L15.*

Discussion

P10L21-23 Awkward sentence please rephrase / breakdown into smaller parts.

*Response: This sentence was shortened as "Our method of inferring time since disturbance to estimate carbon flux and biomass accumulation relies on a number of data products and assumptions that need to be critically evaluated." at lines P13L3 to P13L4.*

P10L25-26 Odd place the begin new paragraph. You appear to be continuing your point from the first paragraph.

*Response: We would like to keep each assumption as one paragraph, we also added more discussion on the first assumption at lines P13L6 to P13L12.*

P10l27 It is not clear what you mean. Are you talking about how the stand-level biomass estimate was calculated or how the real world stand was managed / grew?

*Response: We mean how the real world stand was accumulated to the current amount of biomass. The sentence was edited as "... how that stand-level biomass was actually achieved ..." at line P13L21.*

P10L29 "...or also from a recent disturbance that reduced biomass to the current level." I think you need a reference here.

*Response: Reference "Xu et al., 2012" was added at the end of this sentence at line P13L24.*

P10L30 "...varies depending on the type of stand-replacing disturbance". Are you referring to e.g. clear felling vs fire?

*Response: Correct.*

P11L1-4 Currently you have not described the model used to provide required background for these statements.

*Response: This has now been added as noted above (responses to the comment starting with P7L4 above).*

P11L4 "...initial rise through stand initialization." Are you talking about early phases of forest growth? How long does initialization take?

*Response: Establishment of stands can of course take a variable amount of time depending on many factors. In the modeling (as described in our prior work) we assumed that NPP post disturbance rises to a steady rate over the course of 8 years post-disturbance, and that allocation of NPP to woody biomass also increases over that interval.*

P11L9 "...which are sure to have errors." Are there any estimates of this error?

*Response: We meant to say input maps (maps of biomass, forest type group, site productivity and forest disturbance) have errors. The accuracies of biomass and forest type group were assessed and provided by the data provider, however site productivity and forest disturbance not.*

P11L9-13 Should this not be first introduced in the methods section if these describe errors between field information and the maps you have used to constrain your model. Also, is there a bias associated with these errors? If so, how do you expect these biases to impact your analysis. Might a bias here impact the differing conclusions between here and your previous works?

*Response: We intend to discuss that there are errors associated with these input maps, and how we partly account for these uncertainties in this study. We developed more discussion on these, including:*

*(1) Adding a new discussion paragraph on Kellndorfer NBCD biomass products. It reads as "Second, we assume remote sensing-derived NBCD biomass products were well calibrated by field-derived biomass. However, the correlation coefficients between observed and predicted biomass were estimated to be 0.62-0.75 in the PNW region (Kellndorfer et al., 2012). And at 30 m pixel level, NBCD biomass values were biased with a large number of zero biomass values that had predictions in local biomass products (Huang et al., 2015). Discrepancies in biomass values between remote sensing- and field-derived data lead to biased stand age, as well as associated carbon stocks and fluxes. These were addressed in this study by imposing 20% error to pixel level biomass estimates and replacing zero biomass by the mean biomass of neighboring forest pixels with the same forest type and site productivity." at lines P13L13 to P13L19.*

*(2) Adding more discussion on forest type group, it reads as "Accuracy of forest type group map in the PNW region ranges from 61% to 69% (Ruefenacht et al., 2008); besides, forest type groups for some pixels undefined from original data were assigned as the forest types of the nearest pixels. For the same biomass value, inferred stand age and estimated carbon fluxes can vary greatly given difference in forest type group (Fig. 4 & Fig. 6); however, forest type group induced biases in NEP were not accounted due to the lack of information on associated errors from the spatial assignment of forest type group." at lines P14L5 to P14L9. Uncertainties and associated errors from forest type group were not accounted due to the lack of information on those uncertainties. At the very least a confusion matrix would be required to characterize the probability mis-assignment and the probable alternative assignment of forest type group. Unfortunately such a confusion matrix was not made available for this mapped data product.*

*(3) Developing more discussion on ADS data, and now it reads as "The ADS dataset is known to be limited by the areas flown in the survey years, and likely underestimate the number of trees killed by bark beetles but likely overestimate the area of affected stands (Meddens et al., 2012). Uncertainties from ADS dataset have important consequences for the carbon balance and flux estimates from bark beetle outbreaks, part of them were accounted for by Ghimire et al. (2015)." at lines P14L9 to P14L13.*

*We expect errors from input map will bring bias in stand age and NEP estimates, so we propagated part of these uncertainties into our estimates, and provided a distribution of estimates, including quantiles and standard deviation, instead of a single value.*

P11L27-28 Introducing new information in figures which are not described in the results. This should not be the case. If the figure and comparison is really needed then it should be included in the results and be part of the experimental design. or could be moved to SI.

*Response: We divided this paragraph into three parts. The sentence of introducing comparison was moved to "Materials and Methods" at lines P7L21 to P7L23, comparing results and Fig. 10 (now Fig. 9) was moved to "Results" at lines P11L28 to P11L31, and we keep the discussion on discrepancy from comparison in "Discussion" section at lines P14L27 to P15L4.*

P12L12-15 New analysis should not be introduced in the discussion. Also Figure 11 did not appear in the results either. Again, if these comparison and figure is needed then make it part of the experimental design and introduce it in the results section first.

*Response: Similar to response to previous comment, we divided this paragraph into three parts. The sentence of introducing comparison was moved to "Materials and Methods" at lines P7L23 to P7L24, comparing results and Fig. 11 (now Fig. 10) was moved to "Results" at lines P11L31 to P12L4, and we keep the discussion on discrepancy from comparison in "Discussion" section at lines P15L5 to P15L10.*

P12L14-15 "...distribution agrees well with that for our undisturbed..." poor working rephrase.

*Response: This sentence was edited as "Overall, the pattern of FIA-derived age distribution matches well with that derived from our study, but with our study having consistently lower forest areas at age classes larger than 20." at lines P12L1 to P12L3.*

P12L18, P13L4,L18 Multiple definitions of what is a young forest. Can you reconcile these?

*Response: P12L18: We replaced "stand of young ages" by "stand with ages ranging from 0 to 24" at lines P15L6 to P15L7.*
 *P13L4: We deleted "young" from "young regenerating forests" at line P16L4.*
*P13L18: We replaced "relatively mature (>24 year old) forests" by "undisturbed forests during 1986-2010" at line P16L18.*

P12L22 "A portion of this difference can be attributed to smaller net carbon losses..." If I understand correctly here you mean greater loss / more negative? Comparing between -4 TgC and -7 TgC? Not clear.

*Response: Correct. This study reports more net carbon losses (-7 Tg C $y^{-1}$) than -4 Tg C $y^{-1}$ reported in our previous work (Williams et al., 2014). The sentence was edited as "A portion of this difference can be attributed to larger net carbon losses from forestlands (-7 Tg y-1 carbon loss in this study vs. -4 Tg y-1 carbon loss in Williams et al.) due to recent (1986 to 2010) disturbance by either harvest or fire." at lines P15L21 to P15L23.*

P12L31-33 Is the PNW region representative of forestry in the US?

*Response: We meant western US, not US at P12L31-33 (now P15L30 to P15L32). Bark beetle outbreak in PNW is not representative of forests in western US, where rocky mountain north (RMN) and rocky mountain south (RMS) regions contribute more to the net carbon release due to bark beetle disturbance (36%, 35%), while less (1%) in Pacific Southwest (PSW) region (Ghimire et al. 2015).*

P13L19 Good to see some comparison with other studies. Are there any more available to broaden the discussion?

*Response: Thank you. We have current discussions on NEP comparisons.*

Figures

All of the figure captions need to be expanded to make clear where the data / analysis from each figure comes from and any key features. Also there appears to be substantial repetition of the disturbance figure. Can the figures be re-arrange to minimize this / move some of these maps to the SI.

*Response: We edited all the figure captions to include more detailed descriptions, and deleted Fig. 1.*

*We provided two figures on forest disturbance, one is year of last disturbance map from NAFD, MTBS and ADS, and the other one is integrated disturbance map (disturbance year and type), which is one of the resulting figures and new contribution to current disturbance maps.*

Figure 5. These NEP do not show C loss, even though your analysis does. These figures reinforce the confusion between whether or not you are analysis the C balance of the ecosystem as a whole or just the live biomass. If you are analyzing the whole ecosystem the NEP would surely be negative directly after disturbance due to litter and soil C turnover?

*Response: We analyzed the whole ecosystem, and Fig. 5 (now Fig. 4) was replaced by a figure displaying full range of NEP after harvest (Fig. 4a, 4b), fire (Fig. 4c, 4d) and bark beetle (Fig. 4e, 4f). More texts were added in caption of Fig. 5 (now Fig. 4), "The typical pattern of NEP following a disturbance involves a large negative value immediately after disturbance, a rise for a number of years to reach a maximum rate of carbon uptake, and then a gradual decline to a steady state."*

Figure 10. In your analysis are "Years Since Disturbance" and "Stand Age" the same thing? If so why in the same figure are you referring to this by different names. Particularly as in the caption you refer to both as "Stand age".

*Response: In our study, for undisturbed or stand-replacing disturbed forests, "years since disturbance" and "stand age" are the same thing; while for partial disturbed forest, they are not. We used term "years since disturbance" in this paper, and Pan et al. (2011) used term "stand age", so we keep both to be consistent with original paper. We edited the figure caption of Fig. 10 (now Fig. 9).*

*Response: Thanks for suggested citation, we have included it in our introduction at line P3L24.*

The discussion should compare using the CASA model and using the Raymond et al. approach which relies on an FIA driven empirical model, the Forest Vegetation Simulator (FVS). What are the advantages and disadvantages of each approach, and do they yield similar results (the regions are different but still may be able to compare results for one or two forest types). I also suggest that the discussion should explore in more depth the many assumptions and inferences that have to be made to estimate time since disturbance for "undisturbed" pixels (section 2.2.2). For example, the Kellndorfer biomass map used to estimate biomass of "undisturbed" pixels has fairly high uncertainty at the pixel level; some pixels were assigned forest types based on a nearby neighbor pixel, etc. By the way, the title of section 2.2.2 is an oxymoron – if the pixel is "undisturbed" there should not be a time since disturbance. So instead of "undisturbed" the authors should use a different term to identify pixels that had no detected disturbance since 1986, perhaps something like "recently undisturbed".

*Response: We didn't compare CASA and FVS models in discussion for two reasons: (1) the process of carbon cycle model is not the main focus in this paper, which has been described and discussed in our prior work (Williams et al., 2012, Ghimire et al., 2012, Ghimire et al., 2015). Our objective related to CASA is to use CASA-derived carbon stock and flux trajectories to do mapping; (2) it's not appropriate to compare carbon trajectories developed for different study area. Both CASA and FVS models heavily depend on FIA data, without same/similar FIA input, the comparison won't make a solid point. But when we moved to Rocky Mountain region in our future work, it sounds a good idea to make such comparison.*

*We added a new discussion paragraph on Kellndorfer NBCD biomass products at lines P13L13 to P13L19. It reads as "Second, we assume RS-derived NBCD biomass products were well calibrated by*

*field-derived biomass. However, the correlation coefficients between observed and predicted biomass were estimated to be 0.62-0.75 in the PNW region (Kellndorfer et al., 2012). And at 30 m pixel level, NBCD biomass values were biased with a large number of zero biomass values that had predictions in local biomass products (Huang et al., 2015). Discrepancies in biomass values between RS- and field-derived data lead to biased estimates in stand age and associated carbon stocks and fluxes. These were addressed in this study by imposing 20% error to pixel level biomass estimates and replacing zero biomass by the mean biomass of neighboring forest pixels with the same forest type and site productivity."*

*We also added more discussion on forest type group at lines P14L5 to P14L9. It reads as "Accuracy of forest type group map in the PNW region ranges from 61% to 69% (Ruefenacht et al., 2008); besides, forest type groups for some pixels undefined from original data were assigned as the forest types of the nearest pixels. For the same biomass value, inferred stand age and estimated carbon fluxes can vary greatly given difference in forest type group (Fig. 4 & Fig. 6); however, forest type group induced biases in NEP were not accounted due to the lack of information on associated errors from the spatial assignment of forest type group."*

*The title of section 2.2.2 was edited as "Time since disturbance for recently undisturbed forest pixels".*

Specific comments

The title is too long and redundant. Suggest "High-resolution forest carbon flux mapping in the Pacific Northwest with disturbance legacies inferred from remote sensing and inventory data". Could also leave out "in the Pacific Northwest".

*Response: We apologize that our title didn't fully convey main objectives and focuses of this manuscript, we edited the title as "High-resolution mapping of time since disturbance and forest carbon flux from remote sensing and inventory data to assess harvest, fire, and beetle disturbance legacies in the Pacific Northwest".*

p. 1 line 22: delete the second "probabilistic,"

*Response: Second "probabilistic" was deleted as suggested at line P1L21.*

p. 1 line 26: re-word so that it does not appear that tracts of land can somehow "see".

*Response: "seen" was deleted from the sentence at line P1L24.*

p. 2 line 13: replace "is itself a sort of record of" with "reflects"

*Response: Replaced as suggested at line P2L16.*

p. 2 line 14: replace "general" with "predictable rate of"

*Response: Replaced as suggested at line P2L16.*

p. 2 line 22-23: needs some rewording. The idea is that it is important to include smallscale disturbances down to some minimum threshold, not that disturbances typically are at this small scale.

*Response: The sentence was edited as "characterization of time since disturbance across landscapes at a scale of being able to detect small-scale disturbance events, typically around 100 m or less." at lines P2L25 to P2L26.*

p. 2 lines 27-28: add "at smaller scales" to the end since national forests inventories can provide useful guidance only at larger scales. But importantly note, it is possible to conduct field inventories at very small scales, so the statement is not very correct at all, only partially correct with respect to national forest inventories.

*Response: "at smaller scales" was added at the end of this sentence at line P2L31.*

p. 3 lines 11-12: One objective is clearly stated. What are the others? The last sentence of this paragraph seems to be another objective, but then, I'm confused as to whether the purpose is to develop a method for large-scale monitoring and management, or small-scale, or both?

*Response: We rewrote the last paragraph in Introduction at lines P3L27 to P4L2, where now all of the objectives are listed. It reads as "This study estimates and maps time since disturbance at a fine scale of 30 m from RS-derived products and FIA-derived biomass growth curves, and then maps net ecosystem productivity (NEP) based on disturbance history, time since disturbance and carbon flux legacy. The specific objectives in this study are to: (1) introduce a method for inferring a pixel's representative time since disturbance from RS-derived biomass and disturbance products at the 30 m resolution; (2) map NEP based on pre-existing, model-derived carbon stock and flux trajectories that describe how NEP changes with time following harvest, fire, or bark beetle disturbances of varying severity; (3) propagate uncertainties from RS-derived biomass products and FIA into uncertainty quantification of stand age and NEP. Our research represents an approach to map carbon stocks and fluxes at a high resolution across the conterminous US in support of national carbon monitoring, reporting, and management."*

*The last sentence of this paragraph is misleading, we have edited it as shown above. Our method will be used to map carbon stock and fluxes at a fine scale in the conterminous US.*

p. 3 lines 27-28. Terminology again – "undisturbed" pixels by definition should not have a time since disturbance.

*Response: "undisturbed forest pixels" has been edited as "recently undisturbed forest pixels" at line P4L12.*

p. 3 line 32: Biomass curves were developed by forest type group and productivity class. How were these 2 classes allocated to the 5 NEP classes described on p. 4 lines 2-4?

*Response: Biomass curves by forest type group and productivity class were derived from FIA and used to infer stand age for forest pixels undisturbed during remote sensing observation period.*

*NEP curves were derived from CASA carbon cycle process model with inclusion of disturbance processes. Biomass curves were used to adjust model's rates of NPP and wood turnover for each forest type group and productivity class. At the final stage of the modeling, the disturbance processes imposed stand-replacing harvest, fire or insect-induced partial disturbance to generate carbon stock and flux curves as a function of time since disturbance, and are specific to forest type group, site productivity class, disturbance type and disturbance severity. Detailed descriptions of CASA model were provided at lines P7L29 to P9L9 now.*

p. 3 line 35: add citation after ". . .varying severity".

*Response: Citations "Ghimire et al., 2012, Williams et al. 2014, Ghimire et al., 2015" were added after "... varying severity" at line P4L19.*

p. 6 line 26: replace "stand" with "standard"

*Response: Replaced as suggested at line P7L19.*

p. 8 line 21: sentence that begins with "Again" needs editing.

*Response: This sentence was edited as "Again in contrast, bark beetle outbreak areas for low and high productivity classes are similar in Douglas-fir forests, but beetle outbreak occurrence was about three times more likely in low productivity sites." at lines P10L27 to P10L29.*

p. 9 lines 24-25: the imprint is not so clear to me. Maybe need to highlight somehow on the graphic.

*Response: We included the position description of Biscuit fire in the bracket at line P12L11, "bottom left of Fig. 8a, also refer to bottom left of Fig. 5a & 5b".*

p. 10 lines 12-14: One could argue that inventory data does not provide such a reliable estimate of biomass/age. Both of these variables can be rather difficult to measure/estimate especially with respect to the selection of biomass equations, but also the difficulty of assigning a stand age to stands that are uneven-aged.

*Response: We added more discussion on the first assumption as suggested at lines P13L6 to P13L12. It reads as: "However, both stand age and biomass are difficult to measure and estimate, especially considering the difficulty of assigning a stand age to uneven-aged forest stands, as well as selecting appropriate species-specific biomass equations (Parresol, 1999). If FIA ages are older than actual stand ages, the associated forest biomass will be underestimated, and stand age inferred from biomass products will be overestimated. And younger FIA ages than actual ages will result in an overestimation in biomass accumulation, but an underestimation in biomass-inferred stand ages. Though a possible bias in stand ages, our estimates of carbon stocks and fluxes are not likely to be largely adjusted by a stand age bias within 5 years (Williams et al., 2012)."*

p. 12 lines 1-7: not statedâA˘TFIA does not do a good job of detecting recent disturbances because the remeasurement cycle in the PNW is about 10 years, so the average time lag of the data at any point in time is at least 5 years.

*Response: We added the suggested point in this paragraph at lines P15L8 to P15L9, now it reads as "
[revised manuscript text omitted]